# The Fabaceae in Northeastern Mexico (Subfamily Caesalpinioideae, Mimosoideae Clade, Tribes Mimoseae, Acacieae, and Ingeae)

**DOI:** 10.3390/plants13030403

**Published:** 2024-01-30

**Authors:** Eduardo Estrada-Castillón, José Ángel Villarreal-Quintanilla, Gerardo Cuéllar-Rodríguez, Juan Antonio Encina-Domínguez, José Guadalupe Martínez-Ávalos, Arturo Mora-Olivo, Jaime Sánchez-Salas

**Affiliations:** 1Facultad de Ciencias Forestales, Universidad Autónoma de Nuevo León, Linares 67700, Mexico; aeduardoestradac@prodigy.net.mx (E.E.-C.); luis.cuellarr@uanl.mx (G.C.-R.); 2Departamento de Botánica, Universidad Autónoma Agraria Antonio Narro, Saltillo 25315, Mexico; javillarreal00@hotmail.com (J.Á.V.-Q.); jaencinad@gmail.com (J.A.E.-D.); 3Instituto de Ecología Aplicada, Universidad Autónoma de Tamaulipas, Ciudad Victoria 87019, Mexico; jmartin@uat.edu.mx; 4Facultad de Ciencias Biológicas, Universidad Juárez del Estado de Durango, Gómez Palacio 35010, Mexico; j.sanchez@ujed.mx

**Keywords:** Leguminosae, clade Mimosoideae, taxonomy, diversity, northeastern Mexico

## Abstract

A synoptic compendium of the legumes of the Mimosoideae clade in northeastern Mexico is presented for the first time, including changes in their botanical nomenclature and retypification of genera. Furthermore, based on new information recently published, the taxonomic limits of several new genera segregated from *Acacia* (*Acaciella*, *Mariosousa*, *Senegalia,* and *Vachellia*) and *Prosopis* (*Neltuma* and *Strombocarpa*) are clarified and included. Based on field work, collection of botanical samples over the past 30 years, and reviewing botanical materials in national and international herbaria, we have completed the diversity of legumes of the Mimosoideae clade of northeastern Mexico. Three tribes (Acacieae, Ingeae, and Mimosaeae), 22 genera, 92 species, and 19 infraspecific categories were recorded. Only the genus *Painteria* is endemic to Mexico. Eighty-eight species are native to Mexico, and four are exotic: *Acacia salicina*, *Neptunia prostrata*, *Neltuma chilensis* and *Albizia lebbeck*. Twenty-eight species are endemic to Mexico, nine species are endemic to northeastern Mexico, and four species are endemic to only one state in Mexico. The 22 registered genera represent 44% and 65% of the generic flora of the Mimosoideae clade for Mexico and the planet, respectively, while the 92 species registered represent 3% and 18% of the species of the clade Mimosoideae for the planet and Mexico, respectively. According to the new nomenclature of legumes, the number of genera in the Mimosoideae clade in northern Mexico has increased from 19 to 24.

## 1. Introduction

Fabaceae [1] (Leguminosae) is considered the third largest family of angiosperms after Asteraceae (1623 genera and 24,700 species) and Orchidaceae (730 genera and 28,000 species), with approximately 751 genera and about 19,500 species [2,3]. Many species of the family Leguminosae are very important in human nutrition [4], healthy traditional diets [5], sustainable farming systems [6], nitrogen fixation [7], forage [8], shade for livestock [9], medicine [10], phytochemicals [11], traditional cropping and harvesting systems (milpa) [12], ornamental [13], timber [14], charcoal [15]. They have a cosmopolitan distribution, found in almost all types of climate and vegetation, although with greater diversity in the tropics and subtropics, standing out in diversity in low- to medium-altitude plains or hills. This great diversification of legumes on the world occurred almost in synchrony with other groups of angiosperm plants, as well as mammals, insects, and birds, approximately in the Early Tertiary [16]. The distribution and overlap patterns of legume species have allowed us to identify areas of endemism in Mexico where species of the genera of this clade stand out, such as *Mariosousa*, *Mimosa,* and *Vachellia* [17].

According to characters of aestivation of sepals and petals, it has been recognized that Leguminosae consists of three subfamilies: Mimosoideae (actinomorphic, radial symmetry, valvate aestivation) with few to many stamens; Caesalpinioideae (somewhat zygomorphic, bilateral symmetry, imbricate aestivation), where the adaxial petal is more internal than the lateral petals; and Papilionoideae (zygomorphic, bilateral symmetry, imbricate aestivation), in which the adaxial petal is more external than the lateral petals [18,19,20]. Between 1900–1997, the taxonomic classification status of this group of plants fluctuated, recognizing two main taxa, subfamily [21,22], or family [23,24,25,26]. Over the past three decades, molecular biology studies [20,27,28,29,30,31,32,33,34,35,36,37] have allowed for the solving of, in part, the problem in taxonomy concerning the paraphyly of the subfamily Caesalpinioideae, allowing for the recognition of the subfamily Mimosoideae as a related clade within it, and both together constituting a monophyletic group. Although gaps exist within the classification of this subfamily, especially in proving monophyly at the tribal and generic rank, efforts to resolve them are being completed [38]. As of 2017, the legume phylogeny working group [7] has proposed a new classification system for this family, which also partly resolves the longstanding problem of the paraphyly of the subfamily Caesalpinioideae. These six new subfamilies have been reaffirmed with studies of phylogenomics and plastomes DNA [39,40,41]. Nomenclatural changes to the rank of subfamily, tribe, and genus in the family Leguminosae have dramatically altered the conformation of these monophyletic groups in their origin, evolution, ecology, distribution, and management. Some groups of legumes today are proven to be monophyletic, such as *Acaciella*, *Mariosousa*, *Senegalia*, and *Vachellia*, all of them segregated from *Acacia*; this has allowed for a more homogeneous and clear definition of their genetic, morphological, and ecological relationships and an easier understanding of the separation of these groups.

The diversity of native vascular plants in Mexico has been estimated at 23,314 species and 2854 genera [42]. Leguminosae forms the second largest family of phanerogamic plants in Mexico, represented by 26 tribes, 135 genera, and 1724 species [43]. More recently, the number of legume species in Mexico has increased to 1903 [42], below Asteraceae (3057 species) and above Orchidaceae (1213 species) and Poaceae (1047 species). After Brazil, Mexico ranks second in diversity of legume species in the New World, with a total of 1893 species recorded [13]. Studies of legumes in the north of Mexico over time have shown a general picture of their diversity. In the northwestern region, 285 species and 77 genera of Leguminosae are present on the peninsula of Baja California and the adjacent islands [44]. In Sonora [45], 54 genera and 199 species of Leguminosae have been recorded in the Rio Mayo area; [46] reports 66 genera and 369 species of Leguminosae for the Sonoran Desert. The presence of 352 leguminous taxa for the state of Sonora has been reported [47], with *Dalea* (36), *Desmodium* (30 species), and *Astragalus* (20 species) the most diverse genera. For the state of Chihuahua, 29 genera and 65 species of Legumes in the Basaseachi Waterfall National Park and adjacent areas have been recorded [48], highlighting the genera *Desmodium* (13 species), *Dalea* (7 species) and *Phaseolus* (4 species); 45 genera and 138 species of legumes for the central part of this state have been recorded [49], and 18 genera and 48 species of legumes for the Laguna of Babicora of the central–western portion of that state have been reported [50], highlighting the genera *Astragalus*, *Dalea,* and *Phaseolus*, all with 5 species. In northeastern Mexico, for the state of Coahuila, researchers have documented 54 genera and 213 species of Legumes [51] with *Dalea* (27 species), *Astragalus* (17 species), *Senna* (13 species), *Acacia* (s.l.) (12 species), *Mimosa* (9 species), *Phaseolus* (8 species), and *Vicia* (7 species) as the most diverse genera; 55 genera and 206 species have been recorded for the same state [52]. In the state of Nuevo León, researchers have recorded 55 genera and 206 species of legumes [53], with *Dalea* (28 species), *Desmodium* (19 species), *Mimosa* (15 species), *Senna* (14 species), *Phaseolus* (14 species), *Astragalus* (13 species), *Acacia* (s.l.) (12 species), and *Crotalaria* (9 species) as the most diversified genera. From Tamaulipas, there exist isolated studies documenting legumes which are of beekeeping importance [54]. In the Sierra of San Carlos, 37 species of legumes have been recorded [55], highlighting the presence of *Gleditsia triacanthos*, a rare legume species in Mexico.

The climatic, physiographic, and edaphic heterogeneity of northeastern Mexico sustains a rich and varied flora, where legumes play a predominant role and are widely useful for the inhabitants. The environmental heterogeneity [56] in northeastern Mexico allows for the presence of tropical, subtropical, semi-arid, arid, temperate and cold climates, associated with an extraordinarily heterogeneous relief in northeastern Mexico. This allows for the existence of multiple plant associations, where legumes participate as predominant elements in the different types of scrub in the low plains [49,57] and desert scrubland of the Mexican Altiplano [57], and are frequent elements in oak and oak–pine forests, as well as gypsophilous grasslands and subalpine meadows [57]. In this context and being one of the most economically and ecologically important groups of plants worldwide, modern taxonomic knowledge of the diversity of legumes will allow for the evaluation of the importance of these resources for better management and conservation of many of their polyfunctional species.

Given the absence of a new complete taxonomic study, a renewed study of the recent nomenclature for this group of plants is required due to its enormous richness and the variety of uses of Leguminosae in northeastern Mexico, and, based on the new classification of the family, subfamilies, and genera, the objectives of this study are to: provide a compendium of the family Fabaceae (subfamily Caesalpinioideae, Mimosoideae clade, tribe Ingeae, tribe Mimoseae, and tribe Acacieae) in northeastern Mexico; provide a synopsis of the total species of this clade in northeastern Mexico; and update information concerning their new nomenclatural changes, ecology, distribution, and main uses.

## 2. Results

In northeastern Mexico, the diversity of the Mimosoideae clade is represented by 89 species classed into three tribes (Acacieae, Ingeae, and Mimoseae). The tribe Acacieae is represented by five genera (*Acacia*, *Acaciella*, *Mariosousa*, *Senegalia,* and *Vachellia*) (Figure 1), 27 species, and 5 infraspecific categories; the tribe Mimoseae is represented by six genera (*Desmanthus*, *Leucaena*, *Mimosa*, *Neltuma*, *Neptunia,* and *Strombocarpa*) (Figure 2), 41 species and 10 infraspecific categories; and the tribe Ingeae is represented by 11 genera (*Albizia*, *Calliandra*, *Cojoba*, *Ebenopsis*, *Havardia*, *Inga*, *Lysiloma*, *Enterolobium*, *Painteria*, *Pithecellobium*, and *Zapoteca*) (Figure 3), 24 species, and 7 infraspecific categories, totaling 22 genera, 92 species, and 19 infraspecific categories.

### 2.1. Diversity and Endemism of Mimosoideae Clade in Northeastern Mexico

Of the six genera of the tribe Mimosoideae recorded in northeastern Mexico, four of them (*Desmanthus*, *Leucaena*, *Neltuma,* and *Strombocarpa*) are only distributed in America; the other two (*Mimosa* and *Neptunia*) are also distributed in Asia, Africa, India, Madagascar, and Australia. Of the 11 genera of Ingeae recorded in northeastern Mexico, 10 of them (*Calliandra*, *Cojoba*, *Ebenopsis*, *Havardia*, *Inga*, *Enterolobium*, *Lysiloma*, *Painteria*, *Pithecellobium*, and *Zapoteca*) are native to America, and only 1 (*Albizia*) is also distributed in Africa, Asia, and Malesia.

Of the 21 genera recorded, only *Painteria* is endemic to Mexico. Nine of these genera have at least three species: *Mimosa* (19 species), *Vachellia* (11 species), *Desmanthus* (8 species), *Senegalia* (8 species), *Calliandra* (7 species), *Leucaena* (5 species), *Zapoteca* (4 species), and *Pithecellobium* (3 species). Of the 92 species of the Mimosoideae clade recorded, 88 of them are native to Mexico, although some of the species have a wide distribution on the American continent, such as *Desmanthus paspalaceus*, with bicentric distribution in Mexico (Tamaulipas and Veracruz) and in South America (Uruguay, Paraguay, Bolivia, and Argentina); *Mimosa guranitica* in northeastern Mexico (Tamaulipas), and distantly disjunct in South America (Paraguay, Brazil, and Argentina); and *Calliandra tergemina* ssp. *tergemina* (from Mexico through Central America to Colombia and Venezuela). Three of the exotic species registered are cultivated as ornamental; they are *Acacia salicina*, *Albizia lebbeck*, and *Prosopis chilensis.* The 22 genera and 92 species of the Mimosoideae clade recorded in northeastern Mexico represent 69% and 25% of genera and species, respectively, of the data recorded for Mexico [58]. The Mimosoideae clade groups almost 50 genera and 3000 species [2], although with recent changes in legume systematics [7], genera have increased by several more.

Emphasizing the physiographic, climatic, and edaphic similarities of the north of Mexico with the south of the USA (states of California, Arizona, New Mexico, and Texas) to give a slightly more precise idea of true endemism’s and using Rzedowski’s Megamexico 1 [59] as a reference point, we recorded 27 species endemic to Mexico, 9 species endemic to northeastern Mexico, 8 species endemic to the north of Mexico and Texas (southern region), and 4 species endemic to only a single state in Mexico (Table 1).

Of the 29 species of legumes endemic to Mexico, 5 are herbaceous, 14 are shrubs, and 10 are trees. All genera of Acacieae in northeastern Mexico have at least one legume species endemic to Mexico. *Mariosousa*, *Senegalia,* and *Vachellia* are the genera that have the largest number of endemic species (at least 3 each) of Acacieae in northeastern Mexico. No genus of this tribe has species endemic to a single state in northeastern Mexico. The tribe Mimoseae is the one that has the largest number of endemic legume species (16). Of the 6 genera of Mimoseae, 4 have at least one species endemic to Mexico. In northeastern Mexico, *Mimosa* is the genus with the largest number of endemic species (10). *Mimosa* is also the only genus in northeastern Mexico that has species distributed in a single state, *Mimosa martin delcampoi*, in Tamaulipas and *M. monclovensis* and *M. unipinnata*, both in Coahuila. *Mimosa* is the only genus of Mimoseae in northeastern Mexico that shares 4 species with the southern Texas region.

Despite having the largest diversity in number of genera in northeastern Mexico, the tribe Ingeae is the one with the smallest number of endemisms in the study area; only *Painteria* and *Zapoteca* have a single species each.

### 2.2. Diversity and Distribution of Legumes in Different Regions in Mexico and Southern USA Compared to Northeastern Mexico

There is a range of between 7 and 24 genera of the Mimosoideae clade in different areas of Mexico and southern USA; the lowest diversity occurs in the state of California, the Valley of Mexico, the Sonoran Desert, the state of Durango, and the state of Texas, while the highest diversity occurs in southern Mexico, the state of Chiapas, the Novo Galicia region, northeastern Mexico, the state of Yucatán, the Bajio, the state of Tabasco, and the state of Quintana Roo. Likewise, the range of species present in the different areas is wide (see Appendix A), with the Valley of Mexico and California being the ones with the lowest number of species, while Chiapas and the Novo Galicia region have the highest number of species (Table 2).

Compared to the other 11 areas, northeastern Mexico is the third in terms of the greatest diversity of genera of the Mimosoideae clade in Mexico, followed by the Novo Galicia region, in the center–west of the country. Northeastern Mexico is also the third richest region in species of the clade Mimosoideae behind the state of Chiapas and the Novo Galicia region. Acacieae is the tribe with the smallest number of genera of these areas in Mexico, while Ingeae is the tribe with the greatest number of genera, being more diverse in the state of Chiapas, the state of Yucatán, and northeastern Mexico. The greatest species diversity of the tribe Acacieae occurs in the northeastern Mexico, and the Novo Galicia region, while Mimoseae is more diverse in the Novo Galicia region and the state of Chiapas. The greatest species diversity of the tribe Ingeae is by far in the state of Chiapas, followed by the state of Tabasco and the Novo Galicia region.

### 2.3. Mimosoid Clade

Herbaceous, sub-shrubs, mainly shrubs or trees, unarmed or armed with prickles or spinescent stipules. Glands often present on the petiole, arising in the insertion of pinnae on rachis and along it. Leaves bifoliolate, pinnate or bipinnate, mostly paripinnate. Inflorescences solitary or fasciculate, arranged in spherical or ovoid capitula, spikes, panicles, or racemes. Flowers mostly bisexual, less commonly unisexual, sterile flowers sometimes present, radially symmetric. Sepals 3–6, united or free. Petals 3–5, free or united, rarely absent, aestivation valvate or imbricate, with the adaxial petal the innermost. Stamens diplostemonous or haplostemonous; sometimes 3, 4, or 5 or many (100 or more); free or basally united; similar or heteromorphic. Staminodes sometimes present, anthers with or without a stipitate or sessile gland. Ovaries 1 to many; ovulate. Fruit is a pod, 1 to many seeds; dehiscent along one or both sutures, also lomentaceous or craspedium; valves thin, flexible, chartaceous, or thick and woody; inertly or explosively dehiscent or indehiscent; straight or curved, sometimes coiled. Seeds commonly with a pleurogram on both faces, sometimes with fleshy arils or winged.

Taxa belonging to the Mimosoid clade group number almost 90 genera and 340 species [7]; they are distributed mainly in tropical and subtropical areas, being more abundant in rainforests, deciduous forests, dry savannas, tropical and subtropical scrublands, and desert areas [60]. This clade includes four tribes: Mimoseae Bronn, Ingeae Benth., Acacieae Dumort., and Mimozygantheae Burkart [2]; the first three are present in northeastern Mexico. The genera of the different tribes can easily be differentiated by the number of stamens per flower; the tribe Mimoseae always has 10 stamens or fewer per flower. The tribes Acacieae and Ingeae have more than 10 stamens per flower, but those are free in Acacieae, while in Ingeae, they are monadelphous [61]. The low plains in northeastern Mexico show a shrubby physiognomic pattern often dominated by different species of legumes, and quantitatively in terms of canopy cover and density, legumes are one of the most important elements of the landscape [57,62], where species of the genera *Vachellia*, *Senegalia*, *Havardia*, *Ebenopsis*, *Neltuma,* and *Mimosa* stand out, the last two with several species constituting invasive weeds in disturbed areas.

Tribes of the Mimosoideae clade present in northeastern Mexico.
1A.Flowers with 10 stamens or fewer**Tribe Mimosaeae**1B.Flower with more than 10 stamens22A.Stamens free**Tribe Acacieae**2B.Stamens united, forming a basal tube**Tribe Ingeae**

Taxonomic treatment

Tribe ***Acacieae*** Dumort., Anal. Fam. Pl. 40: 1829.

Trees, shrubs or herbaceous, unarmed or armed with spinescent stipules or prickles (thorns) irregularly distributed along the stem. Leaves bipinnate. Glands present on the petiole (adaxially) and rachis; the pinnule tips may have paraphyllidia. Leaflets generally numerous, commonly opposite in the pinnae rachis. Inflorescences solitary or fascicled, arranged in dense spheric capitula, cylindric spikes or terminal panicles. Flowers 4–5-merous, mostly hermaphrodites, white to yellow. Calyx sepals connate, valvate. Corolla with the petals valvate, rarely absent. Stamens abundant, exerted, the filaments free. Ovary sessile or stipitate. Ovules many, biserially arranged. Fruit dehiscent or indehiscent, exocarp not separated from endocarp, not forming an envelope around each seed. Seed compressed, commonly oblong or elliptic, testa hard, commonly black or brown, pleurogram present, aril present or absent.

According to the segregation and nomination of new genera, the tribe Acacieae is composed of seven genera worldwide: *Acacia* Mill., *Acaciella* Britton and Rose, *Mariosousa* Seigler and Ebinger, *Parasenegalia* Seigler and Ebinger, *Pseudosenegalia* Seigler and Ebinger, *Senegalia* Raf., and *Vachellia* Wight and Arnott, including almost 1380 species [63,64,65].

In northeastern Mexico, five of the genera are present. Species of *Senegalia* and *Vachellia* are often predominant in the low plains and mountain slopes; they constitute not only the dominant elements of the landscape, but also in terms of density and aerial coverage in several plant associations and are very common in disturbed places.
1A.Leaves simple phyllodia; seeds with pulpy aril forming a cap or encircling seed; exotic, cultivated plants***Acacia***1B.Leaves bipinnate; seeds without pulpy aril; native, wild plants22A.Prickles or spinescent stipules absent32B.Prickles or spinescent stipules present43A.Petiole without gland; inflorescences arranged in spheric capitula***Acaciella***3B.Petiole with gland; inflorescences arranged in elongated spikes***Mariosousa***4A.Branches without prickles; stipules spinescent, commonly paired; inflorescences arranged in spheric capitula; ovary sessile or subsessile ***Vachellia***4B.Branches with prickles irregularly distributed along the stem, often solitary (not in pairs); stipules membranous, not spinescent; inflorescences arranged in spheric capitula or elongated spikes; ovary elevated on a short axis***Senegalia***

***Acacia*** Mill. Gard. Dict. Abr. ed. 4, vol. 1. 1754.

**Type**: *Acacia penninervis* Sieber ex DC., Prodr. 2: 52. 1825.

Shrubs or trees, commonly unarmed. Stipules caducous, rarely woody, spinose with 1 globose gland. Leaves almost always alternate even-pinnate of phyllodic, the leaves modified to phyllodes (flattened) and enlarged petiole without leaflets, usually marginally glandular and in the apex. When pinnate, with 1–50 pairs of pinnae per leaf. Leaflets 8–70 pairs per pinna. Inflorescences solitary, fascicled, pseudoracemes or pseudo panicles, axillar or terminal, arranged in spheric capitula or cylindrical spikes. Flowers symmetric (actinomorphic), 4–5-merous. Calyx and corolla cup-shaped, corolla yellow or cream. Stamens 20–150, yellow to cream white. Fruit erect or pendulous, commonly flattened, straight to falcate, linear to oblong, commonly dehiscent, papyraceous, leathery or chartaceous. Seeds 6–10, flattened, ovoid to elliptic, uniseriate.

Genus of ca. 1300 species indigenous to the Southern Hemisphere, almost 1000 species in Australia. Only the species belonging to subg. Phyllodinae remain in the genus *Acacia*; the rest of the species, especially those present on other continents, have been transferred to new genera, namely *Acaciella*, *Mariosousa*, *Parasenegalia*, *Pseudosenegalia*, *Senegalia*, and *Vachellia.* In northeastern Mexico, only 1 species, *Acacia salicina*, recorded as ornamental species in in parks or squares of private neighborhoods in Torreon, Coahuila. Rare.

***Acacia salicina*** T.L. Mitchell, Three Exped. Australia 2: 20. 1838. Basionym: *Acacia salicina* var. *varians* Benth., Fl. Austral. 2: 367. 1864. *Racosperma salicinum* (Lindl.) Pedley, Austrobaileya 2: 355. 1987. *Acacia varians* Benth., T.L. Mitchell, J. Exped. Trop. Australia: 132. 1848. *Acacia salix-tristis* F. Muell., Hooker’s Journal of Botany and Kew Garden Miscellany, 1853.

**Type**: Australia, Subtropical New Holland, 6-IV-1846, *T.L. Mitchell*, *104* (Lectotype: K779902!; Isosyntype: GH00058377!).

**Distinguishing features**: Shrub or tree, up to 12 m tall. Young branches pendulous. Leaves (phyllodes) alternate, pendulous, 4–18 cm × 0.4–3 cm, linear, elliptic to oblanceolate, straight or slightly falcate, glaucous, green to grey-green, glabrous, 1-veined. Foliar gland basal, sometimes 1 gland at the apex and 1 to 3 along the margins. Inflorescences axillar, shorter than phyllodes, fascicled in 2–8 capitula, glabrous, rarely puberulous. Fruit 3–12 × 0.6–1 cm, oblong, woody, beige or light brown, glabrous, the margins thickened, not constricted between the seeds. Seeds 4.5–6 × 3.5–4.5 mm, elliptic to black or dark brown, shiny, arranged longitudinally, with a red funicle that surrounds them.

**Representative examined material**: 4-XII-2023, *J. Sánchez-Salas 511* (UJED)

**Comments**: Species native to Australia, introduced in many countries as an ornamental plant. Registered only in the city of Torreon, Coahuila, cultivated as ornamental in parks of private colonies.

***Acaciella*** Britton and Rose, N. Amer. Fl. 23: 96. 1928.

**Type species**: *Acacia villosa* (Sw.) Willd., Sp. Pl. 4: 1067. 1806. *Mimosa villosa* Sw., Fl. Ind. Occ. 2: 982. 1800.

Unarmed, herbaceous (rarely), shrubs or trees, glabrous or pubescent. Leaves bipinnate. Petiolar gland absent. Pinnae 2 to many pairs. Leaflets few to many pairs per pinna. Inflorescences in racemes or panicles, arranged in spherical capitula or subglobose to oblong spikes. Flowers 5-merous; white, whitish, or yellow. Stamens numerous (100 or more per flower), free. Fruit stipitate, body linear to narrowly oblong, flat, papyraceous or membranous, dehiscent, opening from the apex downwards. Seeds suborbicular, flattened, lentil-like; funicule slender.

A Mexican genus, it includes 15 species [66], all distributed in Mexico except for *A. glauca*. From the south of the USA to Brazil and Argentina, inhabiting a wide range of relief and soils, mainly in grasslands, arid and semiarid scrublands, pine–oak forest, seasonally dry forest and tropical forest, 30–2700 m. Morphologically, the species of *Acaciella* and *Mariosousa* are very similar, since both include unarmed species, and in the absence of inflorescences and flowers they can be confused; however, they can be easily differentiated by the absence of a gland in the petiole leaf in *Acaciella*. In northeastern Mexico, 4 species and 4 infraspecific categories have been recorded.
1A.Leaflets 1.2–2 cm long or longer***A. tequilana*** ssp. ***tequilana***1B.Leaflets 1 cm long or shorter22A.Herbaceous, perennial, commonly prostrate***A. hartwegii***2B.Shrubs or small trees33A.Leaflets with reticulate or brochidodromous venation or pinnately veined ***A. lemmonii***3B.Leaflets with only 1 vein, prominent or barely visibly44A.Leaflets with inconspicuous reticulate venation or sometimes withonly the midrib***A. lemmonii***4B.Leaflets with conspicuous reticulate venation ***A. hartwegii***4A.Stems and foliage glabrescent; leaflets short ciliate or glabrous; pinnae 6 pairs per leaf or fewer (rarely 8 pairs)***A. angustissima*** ssp. ***texensis***4B.Stems and foliage glabrous or pubescent; pinnae 7 or more pairs per leaf55A.Branches, petiole, and leaf rachis glabrous or pulverulent, the trichomes white, appressed; pinnae 11–17 pairs per leaf; calyx glabrous; pod long beaked***A. angustissima*** ssp. ***angustissima***5B.Branches, petiole, and leaf rachis densely hispidulous or pilose, the trichomes yellow, erect; pinnae 18 or more pairs per leaf; calyx hirustuluos; pod abruptly rounded apically ***A. angustissima*** ssp. ***filicioides***

***Acaciella angustissima*** (Mill.) Britton and Rose ssp. ***angustissima.*** Basionym: *Acacia angulosa* Bertol., Fl. Guatimal. 4: 442. 1840. *Acaciella angulosa* (Bertol.) Britton and Rose, N. Amer. Fl. 23: 100. 1928. *Acacia elegans* M. Martens and Galeotti, Bull. Acad. Roy. Sci. Bruxelles 10: 312. 1843. *Acacia insignis* M. Martens and Galeotti, Bull. Acad. Roy. Sci. Bruxelles 10: 313. 1843. *Acacia boliviana* Rusby, Bull. New York Bot. Gard. 4: 348. 1907. *Acacia suffrutescens* Rose, Contr. U.S. Nat. Herb. 12: 409. 1909. *Acaciella suffrutescens* (Rose) Britton and Rose, N. Amer. Fl. 23: 103. 1928. *Acacia hirta* var. *suffrutescens* (Rose) Kearney and Peebles, J. Wash. Acad. Sci. 29: 482. 1939. *Acacia angustissima* var. *suffrutescens* (Rose) Isely, Sida 3: 372. 1969. *Acaciella breviracemosa* Britton and Rose, N. Amer. Fl. 23: 99. 1928. *Acaciella delicata* Britton and Rose, N. Amer. Fl. 23: 100. 1928. *Acacia delicata* (Britton and Rose) Bullock, Bull. Misc. Inform Kew 1939 (1): 1. 1939. *Acaciella costaricensis* Britton and Rose, N. Amer. Fl. 23: 101. 1928. *Acacia pittieriana* Standl., Pub. Field Mus. Nat. Hist., Chicago, Bot. Ser., 13: 489. 1937. *Acaciella rensonii* Britton and Rose, N. Amer. Fl. 23: 101. 1928. *Acaciella ferrisiae* Britton and Rose, N. Amer. Fl. 23: 101. 1928. *Acaciella talpana* Britton and Rose, N. Amer. Fl. 23: 101. 1928. *Acaciella smithii* Britton and Rose, N. Amer. Fl. 23: 101. 1928. *Acacia angustissima* subsp. *smithii* (Britton and Rose) Wiggins, Contr. Dudley Herb. 3: 232 (1942). *Acacia angustissima var. smithii* (Britton and Rose) L. Rico, Anales Jard. Bot. Madrid 58: 258. 2001. *Acaciella ciliata* Britton and Rose, N. Amer. Fl. 23: 101. 1928. *Acaciella santanderensis* Britton and Killip, Ann. New York Aca. Sci.: 140. 1938.

**Type**: Mexico, cordillera, 1840, *H.G. Galeotti 3303* (Isotype: K81920!).

**Distinguishing features**: Unarmed shrub or low tree, up to 8 m tall, with white appressed pubescence. Pinnae 11–17 pairs per leaf. Leaflets 20–40 pairs per pinna; always with flat margin; never involute, at least in the base. Inflorescences axillar or fascicles, arranged in spheric capitula. Fruit 4–9.3 × 1–1.6 cm, flattened, straight, chartaceous, glabrous, reticulate, acute at both ends with a terminal long curved beak.

**Representative examined material**: Coahuila: 27-VI-1936, *L. Wynd*, *F. 312* (MEXU). Nuevo León: 23-VII-2002, *E. Estrada 15051* (CFNL); 30-VII-2002, *E. Estrada 15552* (CFNL); 14-VII-2002, *E. Estrada 14895* (CFNL). Tamaulipas: 29-IX-1996, *G.E. García 65* (MEXU); 31-V-1999, *A. Mora-Olivo 7567* (UAT); 23-IX-1985, *M. Yanez 511* (UAT); *23-IX-1983*, *L. Hernández 695* (UAT); 1-VII-1994, *D. Seigler 14102* (UAT); X-1976, *Medrano 14102* (UAT).

**Comments**: This species (and the variety) comprises the distribution of the genus and is the most widely distributed species, recorded from the southern USA to Argentina in piedmont scrub, oak forest, and mixed pine–oak forests, 700–1600 m. The leaves and fruits are used as fodder for domestic livestock.

***Acaciella angustissima*** ssp. ***filicioides*** (Cav.) L. Rico, Kew Bull. 59 (2): 327. 2004. Basionym: *Mimosa filicioides* Cav., Ic. 1: 55 tab. 78 (1791). *Acacia filicioides* (Cav.) Trel., Annual Rep. Geol. Surv. Arkansas 1888 (4): 178. 1891. *Acaciella filicioides* (Cav.) Britton and Rose, N. Amer. Fl. 23: 100. 1928. *Acacia filicina* Willd. Sp. Pl. 4: 1072. 1806. *Acacia hirsuta* Schltdl., Linnaea 12: 572. 1838. *Acaciella hirsuta* (Schldl.) Britton and Rose, N. Amer. Fl. 23: 99. 1928. *Senegalia hirsuta* (Schldl.) Pittier, Man. Pl. Usual. Venez. (Suppl.): 36. 1939. *Acacia stipellata* Schltdl. Linnaea 12: 574. 1838. *Senegalia popayana* Britton and Killip, Ann. New York Acad. Sc. 35: 143. 1936. *Acaciella holtonii* Britton and Killip, Ann. New York Aca. Sci. 140. 1936. *Acaciella martensis* Britton and Killip, Ann. New York Acad. Sci. 35 (3): 140. 1936.

**Type**: Mexico, IX-1840, *H.G. Galeotti 3203* (Isotype: K478016!). Isolectotype: Mexico, Tioselo, *S.C.*, *s.n.* (P2142694!).

**Distinguishing features**: Unarmed shrub, up to 7 m tall, or low tree, with yellow spread pubescence at branches in young parts, turning glabrous with age. Pinnae 18–32 pairs per leaf. Leaflets hirsutulous, always with a flat margin. Fruit 4.6–6.2 × 1.5 cm, straight to curved, flattened, plus a beak up to 4 mm long, abruptly rounded apically, never gradually beaked.

**Representative examined material**: Coahuila: 26-VI-1936, *L. Wynd 287* (MEXU); *L. Wynd 314* (MEXU).

**Comments**: Recorded only in the state of Coahuila, on mountain slopes, dry shrublands and oak–pine forest, 750–2200 m. This is the only variety whose branches, petiole, and leaf rachis have yellow and erect hispidulous pubescence, and always with 18 or more pairs of pinnae per leaf, the calyx hirsutulous, and the pod apex abruptly rounded. From southern USA through Mexico and Central America to Colombia, Venezuela, and Bolivia.

***Acaciella angustissima* ssp**. ***Texensis*** (Nutt. Ex Torr. and A. Gray) L. Rico. Basionym: *Acacia texensis* Nutt. Ex Torrey and A. Gray, Fl. N. Amer. 1: 404. 1840. *Acaciella texensis* (Nutt. Ex Torrey and A. Gray) Britton and Rose, N. Amer. Fl. 23: 100. 1928. *Acacia filicioides* var. *texensis* (Nutt. Ex Torrey and A. Gray) Small, Bull. New York Bot. Gard. 2: 93 (1901). *Acacia angustissima* var. *Texensis* (Nutt. Ex Torrey and A. Gray) Isely, Sida 3: 372. 1969. *Acacia cuspidata* Schldl., Linnaea 12: 573. 1838. *Acacia angustissima* var. *Cuspidata* (Schldl.) Benson, Amer. J. Bot. 30: 238. 1943. *Acacia hirta* Torrey and A. Gray, Fl. N. Amer. 1: 404. 1840. *Acaciella hirta* (Torr. and A. Gray) Britton and Rose, N. Amer. Fl. 23: 102. 1928. *Acacia angustissima* var. *Hirta* (Torr. and A. Gray) B.L. Robinson, Rhodora 10: 33. 1908. *Acacia angustissima* var. *Chisosiana* Isely, Sida 3: 370. 1969. *Acacia angustissima* var. *Oaxacana* B.L. Turner, Phytologia 81: 14. 1996.

**Type**: USA, Texas, *Drummond 155* (Isotype: K791231!). 

**Distinguishing features**: Unarmed shrub, 3–5 m tall. Young shoots and branches glabrous or glabrescent. Pinnae 6–8 pairs per leaf. Leaflets always with flat margin, glabrous or marginally ciliate. Fruits flattened abruptly rounded apically, never gradually beaked.

**Representative examined material**: Coahuila: 8-VIII-1976, *J. Henrickson 15094* (MEXU). Nuevo León. Tamaulipas: 18-IX-1976, *F. González Medrano*; *A. Castellanos V. P. Zavaleta 9787* (MEXU).

**Comments**: From southern USA to the south of Mexico (Oaxaca). Easy to recognize from the other two varieties, since it has both glabrescent stems and foliage and the smallest number of pairs of pinna per leaf (6). In piedmont scrub, oak forest, and oak–pine forest, 650–1700 m. The leaves and fruits are used as fodder for domestic livestock.

***Acaciella hartwegii*** (Benth.) Britton and Rose, N.L. Britton and al. (eds.), N. Amer. Fl. 23: 102. 1928. Basionym: *Acacia hartwegii* Benth., Pl. Hartw.: 13. 1839. *Acaciella prostrata* Britton and Rose, N. Amer. Fl. 23: 102. 1928. *Acacia guadalajarana* Standl., Publ. Field. Mus. Nat. Hist., Bot. ser. 11: 158. 1936. *Acacia procumbens* (Britton and Rose) Bullock, Kew Bull. 1939: 2 (1939). *Acacia leucothrix* Standl., Contr. U.S. Natl. Herb. 20: 185. 1918.

**Type**: Mexico, Aguascalientes, 1837, *K. T. Hartweg 74* (Isotype: GH00058236!; LD1573107!; P03103404!).

**Distinguishing features**: Commonly as perennial herbaceous, sometimes as shrub, up to 3 m tall. Pinnae 2–8 pairs per leaf. Paraphyllidia 0.75–1 mm long. Leaflets 6–20 pairs per pinna with brochidodromous venation evident abaxially, inconspicuously sub-adaxially, the midvein subcentral. Inflorescences axillar, arranged in capitulate racemes. Flowers 5-merous; white, drying pink or orange. Fruit 4–6 × 0.9–1.2 cm, straight, flattened, chartaceous, reticulate veined, reddish-brown, glabrescent, acute apically.

**Representative examined material**: Tamaulipas: 28-IV-2000, *M. Galván* 824 (MEXU).

**Comments**: Rare in northeastern Mexico, recorded only in Tamaulipas, inhabiting Tamaulipan thorn scrub, 100 m. Outside the area, in Guerrero, Guanajuato, Jalisco, Morelos, Michoacan, Zacatecas, and Edo. Mexico [66].

***Acaciella lemmonii*** (Rose) Britton and Rose, N. Amer. Fl. 23: 103. 1928. Basionym: *Acacia lemmoni* Rose, Contr. U.S. Nat. Herb. 12: 409. 1909. *Acacia angustissima* subsp. *Lemmonii* (Rose) Wiggins, Contr. Dudl. Herb. 3: 230. 1942. *Acaciella shrevei* Britton and Rose, N. Amer. Fl. 23: 105. 1928. *Acacia shrevei* (Britton and Rose) Tidestr., Proc. Biol. Soc. Wash. 48: 40. 1935. *Acacia angustissima* var. *shrevei* (Britton and Rose) Isely, Sida (6): 371. 1969. *Acacia hirta* var. *shrevei* (Britton and Rose) Kearney and Peebles, J. Wash. Acad. Sci. 29: 482. 1939.

**Type**: USA, Huachuca, IX-1882, *J.G. Lemmon s.n.* (Isotype: CAS68!; G364558!).

**Distinguishing features**: Shrub, 2–3 m tall. Pinnae 5–10 pairs per leaf. Leaflets 15–20 pairs per pinna, brochidodromous veined, glabrous, margins short ciliate. Inflorescences axillar, in fascicles of three, arranged in spheric capitula. Flowers white, turning yellow-white when dry. Fruit 4–6 × 1 cm, flattened, straight, chartaceous, reticulate-veined, hirsute, the hairs rounded basally.

**Representative examined material**: Nuevo León: 25-IX-1986, *E. Estrada 683* (MEXU).

**Comments**: Recorded only in the state of Nuevo Léon, piedmont scrub, 550 m, rare. Outside the area, in southern USA (Arizona and Texas) to Chihuahua, Sonora, and Sinaloa. In northeastern Mexico, *A. lemmonii* It is the only species in northeastern Mexico with brochidodromous (the secondary veins attaching to the secondary veins above them) venation.

***Acaciella tequilana*** (S. Watson) Britton and Rose, N.L. Britton and al. (eds.), N. Amer. Fl. 23: 105. 1928. ssp. ***tequilana.*** Basionym: *Acacia penicillata* Standl., Contr. U.S. Natl. Herb. 20: 185. 1919. *Acaciella penicillata* (Standl.) Britton and Rose, N. Amer. Fl. 23: 104. 1828. *Acacia laevis* Standl., Contr. U.S. Natl. Herb. 20: 185. 1919. *Acaciella laevis* (Standl.) Britton and Rose, N. Amer. Fl. 23: 104. 1928.

**Type**: Mexico. Oaxaca, Cerro San Felipe, *C. Conzattii & V. González 564* (MEXU).

**Distinguishing features**: Unarmed, sub-shrub or shrub, up to 2.5 m tall. Young branches and leaves glabrous. Pinnae 2–8 pairs per leaf. Leaflets 6–30 pairs per pinna, 1.2–2.2 cm long. Inflorescences in terminal open panicles, the flowers arranged in spherical capitula. Flowers orange when dry. Fruit 5.3–8.5 × 0.6–1.4 cm, oblong, straight, flat, chartaceous, reticulate-veined, glabrous, acute at both ends.

**Representative examined material**: Tamaulipas: 14-IX-1967, *Rzedowski 45956* (ENCB).

**Comments**: Endemic to Mexico. Recorded only in the state of Tamaulipas at scrublands and oak–pine forest, 900–2400 m. In northeastern Mexico, this is the only species of *Acaciella* whose leaflets are 1.2 cm long or longer. From Chihuahua, Durango, and Tamaulipas to southern Mexico, in Puebla and Oaxaca. The other two subspecies of *A. tequilana* are distributed, one in northwest Mexico (*A. tequilana* ssp. *crinita*), from Sonora and Chiuahua to Durango and Sinaloa, while *A. tequilana* ssp. *pubifoliolata* is recorded only in the states of Colima and Jalisco.

***Mariosousa*** Seigler and Ebinger. Novon 16: 413–420. 2006.

Shrubs or trees, unarmed. Stipules persistent. Leaves bipinnate. Petiole with a solitary small gland. Leaf rachis with a gland between the insertion of the distal pairs of pinnae. Pinnae 1-11 pairs (in northeastern Mexico) per leaf. Leaflets 4 to 40 pairs (in northeastern Mexico), linear to oblong, oblique basally. Inflorescences arranged in cylindrical spikes, involucre of bracts absent. Flowers 5-merous, creamy white. Stamens 50 or more, free. Ovary stipitate. Fruits stipitate, the body flattened, oblong, straight, dehiscent, striate. Seeds flattened, smooth.

Genus consisting of 13 species [67], distributed from southern USA through Mexico to Central America.

The species of *Acaciella* and *Mariosousa* are morphologically very similar, and in the absence of flowers they can be confused; however, they can be easily differentiated by the presence of a gland on the petiole leaf in *Mariosousa*.
1A.Leaflets glabrous or slightly appressed—pubescent abaxially (underneath); minute purple glands absent on rachis and pinnae rachises***M. coulteri***1B.Leaflets glabrous or usually densely sparsely pubescent adaxially and abaxially; minute purple glands present on rachis and pinnae rachises22A.Petiolar gland elevated, with bulbous apex; most leaves with 7 pairs of pinnae or fewer***M. mammifera***2B.Petiolar gland sessile, the apex irregularly elevated; most leaves with 10 or more pairs of pinnae***M. durangensis***

***Mariosousa coulteri*** Seigler and Ebinger, Novon 16 (3): 417. 2006. Basionym: *Acacia coulteri* Benth. in A. Gray, Pl. Wright. 1: 66. 1852. *Senegalia coulteri* (Benth.) Britton and Rose, N. Amer. Fl. 23: 112. 1928.

**Type**: Mexico. Hidalgo. Zimapán. *T. Coulter s.n.* (holotype: K81894!).

**Distinguishing features**: Shrub or tree, up to 8 m tall, unarmed. Petiole with a sessile, circular gland, arising in the upper third, below the proximal pair of pinnae, rarely absent. Leaf rachis with a semicircular gland between the insertion of the distal pair of pinnae. Pinnae 4–11 pairs per leaf. Leaflets 15–35 pairs per pinna; the margin flat, not revolute, and frequently strigulose abaxially. Inflorescences in fascicles, arranged in cylindrical spikes. Fruit 9–17 × 1–2.5 cm, oblong, straight, flattened, flexible, yellowish-brown to dark brown.

**Representative examined material**: Coahuila: 6-VI-1991, *J.A. Villarreal 5958* (ANSM). Nuevo León: 31-V-2003, E. *Estrada 15754* (CFNL); 31-V-2003, *E. Estrada 15755* (CFNL); 5-VIII-1980, *G.B. Hinton 17938* (TEX-LL); 23-VII-2002, *C Yen y Estrada 15020* (CFNL). Tamaulipas: 06-V-1994, *G.B. Hinton 24158* (ANSM); 24 VI-1999, *A. Mora-Olivo 7584* (UAT); 5-V-1992, *J.L. Mora-López 121* (UAT); 24-III-1985, *M. Martínez 215* (UAT); 9-V-1985, *M. Yanez 116* (UAT).

**Comments**: Endemic to Mexico. Also in Sonora and San Luis Potosí, along the states of the Pacific Coast, Puebla, and Morelos. Frequently in piedmont scrub, wet canyons, and oak–pine forest, 485–1650 m. Used as an ornamental species and forage for domestic livestock.

***Mariosousa durangensis*** (Britton and Rose) Seigler and Ebinger, Novon 16 (3). 419. 2006. Basionym: *Senegalia durangensis* Britton and Rose, N. Amer. Fl. 23: 112. 1928. *Acacia durangensis* (Britton and Rose) Jawad, Seigler and Ebinger, Ann. Missouri Bot. Gard. 87: 541. 2000.

**Type**: Mexico. Durango. San Ramón, 21 IV–18 V 1906, *E. J. Palmer 107* (NY3311!).

**Distinguishing features**: Similar morphological characters to previous species; however, this species has membranous or papery leaflets with revolute and glabrous margin.

**Representative examined material**: Tamaulipas: x-1982, *M.H. Cervera 119* (MEXU); 21-VII-1953, *W.E. Manning 53403*, *M.S. Manning* (MEXU); 27-IX-1959, *M.C. Johnston*, *J. Graham 4096* (MEXU); 15-IX-1960, *J. Crutchfield*, *M.C. Johnston 5514* (MEXU); 26-V-1970, *F. González Medrano*, *V.M. Toledo*, *E. Martínez 2994* (MEXU).

**Comments**: Endemic to Mexico. Recorded only in Tamaulipas, piedmont scrubland and transitions to oak–pine forest, 550–1250 m. Also in Durango.

***Mariosousa mammifera*** (Schltdl.) Seigler and Ebinger, Novon 16 (3). 419. 2006. Basionym: *Acacia mammifera* Schltdl., Linnaea 12: 563. 1838. *Senegalia mammifera* (Schltdl.) Britton and Rose, N. Amer. Fl. 23: 112. 1928.

**Type**: Mexico. Hidalgo: Barranca de Acholoya, n.d., *C. A. Ehrenberg s.n.* (Holotype, NY1481!).

**Distinguishing features**: Shrub, 4–5 m tall. Pinnae 3–9 pairs per leaf, leaflets 10–30 pairs per pinna. Petiole with a gland between the insertion of the proximal pair of pinnae, sometimes in the middle portion. Inflorescences axillary, solitary or in fascicles, arranged cylindrical spikes. Peduncles with minute purple glands. Fruit 8–24 × 1.8–3.3 cm, oblong, flattened, straight, glabrous, dehiscent, yellowish-brown or dark green.

**Representative examined material**: Nuevo León: 3-VI-1985, *B. Treviño 422* (CFNL). Nuevo León: 2-X-1985, *E. Estrada 707* (CFNL). 15-VII-1933, *C.H. Muller 532* (TEX-LL). Tamaulipas: 9-V-1986, *A. García Mendoza 2255* (MEXU).

**Comments**: Endemic to Mexico, recorded in Nuevo León and Tamaulipas, in piedmont scrub and transitions to oak forest, 850–1300 m. Also, in San Luis Potosí to Hidalgo and along the Pacific Coast to Oaxaca.

***Senegalia*** Raf., Sylva Tellur. 119. 1838.

**Type**: *Senegalia triacantha* Raf., based on *Mimosa 15enegal* L., Sp. Pl. 1: 521. 1753.

Shrubs or trees unarmed or armed with prickles. Stipular spines absent. Prickles commonly scattered along stems and branches, rarely 2–3 grouped together near the nodes. Leaves bipinnate, the petiole and the proximal pair of pinnae with a sessile or stipitate glands. Ovary stipitate. Inflorescences arranged in spikes or capitula. Fruit dehiscent into 2 valves, indehiscent or separating into indehiscent 1-seeded articles. This genus consists of 217 species worldwide [68], most diversity on the American continent (97 species), Africa (62) and Asia [63,69,70,71,72]. Within Asia, China is the country with the highest number of *Senegalia* species [69].
1A.Leaflets 10 mm wide or wider21B.Leaflets less than 5 mm wide32A.Leaflets 1 pair per pinna, 2–5 cm wide ***S. crassifolia***2B.Leaflets 4 or more pairs per pinna, 9 mm wide or less***S. anisophylla***3A.Inflorescences arranged in spikes, 2–6 times longer than wide43B.Inflorescences arranged in spheric capitula or ovoid capitula, less than twice as long as wide54A.Leaflets 5–9 (-12) mm long, petiole (4-) 6–11 mm long; inflorescence lax; fruit not curling when mature; seeds suborbicular, 7–10 mm long***S. wrightii***4B.Leaflets 3–6 mm long, petiole 2–5 mm long; inflorescence dense; fruit curling when mature; seeds orbicular, 5–7 mm long***S. greggii***5A.Leaflets 15–35 pairs per pinna; fruit coriaceous, velvety pubescent***S. berlandieri***5B.Leaflets 6–20 pairs per pinna66A.Leaflets 15–20 pairs per pinna; pinnae 2–7 pairs per leaf***S.* × *emoryana***6B.Leaflets 6–13 pairs per pinna; pinnae 2–4 pairs per leaf**7**7A.Leaflets cuneate-oblong, glaucous, with obvious veins, not strongly reticulated abaxially (down surface) ***S. roemeriana***7B.Leaflets widely oblong, light green, strongly reticulated abaxially ***S. micrantha***

***Senegalia anisophylla*** (S. Watson) Britton R: Rose, N. Amer. Flora 23: 109. 1928; Basionym: *Acacia anisophylla* S. Watson, Proc. Amer. Acad. Arts 21: 452. 1886.

**Type**: Mexico, Coahuila, mountains, canyons near Jimulco, 14-V-1885, *C. G. Pringle 163!* Isotypes: (NY1451NY!).

**Distinguishing features**: Shrub, up to 3 m tall, armed with thorns scattered in the internodes. Petiole with a gland located between the insertion of the proximal pair of pinnae or immediately below, rachis with a gland at the insertion of the distal pair of pinnae. Pinnae 5–7 pairs per leaf. Leaflets 6–10 pairs per pinna, 2–5 cm wide. Inflorescences solitary, axillar, arranged in spheric capitula. Peduncle with a lanceolate bract in the middle. Fruit 11–13 × 2–2.5 cm, oblong, straight, 13–15 × 8–9 mm, elliptic, flattened.

**Representative examined material**: Coahuila: 5-IX-1980, *A. Rodríguez 168* (ANSM); 13-IX-1971, *J. Henrickson 6730* (MEXU); 9-VIII-1994, *J.A. Villarreal 7771*, *M.A. Carranza* (MEXU); 15-VI-1972, *F. Chiang*; *T. L. Wendt*; *M. C. Johnston 7811* (MEXU); 24-VII-2013, *J. Estrada 116* (CFNL); 27-IV-2007, *E. Estrada 20028* (CFNL). Tamaulipas: 14-I-1977, *F. González-Medrano 10338*, *A. Castellanos*, *V. Álvarez* (MEXU); 23-VIII-1984, *F. González Medrano*; *Verónica Juárez Jaimes 279* (MEXU); 10-XII-1982, *F. Iribe 165* (MEXU).

**Comments**: Endemic to Mexico. Recorded in Coahuila and Tamaulipas, on stony slopes with piedmont scrub and desert scrublands, also in central Mexico (Durango, San Luis Potosí, Querétaro y Zacatecas), 650–1400 m. This species has been mentioned to be a hybrid between *Senegalia* (*Acacia*) *berlandieri* and *S.* (*Acacia*) *crassifolia* [72]. In the area where this species was recorded (Jimulco, Coahuila), both parental species are also present, and undoubtedly the individuals of *S. anisophylla* show their morphological characteristics.

***Senegalia berlandieri*** (Benth.) Britton and Rose, N. Amer. FI. 23: 109. 1928. Basionym: *Acacia berlandieri* Benth., London J. Bot. I: 522. 1842.

**Type**: Monterrey. I-1828, *M. Berlandier 1392*! (Isotype: G364603!, P3103095!)

**Distinguishing features**: Shrub, 1–4 m tall. Young branches with thorns scattered in the internodes or unarmed. Pinnae 6–11 pairs per leaf, leaflets 25–35 pairs per pinna. Peduncles in fascicles, with a circular gland in the middle. Inflorescences arranged in spheric capitula. Flowers light yellow. Fruit 7–13 × 1–2.5 cm, flattened, dehiscent, coriaceous, velvety pubescent, subglabrous or glabrate at maturity, persistent for long time.

**Representative examined material**: Coahuila: 30-IV-2015 *J.A. Encina 4593* (ANSM). Nuevo León: 13-VIII-1988, *T.F. Patterson 6603* (TEX-LL); 13-IV-2003, *E. Estrada 15528* (CFNL); 26-VIII-1936, *M. Taylor 128a* (TEX-LL); 2-III-2003, *E. Estrada 15231* (CFNL); 9-IX-2002, *E. Estrada 15198* (CFNL). Tamaulipas: 26-XI-1990, *J.S. Sifuentes 62* (UAT); 28-VI-1985, *M. Martínez 393* (UAT); *26-XI-1984*, *C.G. Romo* 282c (UAT); 27-XI-1986, *L. Hernández 1665* (UAT); 4-VI-1985, *J. Jiménez 167* (UAT).

**Comments**: Occurring in the Northern Gulf Coastal Plain and North American Plain physiographic provinces, species common in the Tamaulipan thorn scrub, piedmont scrub, rosetophyllous scrubland, oak forest, oak–pine forest, chaparral, and semi-arid scrubland in the High Plains of northeastern Mexico, 350–1990 m. Distributed from the south of the USA through Mexico to Veracruz. Widely used as a forage species for cattle and goats.

***Senegalia crassifolia*** (A. Gray) Britton and Rose, N. Amer. FI. 23: 108. 1928. Basionym: *Acacia crassifolia* A. Gray, Mem. Amer. Acad. Arts n.s. 5: 3 17. 1854.

**Type**: Coahuila, Mexico, in the mountain pass of La Peña, IX-1852, *G. Thurber 829* (Isotype: NY1464!).

**Distinguishing features**: Shrub, 1–3.5 m tall. Branches with thorns scattered in the internodes. Pinnae 2 pairs per leaf. Leaflets 1 pair per pinna, 2–5 × 2–5 cm, rounded, flabellate, suborbicular, gray-yellow-green, the veins yellow on both surfaces, thick and hard, strongly reticulated in both sides. Inflorescences arranged in spherical capitula. Fruit 4–11 × 1–2.5 cm, oblong, slightly curved, hard, with thickened marginal, somewhat bulging over the seeds, dehiscent.

**Representative examined material**: Coahuila: 6-XII-1993, *M.A. Carranza 2006* (ANSM); 7-IX-2007, *J.A. Ávila 237* (CFNL); 21-VI-2007, *E. Estrada 20098* (CFNL); 11-X-2008, *J.A. Ávila 333* (CFNL).

**Comments**: Endemic to Mexico. In rosetophyllous scrublands, stony soils, on low hills and valleys, recorded only in the state of Coahuila. This species hybridizes with *Acacia berlandieri* where the populations of both overlaps; this hybrid was published as *A. anisophylla* W. Watson [73]. Outside the study area, it has been recorded in Durango, San Luis Potosí, and Zacatecas.

***Senegalia*** x ***emoryana*** (Benth.) Britton and Rose, N. Amer. Fl. 23 (2): 109. 1928. Basionym: *Acacia emoryana* Benth. Trans. Linn. Soc. London 30: 522. 1875.

**Type**: USA, Texas, chiefly in the Valley of the Rio Grande, below Doñana, *C.C. Parry*, *J.M*, *Bigelow*, *C. Wright*, *A. Schott 325* (Holotype: K117591! Isotype: NY1436! Lectotype: P2142708!).

**Distinguishing features**: Shrub or small tree, up to 5 m tall. Prickles recurved, scattered along branches or absent. Petiolar gland orbicular to elliptic. Pinnae 2–7 pairs per leaf. Leaflets 15–20 pairs per pinna. Inflorescences axillar, 2–4, arranged globose to subglobose, less than twice as long as capitula width. Fruit 4–15 × 2–3.5 cm, flattened, coriaceous, subglabrous, smooth margin or sometimes constricted between seeds.

**Representative examined material**: Coahuila, 9-VI-1972, *F. Chiang*, *T.L. Wendt*, *M.C. Johnston 7573* (MEXU).

**Comments**: Recorded only in Coahuila, thorn desert scrub, 1250 m. Outside this area, in Chihuahua, Durango, and Sonora. This species is a fertile hybrid of *Senegalia greggii* and *S. berlandieri* [74,75].

***Senegalia greggii*** (A. Gray) Britton and Rose, N. Amer. FI. 23: 11 0. 1928. Basionym: *Acacia greggii* A. Gray, PI. Wright. 1: 65. 1852.

**Type**: Chihuahua. Mexico, west of Patos (dry valley), 10-IV-1847, *Dr. Gregg s.n.* (Holotype: NY1474!).

**Distinguishing features**: Shrub, 2–6 m tall, armed with internodal curved thorns. Petioles with a capitate gland inserted between the insertion of the proximal pair of pinnae. Pinnae 1–4 pairs per leaf. Leaflets 3–6 pairs per pinna. Inflorescences in fascicles, arranged in dense cylindrical spikes. Fruit 5–10 × 1–3 cm, oblong, flattened, straight or coiled, glabrous, occasionally constricted between some seeds, light brown or glaucous, dehiscent, flexible, or late hardened (sub-coriaceous).

**Representative examined material**: Coahuila: 24-VIII-2018, *F. Meráz 193* (ANSM); 22-IV-1977, *R. Grether 637* (MEXU). Nuevo León: 5-VI-2001, *E. Estrada 12892* (CFNL); 16-IV-2001, *E. Estrada 12433* (CFNL); 19-VII-2000, *E. Estrada 14924* (CFNL); 13-VII-2002, *E. Estrada 14798* (CFNL). Tamaulipas: 12-V-1993, *F. González Medrano*; *Francisco*; *G. G. Hernández M.*; *J. G. R. Wong 17797* (MEXU); IV-1969, *Villegas 23* (MEXU).

**Comments**: Occurring in the south of the USA and the north of Mexico. Species typical of the Tamaulipan thorn scrub in the Northern Gulf Coastal Plain, sometimes in piedmont scrub, on the lower part of mountain slopes; frequently associated with mezquitales (*Neltuma* spp.) and sometimes in halophytic vegetation, 150–640 m. Its wood is used in regional industry to manufacture handmade toys.

***Senegalia micrantha*** Britton and Rose, N. Amer. F1. 23: 1 15. 1928. Basionym: *Acacia micrantha* Benth., Trans. Linn. Soc. London 30: 526. 1875. *Acacia parviflora* E.L. Little, Phytologia 6: 506. 1959.

**Type**: Mexico, Tamaulipas: between Las Apuntas and Las Verdosas, Herbarium Berlandierurn Texano-Mexicanurn, *M. Berlandier 3148* (Isolectotype: 140NY!).

**Distinguishing features**: Shrub, 2–5 m tall. Branches armed with internodal curved thorns. Petiole with a triangular gland inserted between the proximal pair of pinnae. Pinnae 3–4 pairs per leaf. Leaflets 5–12 pairs per pinna, light green, strongly reticulate abaxially. Inflorescences arranged in spherical capitula. Peduncles frequently with a small bract above its mid portion and another adjacent to the capitula. Fruit 5–11 × 1.8–2.4 cm, oblong, flattened, flexible, glabrous, ending in a long peak, tardily dehiscent.

**Representative examined material**: Coahuila: 26-IX-2002, *M.A. Carranza*, *I. Ramírez s.n.* (MEXU). Nuevo León: 9-IV-2001, *E. Estrada 12074* (CFNL). 24-VII-1992, *G. Hinton* et al. *22170* (TEX-LL); 11-V-1989, *E. Estrada 1466* (TEX-LL). *E. Estrada 14574* (CFNL). 23-VII-2002, *E. Estrada 15100* (CFNL). Tamaulipas: 25-VI-1985, *M. Martínez 751* (ANSM); 26-VI-1985, *M. Martínez 279* (UAT); 27-VIII-1983, *L. Hernández 646* (UAT); 24-IX-1984, *R. Diaz s.n.* (UAT); 7-XII-1976, *F. González-Medrano 10121* (UAT); 1-VIII-1994, *D. Seigler 14103* (UAT); 6-VIII-1994, *D. Seigler 14239* (UAT).

**Comments**: Endemic to Mexico. In Tamaulipan thorn scrub, piedmont scrub on mountain slopes and arid scrubland in High Plains, stony soils, sparsely vegetated areas, occasionally in oak forest and oak–pine forest 250–1100 m. Very similar to *V. roemeriana*, differing in leaf venation and shape and color of leaflets. Distributed also in San Luis Potosí, Querétaro, Hidalgo, and Guanajuato.

***Senegalia roemeriana*** (Scheele) Britton and Rose, N. Amer. FI. 23: 1 15. 1928. Basionym: *Acacia roemeriana* Scheele, Linnaea 21: 456. 1848.

**Type**: USA. Texas, near Austin. IV-1847, *F. Romer s.n.* (lectotype: K81911!).

**Distinguishing features**: Shrub or small tree, 2–5 m tall. Branches armed with reddish (when young), curved, sub-nodal, and internodal thorns. Pinnae 2–4 pairs per leaf. Leaflets 6–13 pairs per pinna, glabrous, veins evident but not prominent, reticulate, glaucous. Inflorescences arranged in spherical capitula. Fruit 6–10 × 1.5 cm, oblong, slightly curved, light to dark brown.

**Representative examined material**: Coahuila: 18-IV-2017, *J.A. Encina 5734* (ANSM). Nuevo León: 26-VII-2000, *C. Yen y E. Estrada 11676* (CFNL); 9-VI-2001, *J. Medellín n/n* (CFNL); 6-VIII-1986, *E. Estrada 607* (CFNL); 24-IV-1960, *J. Crutchfield, M.C. Johnston 5296*! (TEX-LL); 6-VII-2001, *E. Estrada 12345*! (CFNL). Tamaulipas: 6-XII-1984, *O Briones 1459* (ANSM).

**Comments**: Frequent in the Tamaulipan thorn scrub, piedmont scrub, and oak–pine forests, more common in shallow rocky soils, 350–1200 m. Also found in Texas (USA) and the north of Mexico.

***Senegalia wrightii*** (Benth.) Britton and Rose, N. Amer. Fl. 23. 110.1028. Basionym: *Acacia wrightii* Benth. in A. Gray, Pl. wright. 1: 64. 1852. *Acacia greggii* A. Gray. var. *wrightii* (Benth.) Isely, Sida 3: 378. 1969.

**Type**: USA. Texas, expedition form western Texas to El Paso, New Mexico, hills of the Rio Grande and east to San Antonio, V to IX-1849, *C. Wright 173* (Isotype NY1449!).

**Distinguishing features**: Shrub or small tree, 2–5 m tall. Branches armed with curved internodal thorns. Pinnae 1–4 pairs per leaf. Leaflets 3–6 pairs per pinna, 3–6 × 2–3 mm, obovate. Petiole with a capitate gland inserted in the proximal pair of pinnae. Inflorescences arranged in cylindrical spikes. Fruit 5–10 × 1–3 cm, oblong, flattened, straight or coiled, glabrate, sometimes constricted between seeds, brown or glaucous, dehiscent, flexible, elastic or tardily subcoriaceous.

**Representative examined material**: Coahuila: 18-IV-2017, *J.A. Encina 5719* (ANSM); 28-VII-1982, *C. Diggs*, *M. Nee 3135* (MEXU). Nuevo León: 5-VI-2001, *E. Estrada 12892!* (CFNL); 19-VII-2000, *E. Estrada 14924*! (CFNL); 23-VI-2001, *E. Estrada 12783a*! (CFNL). Tamaulipas: 28-IV-1969, *F. González Medrano 2289*, *J. Sánchez* (MEXU); 3-V-1960, *J. Crutchfield*, *M.C. Johnston 5433* (MEXU); 25-IV-1960, *J. Crutchfield*, *M.C. Johnston 5328* (MEXU); 15-IV-1999, *A. Mora-Olivo 7522* (UAT).

**Comments**: Species characteristic in lowlands of northeastern Mexico, in Tamaulipan thorn scrub, frequent in sandy soils, associated with mesquite (*Neltuma* spp.), 250–700 m. Southeastern USA to the north of Mexico. Its wood is used for the manufacture of handmade toys.

***Vachellia*** Wight and Am., Prodr. Fl. Ind. Orient. 1: 272. 1834.

**Type**: *Vachellia farnesiana* (L.) Wight and Arn., based on *Mimosa farnesiana* L., Sp. Pl. 521: 1753. *Acacia farnesiana* (L.) Willd., Sp. P1. 4: 1083. 1806.

Trees or shrubs. Stipules spinescent, paired. Prickles or thorns absent. Leaves bipinnate. Inflorescences arranged in spheric capitula. Peduncles with a whorl of bracts. Ovary sessile to subsessile. Fruits indehiscent or dehiscent and sometimes with multiserial seeds.

Genus composed of approximately 165 species [69]. Sixty-two of them are distributed in America [73]. In northeastern Mexico, 10 species have been recorded. The species grow in different ecosystems, mostly in arid, semiarid, or xeric scrublands, also common in tropical forest, cloud forest, and oak–pine forest. Some species are associated with secondary vegetation caused by overgrazing, seasonal agriculture, and immoderate felling of primary vegetation.
1A.Stipules 5 mm diameter or wider at base, swollen, myrmecophiles (inhabited by ants) or flattened and sword- or boat-shaped21B.Stipules less than 5 mm in diameter, never swollen at the base32AInflorescences arranged in cylindrical spikes; secondary venation of leaflets usually visible***V. cornigera***2B.Inflorescences arranged in spheric capitula; secondary venation of leaflets usually not easily visible***V. sphaerocephala***3A.Pinnae 12 or more pairs per leaf***V. pennatula***3B.Pinnae 9 or fewer pairs per leaf44A.Inflorescences arranged in cylindrical spikes54B.Inflorescences arranged in spheric capitula**6**5A.Leaflets 2–3 pairs per pinna, large, all or most of them 1 cm wide or wider***V. californica*** ssp. ***pringlei***5B.Leaflets 3–5 pairs per pinna, smaller, the largest 8 mm wide or narrower***V. rigidula***6A.Fruits indehiscent, septate internally; whorl of bracts at the apex of the peduncle immediately below the capitulum76B.Fruits dehiscent, non-internally septate; whorl of bracts near mid-peduncle97A.Petiolar gland elliptic; below, in the middle of, or at the apex of the petiole; fruit 3–6.5 cm long, inflated, black or dark brown, glabrous, without minute reddish glands***V. farnesiana*** ssp. ***farnesiana***7B.Petiolar gland circular, located immediately below the first pair of pinnae; fruits 6.5–15 cm long, pubescent, rarely glabrous, with minute reddish deciduous glands88A.Secondary shoots (brachyblasts) thick, 3–4 mm diameter; pinnae 3–5 pairs per leaf; fruit compressed, 9 mm wide or wider, slightly constricted between the seeds***V. schaffneri***8B.Secondary shoots (brachyblasts) thin, 2–3 mm diameter; pinnae 2–3 pairs per leaf; fruit subcylindrical, 8 mm wide or less, usually constricted between the seeds***V. bravoensis***9A.Petiole wide canaliculate, 3.5 mm long or shorter; fruit non-striated, with numerous reddish, pedicellate, 1 × 0.2 mm glands***V. glandulifera***9B.Petiole narrowly canaliculate, usually 4–11 mm long; fruits striated, glabrous or with tiny reddish-brown glands, not pedicelled, less 1 × 0.2 mm1010A.Leaf rachis 15 mm long or longer; pinnae 3–6 (-7) pairs per leaf; leaflets not glutinous; fruits non-glutinous, with tiny reddish-brown glands***V. constricta***10B.Leaf rachis 5 mm long or shorter; pinnae 1–2 (-3) pairs per leaf; leaflets glutinous; fruits glutinous, glabrous***V. vernicosa***

***Vachellia bravoensis*** (Isely) Seigler and Ebinger. Phytologia 87 (3). 146. 2005. Basionym: *Acacia schaffneri* (S. Watson) F. J. Herm. var. *bravoensis* Isely, Sida 3: 383. 1969.

**Type**: USA, Texas, San Patricio Co., 7 miles S of Taft in clay loam soil, 29-III-1950, *F. B. Jones 100* (BRIT2629!).

**Distinguishing features**: Shrub, 1–3 m tall. Stipules spinescent, paired. Pinnae 2–3 pairs per leaf, leaflets 12–21 pair per pinna. Petiolar gland sessile, arising between the insertion of the proximal pair of pinnae or immediately below. Inflorescences arising from thin brachyblasts, 2–3 mm diameter, arranged in spheric capitula. Peduncles with a small involucre of bracts near the apex of the peduncle, with minute red glands. Fruit 5–12.5 × 0.6–0.8 mm, linear, commonly constricted between seeds, brown to dark brown, puberulent, covered with abundant reddish deciduous glands.

**Representative examined material**: Nuevo León: 7-IV-2001, *E. Estrada 11877* (CFNL); 12-III-1960, *J. Crutchfield* and *M.C. Johnston 5247* (TEX-LL). Tamaulipas: 28-VI-2002, *E. Estrada 14758* (CFNL). Tamaulipas: 2-VII-1986, *J. Torres 179* (UAT).

**Comments**: This species is morphologically like *V. schaffneri*; it can be differentiated from this by the number of pinnae pairs per leaf (2–3), thinner brachyblasts, and cylindrical fruit. It can also be differentiated from *V. farnesiana* via its black color, glabrousness, smoothness, and lack of glands in the fruits. The three species share habitats, at least in Nuevo León and Tamaulipas, inhabiting Tamaulipan thorn scrub. Distributed in the southeastern USA and the north of Mexico.

***Vachellia californica*** Brandegee ssp. ***pringlei*** (Rose) L. Rico, Checkl. Syn. Amer. Sp. Acacia: 57. 2007. Basionym: *Acacia pringlei* Rose, Contr. U.S. Natl. Herb. 3: 316. 1895. *Acaciopsis pringlei* (Rose) Britton and Rose, N.L. Britton and al. (eds.), N. Amer. Fl. 23: 95. 1928. *Vachellia pringlei* (Rose) Siegler and Ebinger, Phytologia 87: 165. 2006. *Acacia conzattii* Standl., Contr. U.S. Natl. Herb. 20: 186. 1919. *Acaciopsis conzattii* (Standl.) Britton and Rose, N.L. Britton and al. (eds.), N. Amer. Fl. 23: 95. 1928. *Acacia unijuga* Rose, Contr. U.S. Natl. Herb. 8: 32. 1901. *Acaciopsis unijuga* (Rose) Britton and Rose, N.L. Britton and al. (eds.), N. Amer. Fl. 23: 95. 1928. *Acaciopsis sesquijuga* Britton and Rose, N.L. Britton and al. (eds.), N. Amer. Fl. 23: 95. 1928. *Acacia sesquijuga* (Britton and Rose) Standl., Publ. Field Mus. Nat. Hist., Bot. Ser. 4: 309. 1929.

**Type**: Mexico, Oaxaca: Tomellin Canyon, alt. 3000 ft. a tree 20–30 ft. high, 22-XII-1894, *C. G. Pringle 6113* (Isotype: M0218454!; UC81083!; MEXU00096049!; HBG520716!; K000081886!; MO-120566!).

Shrub or tree, up to 10 m tall. Pinnae 1 pair per leaf. Leaflets 2–3 pairs per pinna, distally accrescent, oblong–elliptic to obovate. Foliar gland circular, sessile, arising between the insertion of the pinnae. Inflorescences axillar, solitary or in fascicles, arranged in cylindrical spikes. Fruit 6–20 × 0.3–0.5 cm, straight or curved, constricted between the seeds, dehiscent.

**Representative examined material**: Tamaulipas: 11-III-1960, *J.R. Crutchfield*, *M. C. Johnston 5232* (TEX-LL); 4-III-1999, *A. Mora-Olivo 7422* (UAT).

**Comments**: Recorded only in the state of Tamaulipas, in Tamaulipan thorn scrub, at Llera de Canales and Altamira Counties, 100–350 m. In Mexico and Central America.

***Vachellia constricta*** (Benth.) Seigler and Ebinger, Phytologia 87 (3). 152. 2005. Basionym: *Acacia constricta* Benth. in Gray, PI. Wright. 1: 66. 1852. *Acaciopsis constricta* (Benth.) Britton and Rose, N. Amer. Fl. 23: 96. 1928.

**Type**: Mexico, Chihuahua, V to X-1849, *C. Wright 162* (Isotypes: GH58201!)

**Distinguishing features**: Frequently as a small shrub, 1–2.5 m tall. Bark reddish to reddish-brown. Stipules spinescent, paired, white. Pinnae 3–7 pairs per leaf. Leaflets 6–14 pairs per pinna. Petiole with a gland arising between the proximal pair of pinnae, and sometimes in the other pairs of pinnae. Fruit 8–12 × 0.2–0.3 cm, linear, straight to little curved, compressed, constricted between the seeds, brown, conspicuously reticulated, dehiscent, striated, glabrous, with tiny reddish-brown glands.

**Representative examined material**: Coahuila: 8-IX-1990, *M.A. Carranza 745* (ANSM). Nuevo León: 31-V-2003, *E. Estrada 15672* (CFNL); *G. Hinton* et al. *22604* (TEX-LL)**;** *24-VI-2001*, *E. Estrada 12785a* (CFNL). Tamaulipas: 25-I-1983, *L. Hernández 1390* (UAT); 24-VII-1985, *D. Méndez 62* (UAT); 1-VII-1994, *D. Seigler 14101* (UAT); 14-IV-1984, *R. Diaz 104* (UAT); 11-V-1985, *J. Jiménez 131* (UAT).

**Comments**: This species inhabits scrublands in the low and high plains of Nuevo León, 320–1700 m. In Coahuila, it is mainly associated with desert shrublands, and in Tamaulipas, it is frequently associated with, although not very abundant in, Tamaulipan thorn scrub. Distributed from southern USA to central Mexico. 

***Vachellia cornigera*** (L.) Seigler and Ebinger, Phytologia 87 (3): 153. 2005. Basionym: *Mimosa cornigera* L., Sp. Pl. 520: 1753. *Acacia cornigera* (L.) Willd., Sp. PL 4: 1080. 1806. *Tauroceras cornigerum* (L.) Britton and Rose, N. Amer. Fl. 23: 86. 1928.

**Type**: Cuba, 21-IV-1863, *C. Wright 2402* (MO-954183! Isotype: K478118!).

**Distinguishing features**: Shrub or small tree, up to 10 m tall. Stipules spinescent, paired, 3–10 × 0.3–1cm (basally), swollen, myrmecophila, usually round in cross section. Gland (s) cymbiform, 1–2, at the middle or top of the petiole, canoe (boat)-shaped. Pinnae 4–14 pairs per leaf. Leaflet 5–40 pairs per pinna, glabrate with 2–3 lateral veins evident, conspicuous on both surfaces. Peduncles with an involucre of 5 bracts at the base of the peduncle. Inflorescences arranged in cylindrical spikes. Flowers subtended by peltate bracts. Pods 5–10 × 1–2 cm, cylindrical, chartaceous, reddish or maroon, indehiscent, long-beaked, glabrous to minutely puberulent.

**Representative examined material**: Tamaulipas 24-VI-1996, *C. Ramos 109* (MEXU); XII-1964, *F. González Medrano 793* (MEXU); 24-VI-1983, *R. Torres Colín*, *H. Hernández Macías 3093* (MEXU); 21-XII-1995, *G. G. Hernández Mejía 336* (MEXU); 23-III-1984, *S. Rodríguez*, *L. Hernández. G. González 3* (MEXU).

**Comments**: In northeastern Mexico, only two *Vachellia* species have paired spinescent stipules thickened at the base, *V. cornigera* and *V. sphaerocephala*, which can be easily discerned based on their inflorescences. *V. cornigera* has cylindrical spikes, while *V. sphaerocephala* has spherical capitula. *V. cornigera* can also be distinguished from the other species by its swollen stipules, since it is the only one that has peltate floral bracts. In the absence of flowers or fruits, *V. cornigera* can be differentiated from V. *sphaerocephala* by the evident secondary venation present in the first one. *V. cornigera* is frequently found in secondary vegetation, in wet or dry open sunny areas. South of the Tropic of Cancer, along the Gulf and Pacific Coasts to Central America and the Antilles.

***Vachellia farnesiana*** ssp. ***Farnesiana*.** Basionym: *Acacia pedunculata* Willd., Sp. Pl. 4: 1084. 1806. *Mimosa pedunculata* (Willd.) Poir., in Lam. Encycl. Suppl. 1: 81. 1810. *Acacia farnesiana* (L.) Willd. F. *pedunculata* (Willd.) Kuntze, Revis. Gen. pl. 3 (2): 47. 1898.

**Type**: USA, Bayou La Fourche, near cut-off, 16-IV-1931, *J.K. Small*, *E.J. Alexander s.n.* (Isotype: 0000133WIS).

**Distinguishing features**: Shrub or tree, up to 8 m tall. Stipules spinescent, paired. Petiolar gland elliptic, arising in the middle or at the apex of the petiole or immediately below the insertion of the proximal pair of pinnae. Pinnae 2–6 pairs per leaf. Leaflets 10–20 pairs per pinna. Peduncle with a whorl of bracts at the apex, immediately below the capitulum. Inflorescences arranged in spheric capitula. Fruit indehiscent, straight or slightly curved, 3–6.5 cm long, septate internally, inflated, black or dark brown, glabrous.

**Representative examined material**: Coahuila: Coahuila: 25-VI-2015, *M.A. Carranza 4074* (ANSM), 7-VI, 1991, *J.A. Villarreal 6039* (ANSM). Nuevo León: 8-III-2003, *E. Estrada 15292!* (CFNL); 7-VII-2001, *E. Estrada 13017* (CFNL); 2-VII-2000, *E. Estrada 11634* (CFNL); 22-IV-1960, *R.F. Smith M131* (TEX-LL); 5-VII-1969, *J.* and *H. Meras* 3226 (TEX-LL). Tamaulipas: 13-II-1985, *M. Yanez 243* (UAT); 21-V-1985, *M. Martínez 736* (UAT); 8-V-1984, *D. Baro 200* et al. (UAT); 24-IV-1985, *M. Martínez 207* (UAT).

**Comments**: *A. farnesiana* ssp. *Farnesiana* is the species with the largest distribution of the genus, from southern USA through Mexico, Central America and Antilles to Argentina; naturalized in many countries in the Old World. Frequent in disturbed sites, associated with secondary vegetation. Found in all states of Mexico, in almost all the dominant plant associations in Mexico, 450–2300 m. Widely used as a source of coal, wood for construction, and fodder for livestock [62]. *Vachellia farnesiana* is a typical component of areas with anthropogenic disturbance, overgrazing, and excessive extraction of vegetation. Its hard, resistant wood is used in the region for fence posts, house columns, roofs, log making, firewood, furniture, shelving, hand tools, and flooring [62,76].

***Vachellia glandulifera*** (S. Watson) Seigler and Ebinger. Phytologia 87 (3). 158. 2005. Basionym: *Acacia glandulifera* S. Watson, Proc. Amer. Acad. Arts 25: 147. 1890. *Poponax glandulifera* (S. Watson) Britton and Rose, N. Amer. Fl. 23: 88. 1928.

**Type**: Mexico, Coahuila, Carneros Pass., 10-V-1891, *C.G. Pringle 3697* (Holotype: GH58233!)

**Distinguishing features**: Shrub, 0.4–2 m tall; stipules spinescent, paired, straight or slightly curved. Petiole wide canaliculate, 3.5 mm long or shorter, with a sessile gland just below the insertion of the proximal pair of pinnae. Pinnae 1–2 pairs per leaf, usually with a gland between the distal pair of pinnae. Leaflets 5–7 pairs per pinna. Inflorescences arranged in spherical capitula. Peduncles with a whorl of bracts at the middle. Fruit straight or curved, 7.5–9 × 0.4–0.6 cm, compressed, pubescent or rarely glabrous, dehiscent, dark red, with abundant stipitate red to dark red pedicellate glands.

**Representative examined material**: Coahuila: 24-VI-2014, *J.A. Villarreal 9536* (ANSM), 21-VI-2014, *J.A. Encina 4861* (ANSM)**.** Nuevo León: 3-V-1992, *G. Hinton* et al. *21890* (TEX-LL); 15-V-2003, *E. Estrada 15616* (CFNL); 31-V-2003, *E. Estrada 15670* (CFNL); 21-VI-2003, *E. Estrada 15767* (CFNL).

**Comments**: Endemic to the north of Mexico. Also in Chihuahua, Durango, San Luis Potosí, and Zacatecas. Frequent in the High Plains of northeastern Mexico, associated with desert scrublands, 1500–2000 m.

***Vachellia pennatula*** (Schltdl. and Cham.) Seigler and Ebinger Phytologia 87 (3). 164. 2005. ssp. ***pennatula***. Basionym: *Acacia pennatula* subsp. *parvicephala* Seigler and Ebinger, Syst. Bot. 13: 12. 1988. *Inga pennatula* Schltdl. and Cham., Linnaea 5 (4): 593. 1830. *Acacia lanata* M. Martens and Galeotti, Bull. Acad. Roy. Sci. Bruxelles 10:315. 1843.

**Type**: Mexico, Oaxaca. Mixteca Alta, IV-1840, *H. Galeotti 3231* (holotype: BR5187782!).

**Distinguishing features**: Tree, up to 9 m tall. Stipules spinescent, paired. Petiole with a sessile gland, 1–2 mm diameter. Pinnae 17–48 pairs per leaf. Leaflets 30–50 pairs per pinna, commonly shorter than 2 mm. Inflorescences axillar, solitary or in fascicles, arranged in spherical capitula. Fruit 5–13 × 1.2–2.7 cm, straight to slightly curved, brown to dark brown, indehiscent, glabrous.

**Representative examined material**: Tamaulipas: 30-V-1967, *H. Puig 2441* (MEXU).

**Comments**: In northeastern Mexico, recorded only in the state of Tamaulipas (rare); outside that area, it is frequent in disturbed areas, 300–1400 m. This species is easy to identify, since it is the only *Vachellia* in northeastern Mexico with slender spinescent paired stipules having more than 12 pairs of pinnae per leaf. Species with wide distribution [74], from the north of Mexico to the north of South America, Colombia and Ecuador. Its wood is very hard, useful for rural construction, handles for tools, posts for fences, firewood, charcoal, assembly, and joinery work. The leaves are used as forage for livestock.

***Vachellia rigidula*** (Benth.) Seigler and Ebinger. Basionym: *Acacia rigidula* Benth., London J. Bot. 1: 504. 1842. *Acaciopsis rigidula* (Benth.) Britton and Rose, N. Amer. Fl. 23: 94. 1928.

**Type**: United States, Texas, *T. Drummond 161!* (holotype: K297421!).

**Distinguishing features**: Shrub to small tree, up to 8 m tall. Stipules spinescent, paired, white. Pinnae 1 pair per leaf. Leaflets 3–5 pairs per pinna. Petiolar gland, solitary sessile. Inflorescences arising on short brachyblasts, arranged in cylindrical spikes. Flowers cream-colored. Fruit 4–10 × 0.3–0.6 cm, longitudinally striated, constricted between the seeds.

**Representative examined material**: Coahuila: 26-III-1992, *M.A. Carranza 1309* (ANSM), 20-VIII-1987, *J.A. Villarreal 3884* (ANSM). Nuevo León: 8-III-2003, *E. Estrada 15294* (CFNL); *9-IV-2001*, *E. Estrada 12077* (CFNL); 17-IV-2001, *E. Estrada 12467* (CFNL); 2-III-2003, *E. Estrada 15234* (CFNL); 23-VI-2001, E. Estrada 12765! (CFNL) 11-VI-1889, *C.G. Pringle 2526* (TEX-LL). Tamaulipas: 23-III-1999, *A. Mora-Olivo 7474a* (UAT); 21-II-1985, *R. Diaz 278* (UAT); 23-III-1985, *R. Diaz 304* (UAT); 13-VII-1982, *F. González-Medrano 220* (UAT); 9-II-1984, *L. Hernández 950* (UAT); 25-IV-1985, *M. Martínez 293* (UAT).

**Comments**: Species quite common in the lowlands of northeastern Mexico, associated with Tamaulipan thorn scrub, where it is one of the dominant species in density and canopy cover. In Nuevo León, it is one of the dominant species in several plant associations in the Northern Gulf Coastal Plain [74]. From southeastern USA and northeastern Mexico to San Luis Potosí and Veracruz. Roots of *V. rigidula* are used to manufacture products for hair care, and a boiled solution of the bark and leaves are used to control amoebiasis and gum diseases [53].

***Vachellia schaffneri*** (S. Watson) Seigler and Ebinger, Phytologia 87 (3): 167. 2005. Basionym: *Pithecellobium schaffneri* S. Watson, Proc. Amer. Acad. Arts 17: 352. 1882. *Samanea schaffneri* (S. Watson) J. Macbr., Contr. Gray Herb. 59: 2. 1919. *Poponax schaffneri* (S. Watson) Britton and Rose, N. Amer. Fl. 23: 89. 1928. *Acacia schaffneri* (S. Watson) F.J. Herm., J. Wash. Acad. Sci. 38:236. 1948.

**Type**: Mexico, San Luis Potosi, in the mountains around San Luis Potosi, *C.C. Parry* and *E. Palmer 219* (lectotype: GH64040!).

**Distinguishing features**: Tree with flat-topped crown. Stipules spinescent, paired. Pinnae 4–6 pairs per leaf, arising from brachyblasts. Leaflets 12–16 pairs per pinna. Petiole with a sessile gland, located at or just below the insertion of the proximal pair of pinnae, the rachis frequently with a gland between the insertion of the distal pair of pinnae and rarely in other pairs of pinnae. Inflorescence arranged in spheric capitula. Peduncle with minute red caducous glands, with a whorl of bracts immediately adjacent the capitula. Fruit 6–13 × 0.9–1.2 cm, straight or slightly curved, chestnut brown, with numerous minute red deciduous glands, septate between the seeds, indehiscent.

**Representative examined material**: Coahuila: Coahuila: 7-VI-1001, *J.A. Villarreal 6029* (ANSM). Nuevo León: 8-IV-2001, *E. Estrada 11925*! (CFNL); *8-III-2003*, *E. Estrada 15285*! (CFNL); 19-VII-2002, *E. Estrada 14921* (CFNL). Tamaulipas: 2-VII-1986, *M. Martínez 179* (UAT); 4-VI-1985, *J. Jiménez 193* (UAT); 24-VII-1987, *D. Méndez 66* (UAT).

**Comments**: Endemic to Mexico, frequent in disturbed areas, in northeastern Mexico, it has been recorded in Tamaulipan thorn scrub and desert scrublands, also in the High Plains of Chihuahua, Coahuila and Durango to Guanajuato, Puebla, Hidalgo, and Oaxaca, 360–2000 m. The foliage is used as forage and the wood as firewood.

***Vachellia sphaerocephala*** (Schltdl. and Cham.) Seigler and Ebinger, Phytologia 87: 167. 2006. Basionym: *Acacia sphaerocephala* Schltdl. and Cham., Linnaea 5: 594. 1830.

**Type**: Mexico, Veracruz, Actopan, sea level, III-1829. *C. Schiede* and F. *Deppe 684* (lectotype, US615!).

**Distinguishing features**: Shrub, 2–4 m tall. Stipules spinescent, paired, straight or slightly reflexed, 2–8 × 0.6–1.5 cm (basally), basally swollen, myrmecophila. Petiolar glands canoe-shaped, striate on the sides, 1.4–4.4 mm long, arising near the middle of the petiole. Pinnae 5–15 pairs per leaf. Leaflets 15–47 pairs per pinna, 10 mm long or shorter, with 1 vein from the base, lateral veins not obvious. Inflorescences axillary, arranged in globose or subglobose capitula. Peduncle with an involucre of 4 bracts located at the base. Fruit 3–8 × 1.2–1.6 cm, straight, almost cylindrical, glabrous, longitudinally striate, red to maroon, indehiscent, beak 1–3 cm long.

**Representative examined material**: Tamaulipas: 4-III-1999, *A. Mora-Olivo 7423* (UAT); 25-III-1985, *M. Yanez 92* (UAT); 8-V-1984, *D. Baro 196* (UAT); 14-VI-1994, *J.L. Mora-López 531* (UAT); 3-VII-1979, *A. Mora-Olivo 7222* (UAT).

**Comments**: Endemic to Mexico. Usually distinguishable from *A. cornigera,* but in *V. sphaerocephala*, the secondary venation of leaflets is not evident on simple sight. Most frequently inhabiting coastal dunes and dry areas near the coast.

***Vachellia vernicosa*** (Britton and Rose) Seigler and Ebinger, Phytologia 87: 169. 2006. Basionym: *Acacia vernicosa* Standl., Cont. U. S. Natl. Herb. 20:187. 1919. *Acaciopsis vemicosa* Britton and Rose, N. Amer. Fl. 23: 96. 1928. *Acacia constricta* Benth. var. *vemicosa* L. Benson, Amer. J. Bot. 30: 238. 1943. *Acacia neovernicosa* Isely, Sida 3:380. 1969.

**Type**: Mexico, Chihuahua, in the vicinity of Santa Rosalia, alt. 1200 m, 13–15-VI- 1908, *E. J. Palmer 385*! (holotype: US636!. Isotype: NY1506!; GH58245!).

**Distinguishing features**: Shrub, up to 2.8 m tall. Leaves and younger shoots glutinous. Stipules spinescent, paired. Pinnae 1–3 pairs per leaf, rachis up to 5 mm long, arising from brachyblasts. Petiolar gland sessile, inserted between the proximal pair of pinnae, the rachis also with a small gland inserted between the distal pair of pinnae. Leaflets 8–10 pairs per pinna, with no evident secondary venation. Inflorescences on brachyblasts, arranged in spheric capitula. Peduncles with a whorl of involucral bracts near the middle. Fruit 5.7–9 × 3–4 cm, linear, compressed, constricted between seeds, glabrous, glutinous, dehiscent, longitudinally striate.

**Representative examined material**: Coahuila: 5-VI-1990, *J.A. Villarreal 5690* (ANSM). Nuevo León: 7-VII-2001, *E. Estrada 12990*! (CFNL). 8-IX-2001, *E. Estrada 13043*! (CFNL).

**Comments**: Reported in Coahuila and Nuevo Léon, inhabiting dry scrublands in semiarid mountain slopes in the north of Nuevo León (600–700 m), and high plains (1400–1650 m) in Coahuila. Also in SW and SE of the USA, in Mexico, from Chihuahua to Zacatecas and San Lui Potosí.

**Tribe Mimoseae** Bronn, Form. Pl. Legumin. 78, 127, 130. 1822.

Trees, shrubs, vines or annual or biennial herbs, armed with spines or thorns or unarmed. Leaves commonly bipinnate, with or without foliar glands. Inflorescences arranged in spikes, capitula, racemes, or panicles. Flowers 3–6-merous, bisexual, occasionally with staminate or sterile flowers at lower portions of the inflorescence. Calyx valvate. Corolla commonly valvate. Petals free or united basally. Stamens same number as the petals or double, exerted, commonly free, very rarely shortly fused at base, anthers with or without sessile or stipitate apical gland. Ovary sessile or stipitate. Fruit bivalvate, commonly compressed, sometimes cylindrical, torulose, spiral or tetragonal, rarely winged, straight to curved, the valves papery, chartaceous or woody, indehiscent or dehiscent along 1 or both sutures, valves sometimes separate from the persistent margin (replum), dividing transversally into segments (mericarps), each with a single seed. Seeds compressed, without aril, with or without endosperm.

In tropical and subtropical areas of America, Africa, and Asia. Common in arid and semiarid zones, less common in temperate zones. This tribe comprises about 40 genera and 860–880 species worldwide [2], most belonging to the genus *Mimosa* (480 species) [77]. Several genera, such as *Neltuma* and *Leucaena,* are used as timber species in northeastern Mexico [78].
1A.Anthers provided with an apical gland between the theca of the anther21B.Anthers without apical gland between theca42A.Herbaceous plants, unarmed***Neptunia***2B.Creeping shrubby or arboreal plants, armed with spines33A.Plants armed with stipular spines; creeping shrubs, growing in colonies; 50 cm tall or less; inflorescences arranged in spherical capitula***Strombocarpa***3B.Plants armed with axillary spines, uninodal, solitary or in pairs or in spinescent shoots; shrubby or treelike, erect, not growing in colonies, more than 1 m tall; inflorescences arranged in oblong spikes***Neltuma***4A.Plants armed with thorns or unarmed; shrubs with pink flower capitula and only 1–2 pairs of pinnae per leaf; fruit flattened, with a continuous and persistent margin, the valves separating from it when the legume matures, or with a tetragonal appearance (2 entire valves narrower or equal to the width of the persistent margin) separating into 2 equal valves and each of these dividing into 2 merivalves when the fruit ripens; inflorescence peduncles with or without spines***Mimosa***4B.Unarmed plants55A.Prostrate herbaceous or sub-shrubs; fruit 9 mm wide or narrower, seeds arranged obliquely***Desmanthus***5B.Shrubs or trees; fruit 9 mm wide or wider; seeds arranged longitudinally, transversely, or obliquely***Leucaena***

***Desmanthus*** Willd., Sp. Pl. (ed. 4), 4 (2): 888, 1044–1049. 1806.

**Type species**: *Desmanthus virgatus* (L.) Willd., Sp. Pl., ed. 4 [Willd.] 4 (2): 1047. 1806.

Herbaceous, perennial, suffruticose, basally branched, prostrate or erect. Stems angled and grooved. Stipules subulate or setiform, base auriculate, usually persistent and evident. Leaves bipinnate, pinnae 1–15 pairs per leaf. Leaflets 4–30 pairs per pinna. Petiole gland arising between the insertion of the proximal pair of pinnae or at next pinnae insertion. Inflorescences axillar, arranged in subspherical, ovoid to elliptic capitula or noticeably short spikes. Flowers 5-merous, petals free or united only at the base, white, perfect or the lower ones unisexual, staminate or neutral, and with staminodes. Stamens 5–10, free. Fruit linear or broadly oblong, falcate or straight, flattened, marginally dehiscent, subseptate between seeds, thin or strongly coriaceous. Seeds arranged obliquely along the length and width of the pod.

An American genus constituted of 24 species; most of the species (14) occur in Mexico; 7 of them are endemic [2,79].
1A.Young stems densely pubescent, villous or velutinous; stipules pubescent; plants prostrate or decumbent 21B.Young stems glabrous or sparsely pubescent on the edges; stipules usually glabrous; plants erect or prostrate 32A.Young stems villous, the pubescence concentrated on the edges; petiole gland elliptic, 0.6–1.4 mm broad; fruits 3.5–5.5 mm wide, apiculate, densely aggregated on peduncle, valves never curling after dehiscence***D. painteri***2B.Young stems completely velutinous; petiole gland orbicular, 0.3–0.6 mm diameter; fruits 2–3.5 mm wide apiculate or rarely obtuse at apex, not densely aggregated on peduncle, valves curling after dehiscence***D. velutinus***3A.Leaflets with raised venation abaxially***D. obtusus***3B.Leaflets with unraised elevation veins abaxially or only the eccentric midvein visible44A.Leaflets 20–50 pairs per pinna; pinnae often lanceolate, leaflets gradually decreasing in size distally; stipules with small auricles***D. paspalaceus***4B.Leaflets 25 pairs per pinna or fewer; pinnae oblong, leaflets not tapering distally; stipules with well-developed auricles55A.Peduncles 3.5–7.5 cm long; fruits with black, reticulate venation***D. pringlei***5B.Peduncles 0.5–3 cm long; fruits without black, reticulate venation66A.Petiolar gland more than 1, 1 more at the insertion of the distal pair of pinnae; fruit valves curling after dehiscence***D. glandulosus***5B.Petiolar gland 1, at the insertion of the proximal pair of pinnae; fruit valves not curling after dehiscence***D. virgatus***

***Desmanthus glandulosus*** (B.L. Turner) Luckow, Syst. Bot. Monogr. 38: 77. 1993.

Basionym: *Desmanthus virgatus* (L.) Willd. var. *glandulosus* B.L. Turner, Field and Lab. 18 (2): 64–65. 1950.

**Type**: U.S.A. Texas: Terrell Co., 6 mi E of Sanderson, 23.VIII-1947, *Warnock 6710* (Isotype: TEX371136!).

**Distinguishing features**: Herbaceous, erect to decumbent. Stems up to 70 cm long, strongly angled, sparsely pubescent or old stems cylindrical and glabrous. Stipules persistent, sometimes soon deciduous, 1.2–6 mm long. Pinnae 3–6 pairs per leaf. Leaflets 14–26 pairs per pinna. Petiole with a sessile, 0.9–3.2 mm diameter gland, arising between the proximal pair of pinnae and another in the distal pair of pinnae. Inflorescences arranged in capitula, 0.7–1.1 cm diameter. Flowers, most of them perfect, several (3–7) female in the lower part, petals green. Fruit 5.8–10.5 × 0.3–0.4 cm, linear, with a short beak, leathery, glabrous, brown to blackish-brown, reticular-veined, 4-angled, brown.

**Representative examined material**: Coahuila: 12-VIII-1975, *T. Reeves 13024* (ASU0021239!); 17-VII-2008, *J.A. Encina 2586* (ANSM); 17-IX-1989, *E. Estrada 1826* (MEXU); 5-V-1989, *J.A. Villarreal 4854*, *R. Vázquez* (MEXU). Nuevo León: 19-VIII-1988, *T.F. Patterson 68822* (TEX-LL).

**Comments**: Recorded at Coahuila and Nuevo León, in arid shrublands, piedmont scrub, and oak–pine forest, 500–2150 m. Also in the southern USA (New Mexico and Texas).

***Desmanthus obtusus*** S. Watson, Proc. Amer. Acad. Arts 17: 349-,371. 1882. Basionym: *Acuan obtusa* (S. Watson) Heller, Cat. N. Amer. pi. 4. 1898.

**Type**: U.S.A. Texas: T and PRR, VIII-1881, *Havard s.n.* (holotype: GH53727!. Isotype: US169920!).

**Distinguishing features**: Herbaceous, up to 0.5 m tall, basally branched. Pinnae 1–4 pairs per leaf. Foliar gland orbicular or triangular, arising in the proximal pair of pinnae or sometimes absent. Leaflets 6–15 pairs per leaf, 3–6 mm long, conspicuously raised veined from the base and raised secondary veins also present. Inflorescences axillar, arranged in spheric capitula. Flowers 6–14. Petals white to light green, up to 2.9 mm long. Stamens 10, up to 7 mm long. Fruit 2–5 × 0.2–0.3 cm, linear, sometimes constricted between seeds, coriaceous, glabrous, longitudinally reticulate.

**Representative examined material**: Coahuila: 7-VI-1968, *D.J. Pinkava 13051* (ASU21242!). Tamaulipas: 26-IV-1960, *M.C. Johnston 5359C*, *J.R. Crutchfield* (TEX257275!)

**Comments**: From Texas and New Mexico to Coahuila and Tamaulipas, in several plant communities, desert scrubland, Tamaulipan thorn scrub, oak forest, 100–1000 m.

***Desmanthus painteri*** (Britton and Rose) Standl., Publ. Field Mus. Nat. Hist., Bot. Ser. 11: 159. 1936.

**Type**: Mexico, Querétaro de Arteaga, between Higuerillas and San Pablo, *J.N. Rose*, *J.H. Painter* and *J.S. Rose 9810* (NY1755!). Basionym: *Acuan painteri* Britton and Rose, N. Amer. Fl. 23 (2): 134. 1928.

**Distinguishing features**: Perennial, herbaceous, prostrate, up to 70 cm long, glabrous or pubescent along angles on stems. Stipules persistent, filiform, 1.5–5 mm long, red. Pinnae 2–6 pairs per leaf. Leaflets 10–20 pairs per pinna. Petiole with an orbicular to transversely elliptic gland inserted between the proximal pair of pinnae, subsessile, 0.6–1.4 mm in diameter. Inflorescences solitary, arranged in ovoid capitula or noticeably short spikes. Petals 3–4 mm long, pale yellow-green or green with a red apex. Stamens 10. Fruit 2.5–4 × 0.3–0.5 cm, linear, straight, apiculate, dehiscent, glabrous, reddish or dark purple when ripe.

**Representative examined material**: Coahuila: 29-VI-2015, *J.A. Encina 5011* (ANSM), 29-IV-1980, *J.A. Villarreal 659* (ANSM). Nuevo León: 8-X-1986, *Hinton* et al. *18996b* (TEX-LL); 7-VI-2003, *E. Estrada 15673* (CFNL); 17-IV-2001, *E. Estrada 12438* (CFNL); 22-VIII-1984, *M. Lavin 4762* (TEX-LL).

**Comments**: Endemic to Mexico. Recorded in Coahuila and Nuevo Léon, most frequently in low hills. Tamaulipan thorn scrub, piedmont scrub and desert scrublands of high hills, and oak–pine forests of the Sierra Madre Oriental, 265–1930 m. Also through north and central Mexico to Oaxaca.

***Desmanthus paspalaceus*** (Lindman) Burkart, Darwiniana 7: 221. 1946. Basionym: *Acuan virgatum* f. *paspalacea* Lindman, Bih. Kongl. Svenska Vetensk.-Akad. Handl. 24, Afd. 3, 7: 44. 1898.

**Type**: Brazil, Rio Grande do Sul: cachoeira, in campis, 24-II-1893, *Lindman A1201* (holotype: S-R-8651S09–28715!).

**Distinguishing features**: Herbaceous, perennial, erect herb up to 1 m tall, dead stems persistent at the base. Young stems angled, glabrous, turning cylindrical, shiny, red to brown, glabrous with age. Stipules persistent, 2.0–4.3 mm long. Pinnae 1–4 pairs per leaf. Leaflets 23–54 pairs per pinna, gradually decreasing in size distally. Petiole with a 1.5–3.5 mm long, sessile, elliptic–obovate gland, arising between the 2 proximal pairs of pinnae or in the petiole. Inflorescences arranged in capitula or short and compact spike. Petals light green with white margins. Fruit 4.0–8.0 cm × 0.25–0.4 mm, linear, beaked, dehiscent, glabrous, black at maturity, slightly wrinkled with raised reticulate veins.

**Representative examined material**: Tamaulipas: 3-VII-1930, *Bartlett 10043* (MICH1164528!); 19-VII-1930, *H.H. Bartlet 10464* (MICH1164526!); 6-VII-1930, *Bartlett 10118* (MICH1164527!).

**Comments**: Recorded in the mountains of the state of Tamaulipas, rare. Also in South America; most frequent in Argentina, Bolivia, Paraguay, and Uruguay.

***Desmanthus pringlei*** (Britton et Rose) F.J. Herm., J. Wash. Acad. Sci. 38 (7): 237. 1948. Basionym: *Acuan pringlei* Britton et Rose, N. Amer. Fl. 23 (2): 134. 1928.

**Type**: Mexico, Nuevo León, Monterrey; rich shaded places, 26-VI-1888 *Pringle 1902* (holotype: 1759NY!, isotype: M218689!; M218690!; USDA-ARS (NA), NA52774!; US00930729).

**Distinguishing features**: Perennial herbaceous, prostrate or slightly ascending. Stems up to 60 cm long, 4–5, angled, sparsely puberulent, with white trichomes concentrated only on the angles of the stems. Stipules 2.0–5.1 mm long, setiform, auriculate, glabrous. Pinnae 3–7 pairs per leaf; leaflets 15–23 pairs per pinna, glabrous. Petiole with a gland arising between the proximal pair of pinnae, sessile, 0.3–0.9 mm long. Inflorescences arranged in capitula. Petals 2.2–3 mm long, pale green. Fruit, 5–6 × 0.35–0.4 cm, linear or slightly falcate, dehiscent, beaked, reticulate, glabrous, mahogany brown to black.

**Representative examined material**: Nuevo León: *E. Estrada 14917* (CFNL); 5-VII-1985, *M. Luckow 2676* (TEX-LL); 5-VII-1985, *M. Luckow 2678* (TEX-LL); *B.L. Turner* and *A.M. Powell* 1025 (TEX-LL); 20-VI-2002, *E. Estrada 14632!* (CFNL).

**Comments**: Endemic to the central area of Nuevo León, in piedmont scrub, riparian vegetation and oak forests, 290–830 m. Easily distinguished from the other species by its mature pods showing black reticulate venation.

***Desmanthus virgatus*** (L.) Willd., Sp. Pl. 4: 1047. 1806. Basionym: *Mimosa virgata* L., Sp. Pl. 519: 1753. *Acuan virgatum* (L.) Medik., Theodora p. 62. 1786.

**Type**: “in India” (holotype: LINN, microfiche IDC 715:III.3!).

**Distinguishing features**: Herbaceous, perennial, prostate or erect, up to 1.3 m tall. Stems angled. Stipules persistent, lanceolate, basally auriculate up to 9 mm long. Pinnae 2–5 pairs per leaf. Leaflets 10–23 pairs per pinna. Inflorescences arranged in spheric or sub-spheric capitula. Fruit 2–9 × 0. 2–0.4 cm, linear, straight to curved, acute at both ends, chartaceous, reddish, brown to black, glabrous, reticulate veined.

**Representative examined material**: Coahuila: 22-VIII-2007, *M.A. Carranza 4737*, *I. Ramírez* (MEXU)**.** Nuevo León: 13-VII-2002, *E. Estrada 14825* (CFNL); 23-VII-2002, *E. Estrada 14988* (CFNL); 24-VI-2001, *E. Estrada 12828* (CFNL); 24-VI-2001, *E. Estrada 12800* (CFNL)**.** Tamaulipas: 22-IX-1992, *A. Mora-Olivo 5220* (UAT); 18-IV-1994, *A. Mora-Olivo 5351* (UAT); 3-X-1984, *D. Baro 472* (UAT); 3-VI-1977, *A. Mora-Olivo 7219* (UAT); 28-VI-2002, *E. Estrada 14754* (CFNL).

**Comments**: From southern Texas through Mexico, Central America, and Antilles to Brazil and Argentina in South America. The foliage and fruits are used as fodder for domestic cattle.

***Leucaena*** Benth., Hook. J. Bot. 4: 416. 1842.

**Type species**: Mexico, Jalapam, *C.J.W*: *Schiede s.n.* (Isotype: GH277337!)

Unarmed shrubs or trees, up to 20 m tall. Stipules caducous or persistent, asymmetrically winged basally. Leaves bipinnate, with a sessile or stipitate gland in the petiole and 1 to several glands arising in the terminal and subterminal insertion of pairs of pinnae. Pinnae 2 to several pairs per leaf. Leaflets abundant pairs per pinna. Inflorescences axillar, in fascicles, arranged in spherical capitula. Peduncle with a whorl of bracts at distal end. Flowers 5-merous, white, white-cream, yellow. Petals 5, free or united basally. Stamens 10. Fruit short stipitate, pendulous, flattened, oblong to linear oblong, rounded or acute apically; membranous, chartaceous, or coriaceous; dehiscent; brown, reddish; glabrous or pubescent. Seeds circular, ovate to rhomboid, compressed, transversally or obliquely aligned.

A New World genus distributed from Texas through Mexico to Peru. It is composed of 22 species [80], with the highest diversity in Mexico, where there are 17 species, 10 of which are endemic.
1A.Leaflets elliptic, ovate, or lanceolate, slightly asymmetric at base, 1 cm or more broad, leaflets 2–8 pairs per pinna***L. retusa***1B.Leaflets linear, narrow–oblong, strongly asymmetric at the base, less than 1 cm wide, Leaflets 5-7-abundant pairs per pinna22A.Young shoots angled with corky ridges, these visible as distinct striations on the stem ***L. esculenta***
Young shoots cylindrical32B.Petiolar gland stipitate, clove-shaped or sessile and concave, crateriform or patelliform, orifice is wide33A.Petiolar gland sessile, convex, shallowly conical, truncate–conical, verruciform, the orifice is a narrow or invisible pore***L. pulverulenta***3B.Petiolar gland stipitate, clove-shaped or sessile and concave, crateriform or patelliform, orifice is wide44A.Petiolar gland cylindrical, subsessile or stipitate, club-shaped or circular***L. greggii***4B.Petiolar gland concave, broad domed, crateriform, elliptical or circular55A.Leaves 20 × 12 cm or less in length or width; rachis of pinnae 8 cm long or shorter, leaflets 9–13 mm long; capitula 1.2–1.7 cm diameter at anthesis; fruit 9–13 × 1.3–1.8 cm***L. leucocephala*** ssp. ***leucocephala***5B.Leaves 19 × 12 cm or more in length or width; rachis of pinnae 8 cm long or longer; leaflets (11-) 16–21 mm long; capitula greater than 1.8 cm in diameter in anthesis; fruit 12–19 × 1.8–2.1 cm***L. leucocephala*** ssp. ***glabrata***

***Leucaena esculenta*** (Sesse and Mociño ex DC.) Benth. Trans. Linn. Soc. 30: 442. 1875. Basionym: *Acacia esculenta* Sesse and Mociño ex DC., Prodr. 2: 470. 1825. *Mimosa esculenta* Sesse and Mocinlo, P1. Nov. Hisp. 178. 1890. *Leucaena confusa* Britton and Rose, N. Amer. Fl. 23: 128. 1928. *Leucaena doylei* Britton and Rose, N. Amer. FH. 23: 128. 1928.

**Type**: Mexico, Jalisco, Hills, Tequila, 18-X-1893, *C.G. Pringle 4534* (Isotype: MICH1168959!; NDG24031!; MO-127543!; G00371024!; GH00065790!).

**Distinguishing features**: Tree, up to 20 m tall. Shoots strongly angled with corky ridges. Petiole with 1–2 sessile gland (s), elliptic, concave, maroon-red arising at the distal end. Pinnae 25–60 pairs per leaf. Pinnae rachis with 3–4 elliptic glands arising at the base of terminal pair of leaflets. Leaflets 55–85 pairs per pinna. Inflorescences axillar, in fascicles of 2–7, arranged in spheric capitula. Fruit 10–30 × 2.3–2.6 cm, linear–oblong, flattened, reddish-maroon to orange-brown, glossy, margin reticulate.

**Selected examined material**: Tamaulipas: 25-IV-2022, *L. Hernández 8816* (UAT); 25-V-1996, *G. Sánchez 103* (MEXU); 20-XII-1990, *D.S. Seigler*, *J.E. Ebinger*, *H.D. Clarke* and C. *Gratton 13207* (MO-678948).

**Comments**: Endemic to Mexico, rare, recorded only in the state of Tamaulipas, in tropical rain forest 1200 m. In almost all states in Mexico, south of the Tropic of Cancer.

***Leucaena greggii*** S. Watson, Proc. Amer. Acad. Arts 23: 272. 1888. Basionym: *Ryncholeucaena greggii* (S. Watson) Britton and Rose, N. Amer. Fl. 23: 130. 1928.

**Type**: Mexico. Nuevo León: dry ravine east of Rinconada, 25-V-1847, *Gregg n.n.* (Syntype: GH65795!; NY2433!).

**Distinguishing features**: Small tree, up to 7 m tall. Stipules ovate, persistent. Pinnae 7–11 pairs per leaf. Leaflets 25–34 pairs per pinna. Rachis leaf with a cylindrical gland arising in the insertion of each pair of pinnae. Inflorescences axillar, arranged in spheric capitula. Peduncle with an obvious distal whorl of bracts. Petals yellow-green, filaments bright yellow (egg yolk color). Fruit pendulous, 12–18 × 0.9–1.5 cm, oblong or linear, flattened, slightly falcate, woody with age, brown to orange-brown, barely reticulate, glabrous, thickened marginally, dehiscent.

**Selected examined material**: Coahuila: Coahuila: 20-VI-1987, *A. Rodríguez 837* (ANSM), 5-VI-1992, *J.A. Villarreal 6613* (ANSM). Nuevo León: 9-IV-2001, *E. Estrada 12051*! (CFNL); 1-XI-1991, *G. Hinton* et al. *21733*! (TEX-LL); 10-V-2003, *E. Estrada 15568*! (CFNL); 12-IV-2003, *E. Estrada 15473*! (CFNL).

**Comments**: Endemic to the north of Mexico. Recorded only for the states of Coahuila and Nuevo León. In semiarid mountains, on sunny slopes, and in wet creeks. In soils with high calcium content, often in piedmont scrub, arid oak shrublands (chaparrales), and arid shrublands, 890–1900 m. In the flower stage, easily recognized by its egg yolk-colored capitula; the other species show white capitula. Used as fodder and, due to its showy foliage and contrasting yellow inflorescences, as an ornamental plant in regional private gardens.

***Leucaena leucocephala*** (Lam.) De Wit, Taxon 10: 53. 1961. ssp. ***leucocephala*.** *Mimosa leucocephala* Lamarck, Encycl. Meth. Bot. 1: 12. 1783. *Acacia leucocephala* (Lamarck) Link, Enum. hort. berol. 2: 444. 1822.

**Type**: (Holotype: P-LA, microfiche: K!) (according to Hughes (1998), not seen).

**Distinguishing features**: Small tree, up to 6 m tall. Pinnae 5–8 pairs per leaf. Leaflets 13–17 pairs per pinna, 10–12.5–13 × 2.4–3.3 mm. Rachis of pinnae 8 cm long or shorter. Petiole with a 2–2.3 × 1.4 mm elliptic gland. Inflorescences in fascicles, arranged in spheric capitula, 1.2–1.7 cm diameter at anthesis. Fruit 10–13 × 1.3–1.6 cm, flattened, oblong, apically rounded, but beaked, brown to light brown.

**Selected examined material**: Tamaulipas: II-1982, *Medrano 12206* (MEXU).

**Comments**: Species rare in northeastern Mexico, recorded for only a single locality (1 km south of El Abra, 10 km south of Cd. Mante). Outside of this area, it is distributed along the Gulf Coast from northern Veracruz to the Yucatán Peninsula and the state of Guerrero, also in the Antilles and Central America (Belize and Panama).

***Leucaena leucocephala*** ssp. ***glabrata*** (Rose) S. Zarate, Phytologia 63 (4): 305. 1987. Basionym: *Leucaena glabrata* Rose, Contrib. U.S. Natl. Herb. 5 (3): 140. 1897.

**Type**: Mexico. Guerrero: vicinity of Acapulco, X 1894-XII 1895, *Palmer 368* (Isotype: K527949!).

**Distinguishing features**: Tree, up to 18 m tall. Pinnae 4–9 pairs per leaf. Leaflets 15–21 pairs per pinna, 10–21 × 2.5–4.5 mm. Petiole with an elliptic to circular gland 2–3 × 1.5 mm. Rachis of pinnae 8 cm long or longer. Inflorescences in fascicles, arranged in spheric capitula, 1.2–1.7 cm diameter at anthesis. Fruits 12–19 × 1.8–2.2 cm, oblong, apically rounded, beaked, brown to orange-brown, glabrous, slightly lustrous.

**Selected examined material**: Coahuila: 5-VI-1992, *J.A. Villarreal 6622* (ANSM), 3-X.2008, *J. Valdés 3112* (ANSM). Nuevo León: 8-IV-2001, *E. Estrada 11920* (CFNL); 5-VI-2001, *E. Estrada* et al. *12883* (CFNL). 2-VII-2000, *C. Yen y E. Estrada 11505* (CFNL); *E. Estrada 11586* (CFNL); 30-VII-1970, *J.R. Sanok 43* (TEX-LL); 17-IV-2001, *E. Estrada 12498* (CFNL). Tamaulipas: 10-VIII-1992, *J.L. Mora-López 164* (UAT); 21-X-1983, *L. Hernández 771* (UAT); 25-II-1985, *R. Diaz 299* (UAT); 22-IX-1992, *A. Mora-Olivo 5218* (UAT).

**Comments**: Along coasts in the states of northern Mexico, through all states south of the Tropic of Cancer. Also in the south of the USA, Bahamas, Central America to Peru, Bolivia, and Brazil. Common in areas with human settlements, yards, and sidewalks, and places with disturbances. Most abundant below 1500 m. Widely used as forage [73,76]. The young pods and stems are used as food [73]; the foliage and fruits are given to domestic cattle as fodder. The leaves and fruits of *Leucaena leucocephala* subsp. *glabrata* mixed with the fleshy stems of *Opuntia ficus-indica* and *Acanthocereus tetragonus* are frequently used to make compost [74].

***Leucaena pulverulenta*** (Schltdl.) Benth., Hooker J. Bot. 4: 417. 1842. Basionym: *Acacia pulverulenta* Schltdl., Linnaea 12: 571. 1838.

**Type**: Mexico. Veracruz: “ad ripam fluminis Misantlensis, pr. San Antonio, reg. calidae”, 19°56′ N, 96°52′ W, Feb, *Schiede & Deppe s.n.* (Isotype: BM952388!); holotype: HAL0107639!)

**Distinguishing features**: Tree, 5–16 m tall. Young leaves white puberulent. Petiole with an elliptic, columnar shape; 3 × 1.5 mm; gland arising between the proximal pair of pinnae. Pinnae 12–18 pairs per leaf. Pinnae rachis with 1 or 2 glands, arising between distal pairs of leaflets. Leaflets 55–75 pairs per pinna. Inflorescences in fascicles, arranged in spheric capitula. Fruit 12–20 × 1.4–2.4 cm, oblong, with a short beak, flattened, thin, chartaceous, dark brown, glabrous, dehiscent.

**Selected examined material**: Nuevo León: 10-XI-2001, *E. Estrada 13234*! (CFNL); 15-VI-1989, *E. Estrada C. 1518*! (TEX-LL); *C.E. Hughes 1047*! (CFNL); 1-VI-1987, *E. Estrada 1277*! (CFNL); 1-VII-1956, *B.L. Turner 3979*! (TEX-LL). Tamaulipas: 28-IV-1985, *M. Martínez 334* (ANSM), 28-VI-1983, *R. Torres 3126* (ANSM); 11-V-2000, *A. Mora-Olivo 8157* (UAT); 22-VII-1989, *J.L. Mora-López 48* (UAT); 19-VI-1985, *L. Hernández 1453* (UAT); 26-IV-1985, *L. Hernández 334* (UAT); 5-V-1992, *J.L. Mora-López 108* (UAT).

**Comments**: Low plains and east-facing slopes of the Sierra Madre Oriental in Nuevo León and Tamaulipas, associated with piedmonts scrub and oak–pine forest, 550–1600 m. Also in southern Texas (USA) to central Veracruz. Widely used ornamentally in private and public gardens. The leaves and fruits are used as fodder for cattle.

***Leucaena retusa*** Benth. in Gray, Plantae Wrigh. 1: 64. 1852. Basionym: *Caudoleucaena retusa* (Benth.) Britton and Rose, N. Amer. Fl. 23: 131. 1928.

**Type**: U.S.A., Texas: bottom of the Rio Nueces, VI-1849, *Wright 171* (Isotype: GH65780!; Syntype: US1108114!).

**Distinguishing features**: Shrub or tree, 2–7 m tall. Petiole with several cylindrical, columnar glands in the rachis at the insertion of each pair of pinnae. Pinnae 2–5 pairs per leaf. Leaflets 4–8 pairs per pinna. Inflorescences axillar, in fascicles, arranged in spheric capitula. Fruit 11.5–22 × 1–1.4 cm, linear to oblong, straight or slightly falcate, cuneate apically, compressed, reddish-brown, glabrous, strongly reticulate, strongly coriaceous, almost lignified, the margins thickened, dehiscent.

**Selected examined material**: Coahuila: 11-X-1991, *M.A. Carranza 990* (ANSM), 11-IX-1991, *L. García 1118* (ANSM), 15-IX-1992, *J.A. Villarreal 7022* (ANSM).

**Comments**: Recorded only for central and northern Coahuila, in rocky soils, arid shrublands, on dry and cool slopes, and in canyons, 1200–1850 m. From the south of Texas and New Mexico (USA), rare in Chihuahua, associated with oak, juniper forest, and chaparral.

***Mimosa*** L., Sp. Pl. 516: 1753 and Gen. pl. ed. 5: 233. 1754

**Type**: *Mimosa pudica* L., Sp. Pl.: 518. 1753.

Shrubs or herbaceous; unarmed or armed with 1–3 straight, recurved, or antrorse thorns; sometimes armed on several or all ribs of each internode with recurved thorns, these extending to the leaf-axes. Leaves commonly bipinnate, rarely unipinnate, the first pair of leaflets of each pinna commonly differentiated into paraphyllidia. Inflorescences solitary or in fascicles, axillar or terminal. Flowers 3–6-merous, calyx campanulate, gamosepalous, corolla gamopetalous. Stamens as many or twice as many as corolla-lobes, the filaments free. Fruit flattened, with continuous replum corresponding to the sutures, the valves continuous and separating from replum, either along both sides or from apex downward, in a single piece, or craspedial.

Genus of ±480 species [77], of which 90% are native to the New World, 102 in Mexico [81]. At least 14 species have been recorded in northeastern Mexico, associated with Tamaulipan thorn scrub (100–450 m), mountain slopes, in piedmont scrub and oak forest (650–1300 m), frequently present in high plains (1400–1800 m); associated with arid scrublands, chaparrales, conifer forests, and rain forest. Quite frequent in secondary vegetation and areas with disturbance by cultivation, overgrazing and immoderate felling of vegetation.
1A.Flowers 3-merous***M. guaranitica***1B.Flowers 4–6-merous22A.Flowers almost always 4-merous; stamens always 4; if 5 stamens, then only 1 pair of pinnae per leaf and 2 pairs of leaflets per pinna, but the inner leaflet of the proximal pair reduced or absent32B.Flowers 4–6-merous; stamens 5–1053A.Leaflets 11–36 pairs per pinna; flowers always 4-merous; stamens 4***M. pudica*** ssp. ***hispida***3B.Leaflets 2 pairs per pinna; flowers 4–5-merous; stamens 4–544A.Pod thinly strigulose and pulverulent or glabrate***M. albida*** ssp. ***glabrior***4B.Pod, including valves and replum coarsely strigose***M. albida*** ssp. ***albida***5A.Stems, leaf rachises and frequently leaflets margins with bulbous or dilatated basally, flagelliform setae65B.Stems leaf rachises and frequently leaflets margins lacking with bulbous or dilated basally, flagelliform setae**7**6AHerbaceous, the stems compressed, rooting at nodes; capitula elliptic or short cylindric***M. strigillosa***6BStems woody, erect, never compressed, never rooting at nodes; capitula subglobose***M. pigra*** ssp. ***asperata***7AStems and rachis of leaves serially armed with curved thorns87B.Stems unarmed, or if armed, then with groups of 1–3 thorns per node or internode, if the thorns are more numerous, then straight or nearly so and the rachis of the leaves rarely thorny128A.Flowers always 4-merous; scandent sub-shrubs; filaments white (capitula with white appearance)***M. malacophylla***8B.Flowers 5-merous; herbaceous prostrate; filaments pink (capitula with pink appearance)99A.Leaflets with evident reticulate venation***M. paucijuga***9B.Leaflets with no evident reticulate venation1010A.Larger leaves with the petiole 2.5–3.5 times longer than the rachis leaf (Tamaulipas)***M. latidens***10B.Larger leaves with the petiole 1.5–2.5 times longer than rachis leaf1111A.Pinnae 1–3 pairs per leaf; leaflets with rounded or obtuse apices; fruit 3–5 cm long***M. potosina***11B.Pinnae 3–4 pairs per leaf; leaflets with acute apices; fruit 6–8 cm long***M. monclovensis***12A.Plants unarmed; leaflets 1–4 cm long1312B.Plants armed with thorns; leaflets shorter than 1 cm long1413A.Each pinna with 2–5 pairs of leaflets; fruit 6–8.5 mm wide***M. leucaenoides***13B.Each pinnae with 1pair of leaflets; fruit 9–12 mm wide***M. martin delcampoi***14A.Leaves unipinnate; leaflets 1–3 pairs per pinna arranged on a primary axis***M. unipinnata***14B.Leaves bipinnate; leaflets developing into secondary axes1515A.Largest leaves with 5 or more pairs of pinnae or largest pinnae with 9 or more pairs of leaflets***M. biuncifera***15B.Largest leaves with 1–4 pairs of pinnae or largest pinnae with 10 or fewer pairs of leaflets1616A.Infrapetiolar thorn arising immediately below the node, alone or associated with 1–2 infrastipular thorns, these displaced towards the internode1716B.Infrapetiolar thorn solitary or associated with 1–2 infrastipular thorns, but all 3 displaced towards the internode1817A.Leaflets sericeous in both faces; fruit is a craspedium; all capitula or some of them axillary on new branchlets***M. monancistra***17B.Leaflets glabrous; fruit with valvate dehiscence; all capitula arising from brachyblasts***M. texana***18A.Leaflets or puberulent; valves of the pod puberulent, velutinous, hispid, villous or setose1918B.Leaflets and fruits glabrous2119A.Fruits densely setose; the setae in turn setose with the apex slightly uncinulate; leaves 4.5–7 cm long, pinnae 3–5 pairs per leaf***M. setuliseta***19B.Fruits with villous and uncinulate setae; leaves 2.8 cm long or shorter; pinnae 1–3 pairs per leaf2020A.Calyx 0.8–1 mm long, ciliate; fruit with cylindrical setae, villous at the base and uncinulate apically; leaves with sericeous pubescence***M. emoryana*** ssp. ***emoryana***20B.Calyx 0.5–0.8 mm long, conspicuously ciliated; fruit with flattened villous setae, uncinulate apically; leaves with dense sericeous pubescence***M. emoryana*** ssp. ***canescens***21A.Petioles rounded dorsally with a ventral groove; flowers sessile; calyx 0.9–1.5 mm long; leaves with 1 pair of pinnae, each pinna with 1 pair of leaflets (rarely 2 pairs); valves separating from replum in a single piece***M. zygophylla***21B.Petioles dorsoventrally compressed, with 2 grooves, both dorsally and ventrally; short-stalked flowers; calyx 0.4–0.9 mm long; leaves with 2 or more pairs of pinnae and pinnae with 2 or more pairs of leaflets***M. turneri***

***Mimosa albida*** Humb. and Bonpl. ex Willd., Sp. Pl., ed. 4, 4: 1030. 1806. ssp. ***albida***. Basionym: *Mimosa strigosa* Willd. Sp. Pl. 4: 1030. 1806. *Mimosa albida* var. *strigosa* (Willd.) B.L. Rob., Proc. Amer. Acad. Arts33: 311. 1898. *Mimosa racemosa* Schltdl., Linnaea 12: 557. 1838.

**Type**: Peru, Moche, *A.J.A. Bonpland 375-b* (P00679330!).

**Distinguishing features**: Herbaceous perennial or sub-shrub. Stems prostrate or decumbent, armed with scattered and curved thorns, setose or setulose or both, and pulverulent. Young branches and leaves occasionally with milky latex. Leaves bipinnate. Pinnae 1 pair per leaf. Leaflets 2 pairs per pinna, but the inner leaflet of the proximal pair reduced or absent, 2.5–9 × 1.1–3.5 cm, ovate-acuminate, broadest in the middle, apically acuminate rarely obovate, basally semi-cordate, bicolored, setose or pulverulent on both faces, rarely glabrous. The anterior leaflet similar in shape and size, sometimes smaller. Inflorescences axillar, in fascicles, arranged in pink spheric capitula. Fruits sessile or stipitate, stipe 1–4 mm long, the body 1–4.3 × 0.5 cm, coarsely strigose.

**Selected examined material**: Tamaulipas: 27-IX-1969, *M.C*: *Johnston4061A* (MEXU)

**Comments**: Recorded only in the state of Tamaulipas, in lowlands, semi-deciduous woods, tropical forest, and tropical rain forest, 100–700 m. From northeastern Mexico through Central America to South America, Peru, and Bolivia.

***Mimosa albida*** Humb. and Bonpl. ex Willd. ssp. ***glabrior*** B.L. Rob., Proc. Amer. Acad. Arts 33: 311. 1898. Basionym: *Mimosa sesquijugata* Donn. Sm., Bot. Gaz.13: 74. 1888. *Mimosa manzanilloana* Rose, Contr. U.S. Natl. Herb.1: 326. 1895. *Mimosa albida* var. *euryphylla* B.L. Rob., Proc. Amer. Acad. Arts33: 311. 1898. *Mimosa mazatlana* M.E. Jones, Contr. W. Bot. 15: 133. 1929.

**Type**: Mexico, Manzanillo, 1-XII-1890/31-XII-1890, *E. Palmer 905* (Holotype: US881!; Isotype: BM952345!; P705099!; UC82335!; GH65119!; GH65120!; MO-127558!; MICH1104201!; MEXU191465!).

**Distinguishing features**: Very similar morphologically to *M. albida* var. *albida*, but with the fruits thinly strigulose and pulverulent in addition to pulverulent pubescence, or glabrate.

**Selected examined material**: Tamaulipas: 20-V-1973, *M.C. Johnston 11160*, *T. Wendt*, *F. Chiang C.* (LL00230457!).

**Comments**: From northern Mexico to Guatemala, in Tamaulipan thorn scrub, semi-deciduous woods, tropical forest, tropical rain forest, 300–800 m.

***Mimosa biuncifera*** Benth. Pl. Hartweg., 12. 1839. Basionym: *Mimosa lindheimeri* A. Gray, Boston J. Nat. Hist. 6: 181. 1850. *Mimosa biuncifera* var. *lindheimeri* (A. Gray) B.L. Rob., Proc. Amer. Acad. Arts 33: 328. 1898. *Mimosopsis lindheimeri* (A. Gray) Britton and Rose, N. Amer. R. 23 (3): 177. 1928. *M. flexuosa* Benth. in A. Gray, Pl. wright. 1: 62. 1852. *Mimosa biuncifera* var. *flexuosa* B.L. Rob., Proc. Amer. Acad. Arts 33: 327. 1898. *Mimosopsis flexuosa* (Humb. and Bonpl. ex Willd.) Poir., J.B.A.M.de Lamarck, Encycl., Suppl. 1: 79. 1810. *Mimosa warnockii* Field and Lab. 24: 15. 1956. *Mimosopsis arida* Britton and Rose, N. Amer. R. 23 (3): 178. 1928. *M. biuncifera* var. *glabrescens* A. Gray, Pl. wright. 2: 51. 1853. *Mimosa aculeaticarpa* var. *biuncifera* (Benth.) Barneby, Mem. New York Bot. Gard. 65: 98. 1991.

**Type**: Mexico, Zacatecas, *K.T. Hartweg 69* (K82095!).

**Distinguishing features: Shrub,** 0.5–3 m tall, branches ribbed; armed with infra-stipular horns, paired or solitary, rarely in groups of 3. Pinnae 5 or more pairs per leaf. Leaflets 9 or more pairs per pinna. Inflorescences solitary or in fascicles, arranged in spheric capitula. Flowers 5-merous, white or pinkish on the lobes. Stamens 10. Fruit 1–4.5 × 0.5–1.1 cm, linear flattened, puberulent with resinous dots, reddish-brown, spiny at margins, rarely unarmed.

**Selected examined material**: Coahuila: 9-X-1992, *J.A. Villarreal 7156* (ANSM), 31-VIII-1997, *M.A. Carranza 2606* (ANSM); Nuevo León: VII-1977, *C. Wells y G. Nesom 416* (TEX-LL); 21-VI-1969, *G. Hinton* et al. *17106* (TEX-LL); 13-VII-1989, *E. Estrada 1591* (TEX-LL); 26-VIII-1936, *M. Taylor 212* (TEX-LL); 10-VI-2003, *E. Estrada 15581* (CFNL). Tamaulipas: 23-VI-1996, *C. Ramos 21* (MEXU).

**Comments**: In scrublands, oak forest, oak–pine forest, chaparral, and occasionally in gypsophilous grassland, 900–2500 m. From Nuevo León and Tamaulipas to Chiapas and Michoacán. This is the only species of shrubby *Mimosa* with curved thorns and 5 or more pairs of pinnae per leaf in northeastern Mexico. The other species with 5 or more pairs of pinnae per leaf in northeastern Mexico, *M. pigra* ssp. *asperata*, has straight white thorns. Outside of the area, in the southern USA (Arizona, New Mexico, and Texas), in Mexico, from Chihuahua to Tamaulipas, San Luis and Zacatecas.

***Mimosa emoryana*** Benth., Trans. Linn. Soc. London 30 (3): 426. 1875. ssp. ***emoryana***. Basionym: *Mimosa emoryana* Benth., Trans. Linn. Soc. London 30: 426. 1875.

**Type**: Mexico, Chiefly in the valley f the Rio Grande, below Doña Ana, *C.C. Parry*, *D.J.M. Bigelow*, *C. Wright*, *A. Schott 302* (Holotype: K82489!)

**Distinguishing features**: Erect shrub, up to 1.7 m tall; branches armed with recurved thorns below nodes with 1 infra-petiolar and 2 infra-stipular horns. Pinnae 2–4 pairs per leaf. Leaflets 3–7 pairs per pinna, sericeous. Inflorescences axillar or in small brachyblasts, arranged in spherical or obovoid capitula. Flowers 5-merous. Corolla white. Stamens 10, filaments pink. Fruit 2–5 × 0.5–0.6 cm, linear–oblong, densely puberulent, setose, setae scabrulous, cylindric, villous basally and uncinulate distally, falling off in single-seeded articles.

**Selected examined material**: Coahuila: 11-X-1991, *M.A. Carranza 984* (ANSM), 8-V-1992, *A. Rodríguez 1576* (ANSM), 15-IX-1993, *J.A. Villarreal 7382* (ASM); Nuevo León: 7-VII-2001, *E. Estrada* et al., *12992* (CFNL). 7-VII-2001, *E. Estrada* et al., *13031* (CFNL).

**Comments**: Recorded in the states of Coahuila and Nuevo León. In desert scrublands, desert thorn scrub, stony hills, limestone soils, ecotones of Tamaulipan scrub and Chihuahuan desert, center–north of Nuevo León, 450–560 m. Outside the study area, from southern Texas through Chihuahua to northwestern Durango.

***Mimosa emoryana*** ssp. ***canescens*** Villarreal, Acta Botanica Mexicana 20: 50. 1992.

**Type**: Mexico, Durango, Municipio Cuencamé, 4 mi. north of Perdiceña, turnoff along Hwy 40, 50 ft. N of Microondas est. Sierra Lorenzo in igneous rocky slopes 29°09′ N 103°48′ W, alt 4500 ft., 13-III-1973, *J. Henrickson* and T. *Wendt 12312* (Holotype: LL00371149!).

**Distinguishing features**: It differs from the previous variety by its densely sericeous pubescence on the leaves, as well as by having smaller leaflets and its fruits having villous flattened, uncinulate at the apex setae.

**Selected examined material**: Coahuila: 17-X-1989 *D. Castillo 1090* (ANSM); 23-VIII-1984, *CDRI 1062* (TEX).

**Comments**: Endemic to northern Mexico. In the study area, recorded only in the state of Coahuila. Also distributed in Chihuahua and Durango, associated with desert thorn scrub in igneous rocky slopes; gravelly, calcareous loam, and limestone soils.

***Mimosa guaranitica*** Chodat and Hassler, Bull. Herb. Boissier II, 4 (6): 555. 1904.

**Type**: Paraguay, 1901, *É. Hassler 8326* (Isotype: F58365!); Paraguay, in valle Fluminis Y-aca, XII, *Hassler 6764* (Syntype: S13–12204!); Paraguay, near Venezuela, non-date, *E. Hassler 7009* (Isosyntype: UC934947!)

**Distinguishing features**: Unarmed herbaceous or sub-shrubs. Stems arising from a woody rootstock, viscid–villosulous with tiny, curved hairs along with some longer straight ones and some retrorse ones, and abundant gland-tipped setulae. Pinnae commonly 3 pairs per leaf. Leaflets 6–13 pairs per pinna, decrescent proximally, setulose-fimbriolate. Inflorescences pink, arranged in spheric capitula. Flowers 3-merous, rarely 4-merous. Stamens 6, filaments pink. Fruit 2–4.3 × 0.3–0.5 cm, linear, replum undulate, glabrous or densely puberulent with or without gland-tipped setulae, braking and dehiscent in individual articles.

**Selected examined material**: Tamaulipas: 27-X-1959, *M.C. Johnston 4544*, *J.G. Graham* (TEX-LL); TEX230343!; MICH37707!.

**Comments**: Rare species in Mexico, recorded only in two localities in Tamaulipas. Also in South America (Paraguay, Argentina, and Brazil).

***Mimosa latidens*** (Small) B.L. Turner, Phytologia 76: 414. 1994. Basionym: *Morongia latidens* Small. Bull. New Tork Bot. Gard. 2: 98. 1901. *Schrankia latidens* (Small) K. Schum., in Engler, Just’s Bot. Jahresber. 29 (1): 540. 1903. *Leptoglottis latidens* (Small) Small ex Britton and Rose, N. Amer. Fl. 23 (3): 142. 1928.

**Type**: USA, Texas, Karnes County, *Heller 1779* (Isotype: K791093!; MO-128321!; K791092!).

**Distinguishing features**: Herbaceous prostrate with thin stems. Pinnae 1–3 pairs per leaf, pinnae 2.5 cm long or shorter. Leaflets 5–8 pairs per pinna, up to 5.5 mm long, the midrib not evident. Petiole 1–2.5 cm, 2.5–3.5 times as long as the rachis leaf. Inflorescences arranged in spheric pink capitula. Fruit 1.7–5 × 0.3–0.4 cm, the replum 1.5–2.5 mm thick, the valves thorny, horns 2–3 mm long, straight, or curved.

**Selected examined material**: Coahuila: 28-VI-1982, *J.A. Villarreal 1613* (ANSM), 21-V-1980, *R. López 684* (ANSM), 14-VI-1987, *D. Castillo 563* (ANSM); Tamaulipas: 10-IX-1984, *J.L. Mora-López 498* (UAT); 23-V-1985, *D. Baro 753* (UAT).

**Comments**: Recorded in Coahuila and Tamaulipas, in low plains, 10–200 m. Morphologically very similar to *M. potosina* and *M. monclovensis* both, with the petiole 1.5–2.5 times as long as the rachis leaf, and neither of the two species present in Tamaulipas. Also in Texas. *Mimosa roemeriana* Scheele has been reported in Nuevo León and Tamaulipas; however, no specimens of this species have been found in national and foreign herbaria that appear in [82].

***Mimosa leucaenoides*** London J. Bot. 5: 89. 1846.

**Type**: Mexico, Hidalgo Zimapan, non-date, *J. Couter s.n.* (Holotype: K82487!).

**Distinguishing features**: Shrub, unarmed, 1–2.5 m tall. Branches puberulent or glabrous with resinous dots. Pinnae 1–2 pairs per leaf, leaflets 2–5 pairs per pinna, 1–4 × 0.6–3 cm, prominently reticulate on both faces, margins thickened with resinous dots and sparsely ciliated. Inflorescences solitary or in fascicles, arranged in spheric capitula. Flowers 4-merous, white. Stamens 8. Fruit 2.5–8.5 × 1–1.2 cm, straight, with abundant resinous dots, margin unarmed, breaking in several individual, indehiscent articles.

**Selected examined material**: Nuevo León–Tamaulipas border, 3-VIII-1972, *F. González Medrano* et al. *4200* (MEXU); 12-XI-1959, *J. Graham*, *M.C. Johnston* 4664 (MEXU); 23-XI-1967, *R.H. Magaña 7523* (MEXU); 10-VII-1983, *F. González Medrano 7256* (MEXU).

**Comments**: Endemic to Mexico. In Tamaulipan thorn scrub (360 m), piedmonts scrub (600–700 m), tropical deciduous forest (1200 m), and oak forest (1800 m). Outside the study area, it has been reported in San Luis Potosí, Guanajuato, Hidalgo, and Querétaro.

***Mimosa malacophylla*** A. Gray, Boston J. Nat. Hist. 6 (2): 182–183. 1850.

**Type**: Mexico, Nuevo León, “On the Rio Grande, Texas, Mr. Charles Wright, from Rinconada, near Monterey, Northern Mexico, 1848, *Gregg 207*! (Lectotype: GH65133!). Syntype: Mexico, Nuevo León, Valley near of Monterrey, 27-V-1854, *J. Gregg s.n.* (GH65134!)

**Distinguishing features**: Scandent or prostrate sub-shrub, 1–5 m long. Stems, branches, and petiole ribbed; armed with curved thorns along ribs. Pinnae 3–6 pairs per leaf. Leaflets 4–7 pairs per pinna. Inflorescences terminal or axillar in leaves in fascicles, arranged in spheric to ovoid capitula. Flowers 4–5-merous, white. Stamens 5–10. Fruit 4–9 × 0.8–1.2 cm, oblong, compressed, margin unarmed or rarely with few thorns, glabrous, laterally constricted between seeds, margin persistent; separating into 1-seeded, indehiscent articles.

**Selected examined material**: Coahuila: 28-IX-1999, *J.A. Villarreal 8918* (ANSM), 23-VI, 2010, *J.A. Encina 2742* (ANSM). Nuevo León: 5-VII-2001, *E. Estrada 12888*! (CFNL); 8-VI-2002, E. Estrada 14608! (CFNL); 28-IX-1982, *N.L. Bendek A. 164*! (TEX-LL); 22-VI-1937, *M.T. Edwards 311*! (TEX-LL). Tamaulipas: 29-IX-1983, *L. Hernández 720* (UAT).

**Comments**: In Tamaulipan thorn scrub, piedmont scrub, riparian communities, and oak–pine forest; occasionally as an invasive and aggressive weed in abandoned crop fields and along fences, 260–1025 m. Distributed also in south Texas.

***Mimosa martin delcampoi*** Medrano, Bol. Soc. Bot. Mexico 43: 40. 1983.

**Type**: Mexico, Tamaulipas; 7 km north of Magdaleno Aguilar (Santiaguillo). Municipio de Jaumave, 14-IX-1976, *F. Gonzalez Medrano*, *9818* (ENCB) (Isotype: K82485!).

**Distinguishing features: Shrub,** up to 2 m tall. Branches purple, glabrous. Pinnae 1 pair per leaf. Leaflets 2 pairs per pinna, 2–4 × 1.8–3 cm, leathery, dark green, prominent venation in both faces. Inflorescences leafless, arranged in spherical pink capitula. Flowers 4–5-merous. Stamens 10. Calyx campanulate up to 1 mm long. Corolla up to 3.6 mm long. Filaments pink. Fruit 2–5 × 0.9–1.2 cm, oblong, brown, membranous, separating into 1-seeded, indehiscent articles 5.5–7 mm long.

**Selected examined material**: Tamaulipas: 14-IX-1976, *F. Gonzalez Medrano*, *9818* (MEXU); 18-IX-1985, *R. Díaz 489* (ANSM), 16-VI-1984, *J. Valdés 107* (ANSM).

**Comments**: Endemic to Tamaulipas, recorded only in the municipalities of Bustamente and Palmillas. In Tamaulipan thorn scrub, limestone or gypsum soils, 1700–2200 m. In the type locality of *Mimosa martin delcampoi*, it coexists with the other related unarmed shrubby species such as *M. leucaenoides*, inhabiting arid scrublands on clayish soils.

***Mimosa monancistra*** Benth., Pl. Hartweg. 12. 1839.

**Type**: Mexico, Aguas Calientes, 1875, *Hartweg 70* (Holotype: K82495!; Isotype: E383742!; US890!; NY2580!).

**Distinguishing features: Shrub,** 0.5–2.5 m tall, branches in zigzag, all armed or only at a few nodes. Pinnae 2–5 pairs per leaf, rachis minutely appressed pubescent. Leaflets 4–9 pairs per pinna, densely appressed–pubescent or subglabrous. Inflorescences axillar, in racemiform branches, 8–25 cm long, solitary or in fascicles, arranged in pink spheric to ovoid capitula. Flowers 5-merous. Stamens 9–10, filaments pink. Fruit stipitate, 2–4.5 × 0.4–0.7 cm, oblong, curved, compressed between the seeds, articles 3–8, puberulent and setose.

**Selected examined material**: Nuevo León: 9-X-1959, *J. Graham* 4250 (TEX-LL); 12-XI-1959, *J. Graham 4650* (TEX-LL); 6-X-1988, *N. Reid s.n.* (CFNL); 15-XI-1990, *E. Estrada 1947* (CFNL); 22-VI-2003, E. Estrada 15759 (CFNL). Tamaulipas: 7-IX-1986, *E. Estrada 663* (CFNL).

**Comments**: Endemic to Mexico. In Tamaulipan thorn scrub, piedmont scrub, and desert scrublands at high plains, 280–1500 m. Also in Durango, San Luis Potosi, Jalisco, El Bajio Region, and the Transverse Neovolcanic Range areas.

***Mimosa monclovensis*** R. Grether and M.F. Simon, Phytoneuron 39: 1–3. 2018. Basionym: *Schrankia subinermis* S. Wats., Proc. Amer. Acad. Arts 17: 350. 1882. *Leptoglottis subinermis* (S. Wats.) Britton and Rose, N. Amer. Fl. 23 (3): 141. 1928. *Mimosa subinermis* (S. Wats.) B.L. Turner, Phytologia 76: 424. 1994.

**Type**: Mexico, Coahuila, mountains 24 miles N of Monclova, 1/6-IX-1880, *E. Palmer 302*! (holotype: GH 00063784!; isotypes: G00367736!).

*Mimosa quadrivalvis* L. var. *nelsonii* (Britton and Rose) Barneby, Mem. New York Bot. Gard. 65: 302. 1991. *Leptoglottis nelsonii* Britton and Rose, N. Amer. Fl. 23 (3): 142. 1928.

**Distinguishing features**: Prostrate herbaceous, up to 1 m long, branched from the base. Pinnae 3–4 pairs per leaf. Leaflets 9–14 pairs per pinna with reticulate veins not evident. Petioles of the larger leaves mainly 1.5–2.5 times as long as the leaf rachis. Inflorescences arranged in spheric capitula 1–1.2 cm diameter. Flowers 5-merous. Fruit sessile, 6–10 × 0.4–0.5 cm, long acute apically, peak 5–10 mm long.

**Selected examined material**: Coahuila: 6-IX-1890, *E. Palmer 302* (PH22629!; K000082477!)

**Comments**: Endemic to Coahuila. Recorded only in the arid mountains of the city of Monclova, associated with desert scrublands. Morphologically like *M. latidens* but with 6–9 pairs of leaflets per pinna, the capitula 1–1.8 cm in diameter, stipitate, 2.5–6 × 0.3–0.4 cm, apex rostrate, the rostrum 2–6 mm. Also, like *M. potosina*, but with 1–2 (rarely 3) pairs of pinnae per leaf, the leaflets apically rounded or obtuse, and shorter fruits, 3–5 cm long.

***Mimosa paucijuga*** (Britton and Rose) B.L. Turner, Phytologia 76 (5): 424. 1994. Basionym: *Leptoglottis paucijuga* Britton and Rose, N. Amer. Fl. 23: 139.1938.

**Type**: Mexico, Nuevo León, Monterrey, 1848, *Eaton* and *Edwards 17* (Isotype: K82476!).

**Distinguishing features**: Prostrate herbaceous, up to 2 m long, branched from the base. Stems 4–5 ribbed, glabrous, armed with abundant curved horns. Pinnae 1–4 pairs per leaf, rachis of longer leaves 0.5–1.3 cm long. Leaflets 5–10 pairs per pinna, oblong to oblanceolate, venation prominent on lower surface, midvein branched on or above middle portion. Inflorescences arranged in pink spheric capitula. Flowers 4–5-merous. Stamens 10, pink. Fruit ascending, tetragonal, 1.7–5 × 0.3–0.4 cm, straight or curved, armed with curved horns, apex acute, glabrous, or densely puberulent.

**Selected examined material**: Nuevo León: 1848, *Eaton* and *Edwards 17* (K000082476!); V-1911, *G. Arsené* and *Albon 6132* (MO-128322!).

**Comments**: Endemic to Mexico. Recorded in piedmont scrub on mountain slopes, on calcareous soils at or below 500 m. *M. paucijuga* is the only native herbaceous *Mimosa* with raised venation in leaflets. Remarkably like *M. latidens,* in which leaflet venation is not evident on the lower surface.

***Mimosa pigra*** ssp. ***asperata*** (L.) Zarucchi, Vincent and Gandhi, Phytoneuron, Syst. nat. ed. 10, 1312. 1759. Basionym: *Mimosa berlandieri* A. Gray in W.H. Emory, Rep. U.S. Mex. Bound. 2 (1): 61. 1859. *Mimosa asperata* var. *berlandieri* (A. Gray) B.L. Rob., Proc. Amer. Acad. Arts 33: 331. 1898. *Mimosa pigra* var. *berlandieri* (A. Gray) B.L. Turner, Field and Lab. 24: 1956.

**Type**: Mexico, Environs of Matamoras [Matamoros], 1839, *Berlandier 3146* (Lectotype: GH00065067!).

**Distinguishing features: Shrub,** 1–1.8 m tall. Stems armed intranodally, straight, laterally compressed, widened at the base; whitish horns, these also present in the intra-pinnal rachis. Young stems, petioles densely gray-puberulent, strigulous or hirsute. Pinnae 4–8 pairs per leaf. Leaflets 22–33 pairs per pinna. Inflorescences axillar, solitary or in fascicles. Flowers 4-merous, whitish. Stamens 8, filaments pink, white with age. Fruit 3–7.5 × 0.8–1.3 cm, replum straight; reddish, brown, or black; puberulent with sub-adpressed setae or erect; splitting into single-seeded, indehiscent mericarps.

**Selected examined material**: Nuevo León: 28-X-2009, *B. Soto* s.n. (CFNL). Tamaulipas: III-1989, *A. Brito 2778* (UAT); 27-V-1997, *A. Mora-Olivo 7085*, (UAT); 23-III-1999, *A. Mora-Olivo 7474a* (UAT); 18-II-1961, *R.M. King 3828* (TEX-LL); 14-IX-1967, *J. Rzedowski 24573* (TEX-LL); 19-VII-1994, *A. Mora-Olivo 5445* (UAT); 10-XI-1995, *A. Mora-Olivo 5635* (UAT).

**Comments**: Rare in northeastern Mexico, recorded only in areas of secondary vegetation surrounded by Tamaulipan thorn scrub, 560 m. From Texas (USA) along the Gulf Coast in Mexico to Tabasco, few records from Cuba and Nicaragua. *Mimosa pigra* ssp. *asperata* was treated by Barneby (1991) as distinct species, *M. pigra* and *M. asperata*; however, with the application of the International Code of Nomenclature for Algae, Fungi, and Plants (ICNAFP) [83], the species *asperata* is included as a variety of *M. pigra*.

***Mimosa potosina*** (Britton and Rose) B.L. Turner, Phytologia 76 (5): 424. 1994. Basionym: *Leptoglottis potosina* Britton and Rose, N. Amer. Fl. 23: 143. 1928.

**Type**: Mexico, San Luis Potosí, Minas de San Rafael, V-1911, *Purpus 5177* (Isotype: MO-128321!; GH65823!).

**Distinguishing features**: Herbaceous, prostrate. Pinnae 1–3 pairs per leaf. Leaflets mainly with obtuse to rounded apices, without evident reticulation veins. Petiole 1.5 to 2.5 times as long as the leaf rachis. Fruits 3–5 cm long.

**Selected examined material**: Nuevo León: 15-VI-1989, *E. Estrada* (CFNL, TEX-LL); 5-VIII-1936, *M. Taylor 65* (TEX-LL); 5-VIII-1970, *L.D. Flyr 1553* (TEX-LL).

**Comments**: Endemic to Mexico. Similar morphologically to *M. monclovensis* but this last species with 3–4 pairs of pinnae per leaf, the leaflets with acute apices, and longer fruits, 6–8 cm long.

***Mimosa pudica*** ssp. ***hispida*** Brennan, Kew Bull 1 (2): 186–187. 1955. Basionym: *Mimosa striato-stipula* Steud. Flora 26: 758. 1843. *Mimosa andreana* Britton and Rose in Britton and Killip, Ann, New York Acad. Sci. 35. 151: 1936. *M. pudica* forma *hispidior* Benth. 1875: 397, ex parte. *M. pudica* fma. *hispidior* Fawcett and Rendle, Fl. Jamaica 4: 134. 1920, ex parte. *M. pudica* var. *hispida* sensu Bassler, 1985: 606.

**Type**: Indonesia, Java. *Junghuhn 779* (holotype: K000791072!).

**Distinguishing features**: Prostrate herbaceous or sub-shrub, stems up to 1 m long, hispidulous, armed below or near the nodes with 1 pair of thorns and sometimes with additional internodal thorns. Pinnae 2 pairs, rarely 1 or 3 pairs per leaf. Leaflets 11–36 pairs per pinna, adaxially and abaxially sticky. Petiole and peduncles glabrous with setae up to 3.5 mm long. Inflorescences arranged in pink spheric capitula. Flowers always 4-merous Stamens 4. Filaments pink. Fruit 8–15 × 2.5–5 mm, oblong hispid, with brown setae, separating into single-seeded mericaps.

**Selected examined specimens**: Nuevo León: 4-VI-2010, *M. Garza L. s.n.* (CFNL).

**Comments**: Recorded cultivated in gardens in the center (Rayones Municipality) of Nuevo León. Also in southern Mexico, Antilles, Central America to South America (Colombia and Brazil). Used as ornamental in private gardens.

***Mimosa setuliseta*** Villarreal, Acta Botanica Mexicana 20: 45. 1992.

**Type**: Mexico, Durango, Municipio Lerdo, Sierra del Rosario, 40 km southwest of Cd. Lerdo, 25°25′ N, 103°45′ W, 1800 m, 9-XI-1990. *J.A. Villarreal 5790!* (Holotype: MEXU577147!; TEX371160!; NY444456!).

**Distinguishing features: Shrub,** up to 1.7 m tall. Young branches brown-purple, sericeous pubescent. Pinnae 3–5 pairs per leaf. Leaflets 3–5 pairs per pinna, glaucous, lighter abaxially. Inflorescences axillar, arranged in pink ovoid or short cylindric capitula. Flowers 5-merous, reddish-pink. Stamens 10. Fruit 4–5 × 0.1–1 cm, oblong, the valves covered by setae densely setose, separating into 1-seeded, indehiscent articles, margin smooth, not prickly.

**Selected examined material**: Tamaulipas: 3-VII-1985, *P. Hiriart 814*! (UAT); 26-IX-1984, *P. Diaz Pérez 72* (UAT).

**Comments**: Endemic to Mexico. Recorded outside the study area only in Tamaulipas. Also in Durango and San Luis Potosí [84]. Species closely related to *M. pringlei*; however, in this last species, the fruit setae are only villous and uncinulate, never setose.

***Mimosa strigillosa*** Torr. and A, Gray, Fl. N. Amer. 1: 399. 1840. Basionym: *Mimosa sabulicola* Chodat and Hassler, Bull. Herb. Boissier II, 4: 548. 1904. *Mimosa dolichocephala* var. *sabulicola* (Chodat and Hassler) Hassler, Repert. Spec. Nov. Regni Veg. 9: 6. 1910.

**Type**: Paraguay, Prope Concepcionin sabulosis insulae, Chaco-I, VIII- 1901–1902, *E. Hassler 7209* (G400059!).

**Distinguishing features**: Herbaceous, unarmed, rarely armed. Stems prostrate, compressed, sometimes rooting in the nodes, up to 0.7 m long, strigose, with appressed trichomes, sometimes with fan-shaped setae or setulose ciliolate. Pinnae 3–7 pairs per leaf. Leaflets sensitive, 8–21 pairs per pinna, decreasing in size at both ends of rachis. Inflorescences axillar, arranged in subcylindrical or ellipsoid capitula, sometimes wider than long. Flowers 4–5-merous, diplostemonous. Corolla strigulose–setulose at lobes outside. Stamens pink. Fruits craspedial, 0.8–1.7 × 5–8 mm, papery, bulging over each seed, hispid or strigose with sub appressed setae, the articles free-falling, indehiscent.

**Selected examined material**: Tamaulipas: 28-VI-1994, *A. Mora-Olivo 5381* (UAT); 10-IX-1994, *J.L. Mora-López 499* (UAT); 2-X-1984, *D. Baro 455a* et al. (UAT); 30-VI-1984, *D. Baro 279* (UAT).

**Comments**: Recorded only in the state of Tamaulipas. Wet and riparian places; 250–300 m. A species with bicentrical dispersion. In southeastern North America and northeastern Mexico (Tamaulipas) and South America (Paraguay, Uruguay, and Argentina). It is the only *Mimosa* species with herbaceous habit and its stems, leaf axes and peduncles invested with setae.

***Mimosa texana*** Small Smithsonian Contr. Knowl. 3 (5): 61. 1852. ssp. ***texana*.** Basionym: *Mimosa texana* Small, Bull. New York Bot. Gard. 2 (6): 99. 1901. *Mimosa borealis* var. *texana* A. Gray, Pl. Wright. 1: 61. 1852. *Mimosopsis wherryana* Britton in Britton and Rose, N. Amer. R. 23 (3): 177. 1928. *Mimosa wherryana* (Britton) Standl., Trop. Woods 34: 40. 1933. *Mimosa lindheimeri* A. Gray, Pl. wright. 2: 51. 1853.

**Type**: USA, Texas, hills near Austin, 1-I-1849, *Wright 159* (Lectotype: GH40800!): Isolectotype (GH40802!); syntype (USA, New Mexico, W. of Chiricahua mountains, 1851, *C. Wright s.n.* (NY2486!).

**Distinguishing features:** Shrub, 0.5–2 m tall. Stems in zigzag, armed at all or at some nodes with an infra-petiolar curved horns, sometimes accompanied by 1–2 smaller infra-stipular horns. Pinnae 1–4 pairs per leaf. Leaflets 3–7 pairs per pinna, glabrous. Inflorescences arranged in spheric capitula. Flowers 5-merous, white to light yellow. Stamens 10, pink and whitish. Fruit 1.8–4.5 × 0.4–0.6 cm, oblong, straight or curved, slightly constricted marginally, margin prickly or unarmed, dehiscent, glabrous, brown, reddish-brown, minutely pubescent when young.

**Selected examined material**: Coahuila: 19-V-1974, *J. Marroquín 2762* (ANSM), 3-VIII-1995, *M.A. Carranza 2214* (ANSM). Nuevo León: 2-V-1994, *G. Hinton* et al. *24067* (TEX-LL); 18-XI-2001, *E. Estrada 13284* (CFNL); 12-VIII-1988, *T.F. Patterson 6501* (TEX-LL); 13-IV-2003, *E. Estrada 15515* (CFNL). 8-VIII-1936, *M. Taylor 97* (TEX-LL); 30-X-2002, *E. Estrada 15212* (CFNL). Tamaulipas: 16-IV-1999, *A. Mora-Olivo 7563* (UAT); 23-III-1999, *A. Mora-Olivo 7470* (UAT); 27-V-1993, *J.L. Mora-López 316* (UAT) Seneg.

**Comments**: Piedmont scrub, oak forest, coniferous forests, gypsophilous grassland and halophytic communities, abundant in overgrazed areas, 480–2350 m. Also in the south of Texas (USA) and San Luis Potosí.

***Mimosa turneri*** Barneby, Brittonia 38:4. 1986

**Type**: USA, Texas. Presidio County, along Rio Grande 21 road miles (34 km) upstream from Lajitas, 30-V-1985, *R. Barneby 17970* (NY2504!).

**Distinguishing features:** Shrub, erect, up to 2 m tall, branches armed with straight or curved thorns. Pinnae 1–3 pairs per leaf. Leaflets 2–3 pairs per pinna. Petiole dorsoventrally flattened. Inflorescences arranged in pink spheric capitula. Flowers 5-merous. Stamens 10. Fruit 1.5–6 × 0.5–0.8 cm, linear–undulate, arched, unarmed or with few spines, glabrous, brown, bulging on each seed, breaking into individual dehiscent articles.

**Selected examined material**: Coahuila: 27-III-2012, *J.A. Encina 3125* (ANSM), 19-IV-2017, *J.A. Encina 5762* (ANSM). Nuevo León: 5-VII-1973, *M.C. Johnston*, *T.L. Wendt*, *F. Chiang 11614* (TEX-LL); 8-IX-2001, *E. Estrada 13067* et al. (CFNL);.

**Comments**: Recorded only in the Nuevo León and Coahuila geopolitical border, associated with desert scrublands in limestone or volcanic soils, 500–700 m. Also in Chihuahua and the south of Texas (USA).

***Mimosa unipinnata*** Parfitt and Pinkava, Brittonia 30: 172. 1978.

**Type**: Mexico. Coahuila: Cuatro Cienegas Basin, Sierra de San Marcos opposite Laguna Grande, 1000-1500 m, 14-VIII-1975, *Reeves* and *Pinkava P13073* (Holotype: ASU19172!; Isotype: (NY2624!). Paratype: (Mexico, Coahuila, top Sierra San Marcos, 8-VIII-1968, *W.L. Minckley* s.n., ASU19173!).

**Distinguishing features**: Shrub, 0.5–1.2 m tall. Stems armed, recurved infrapetiolar horns in most of internodes or immediately below several nodes with 1 stout, infra-petiolar thorn. Leaves unipinnate. Leaflets 1–3 pairs per leaf. Inflorescences arising in brachyblasts, solitary or in fascicles, arranged in pink spheric capitula. Flowers 4-merous. Corolla pink. Fruit 3.5–7.5 × 0.8–1 cm, narrow, oblong, smooth, bulging over each seed, replum with few prickles, separating into free-falling articles.

**Selected examined material**: Coahuila: 11-VI-2007, *J.A. Encina 2407* (ANSM), 16-VI-2008, *J.A. Encina 2572* (ANSM).

**Comments**: Endemic to central Coahuila (Cuatrociénegas municipality). *M. unipinnata* is the only species in the study area with unipinnate (not bipinnate) leaves. On foothills, in calcareous or gypseous soils, associated with desert scrub, 950–1450 m.

***Mimosa zygophylla*** Benth., Smithsonian Contr. Knowl. 3 (5): 61. 1852. Basionym: *Mimosopsis zygophylla* (A. Gray) Britton and Rose, N. Amer. Fl. 23 (3): 175. 1928.

**Type**: Mexico, Coahuila. La Vaqueria towards San Juan, 30 miles from Saltillo, Wislizenus 300, GH (GH65051!); Syntype (Mexico, 1848–1849, *J. Gregg 182* (GH65049); Isosyntype: Mexico, Coahuila, Buenavista, 22-V-1847, *J. Gregg s.n.* (NY2631!).

**Distinguishing features:** Shrub, 0.3–1.3 m tall; stems armed with abundant internodal thorns. Pinnae 1 per leaf. Leaflets 1 pair per pinna, rarely 2 pairs. Inflorescences arranged in pink spheric capitula. Flowers 4-merous. Corolla, the lobes reddish. Stamens 6–8, filaments pink. Fruit 1.8–6.5 × 0.4–0.7 cm, oblong, mahogany brown, glabrous; margin generally unarmed or armed with small, curved horns.

**Selected examined material**: Coahuila: 9-V-1987, *J.A. Villarreal 3705* (ANSM), 16-VII-1993, *J. Valdés 1831* (ANSM), 13 V 1988, *J. Valdés 1840* (ANSM). Nuevo León: 29, 9-IX-1971, *J. Henrickson 6598* (TEX-LL); 1-VI-1997, *G. Hinton* et al. *27066* (TEX-LL); 18-X-2001, *E. Estrada 13154* (CFNL). 24-V-1973, *M.C. Johnston*, *T.L. Wendt*, *F. Chiang 11200* (TEX-LL); *E. Estrada 15765* (CFNL).

**Comments**: Endemic to the north of Mexico (Chihuahuan Desert), recorded only in Coahuila and Nuevo León, in desert scrub and halophytic vegetation, 660–1980 m. Outside the study area, in arid areas of Durango, San Luis Potosí, and Zacatecas.

***Neltuma* Raf., Sylva Tellur.**: **119. 1838.**

Type species: *Neltuma juliflora* (Sw.) Raf., Sylva Tellur. 119 (1838).

Trees or shrubs, erect to prostrate. Prickles solitary or paired, uni-nodal, axillar, straight, hard, cylindrical, subulate, not always present at all nodes. Brachyblasts congested, blackish. Stipules triangular and dry. Pinnae 1–3 pairs per leaf. Leaflets up to 40 pairs per pinna. Inflorescences axillary, solitary or fascicled, arranged in cylindrical spikes. Flowers, 5-merous, white, yellow to greenish-yellow, fragrant. Petals almost free. Stamens 10, free, the style exerted, anthers with a minute claviform gland arising from the connective. Fruits linear or compressed, turgid, sometimes red-tinged, indehiscent, glabrous, straight to slightly curved, smooth, exocarp hard, mesocarp pulpy, dry, frequently sweet, segmented in subquadrate closed seed chambers. Seeds brown, compressed.

A genus of 43 described species [85] in arid areas of North and South America.
1A.Leaflets (15-) 20–30 (-44) pairs per pinna, 3–15 × 1–1.3 mm, 2–7 times as long as wide, separated by intervals less than their width21B.Leaflets (7-) 8–24 pairs per pinna, 15–65 × 1.5–6 mm, 5–15 times as long as wide, spaced 5–18 mm apart42A.Pinnae 2.5–4 cm long; leaflets up to 7 mm long; fruits 6–8 mm wide***N. palmeri***2B.Pinnae 3–13 cm long; leaflets 5–30 mm long33A.Leaflets 5–10 mm long, not separated by more than their own wide; fruit 7–14 mm wide***N. laevigata***3B.Leaflets 10–30 mm long, separated by more than their own wide; fruit 10–18 mm wide***N. chilensis*** ssp. ***chilensis***4A.Leaflets 7–17 pairs per pinna, 2–6.5 cm long, linear–oblong, 5–15 times as long as broad***N. glandulosa*** ssp. ***glandulosa***4B.Leaflets 8–24 pairs per pinna, 1–2.5 cm long, oblong, 5–9 times as long as broad***N. odorata***

***Neltuma chilensis*** (Molina) C.E. Hughes and G.P. Lewis, Phytokeys 205: 174. 2022. ssp. ***chilensis***. Basionym: *Acacia siliquastrum* Cav. ex Lag., Gen. Sp. Pl. 16: 1816. *Prosopis siliquastrum* (Cav. ex Lag.) DC., Prodr. 2: 447. 1825. *Prosopis siliquosa* St.-Lag., Ann. Soc. Bot. Lyon 7: 132. 1880, orth. var. *Prosopis schinopoma* Stuck., Bull. Acad. Int. Géogr. Bot. 13: 87. 1904. *Prosopis chilensis* (Molina) Stuntz, Invent. Seeds PI. Import., U. S. Dept. Agr. Bur. PI. Industr. 31: 85. 1914.

**Type**: Argentina, Córdoba. Cruz del Eje. Sierra de Córdoba. 30°42′51′′ S 65°3′35′′ W, 12-XI-1898, *T. J. V. Stuckert 4911* (Syntype: CORD2946!)

**Distinguishing features**: Tree, up to 6 m tall (in our area). Young branches flexuose, knotty. Spines axillar, geminate, uninodal, straight, up to 4 cm long. Pinnae 1–3 pairs per leaf. Leaflets 10–30 pairs per pinna, linear, glabrate or marginally ciliate, 1–3 cm × 0.1–3 cm., light green, only the midvein evident. Inflorescences axillar, arranged in dense spiciform racemes. Flowers green-white to yellowish. Fruit 10–17 × 1–1.8 cm linear, compressed, segments transverse rectangular, broader than long.

**Selected examined material**: Coahuila: 15-XII-2023, *J. Sánchez-Salas 510* (UJED).

**Comments**: Native to South America (Argentina, Peru, Bolivia, and Chile) introduced as an ornamental species in private colony parks in Torreon, Coahuila. Rare.

***Neltuma glandulosa*** (Torr.) Britton and Rose ssp. ***glandulosa*.** Basionym: *Algarobia glandulosa* (Torrey) Torrey et A. Gray, Fl. N. Amer. 1: 399. 1840. *Prosopis juliflora* (Sw.) DC. var. *glandulosa* (Torrey) Cockerell. New Mexico. Agric. Expt. Sta. Bull. 15: 58. 1895. *Prosopis juliflora* var. *constricta* Sarg., Trees and Shrubs 2: 249. 1913. *Neltuma constricta* (Sarg.) Britton and Rose, in N.L. Britton and al. (eds.), N. Amer. Fl. 23: 186. 1928. *Neltuma neomexicana* Britton, in N.L. Britton and al. (eds.), N. Amer. Fl. 23: 186. 1928. *Prosopis bonplanda* P.R. Earl and Lux. Publ. Biol. FCB/UANL. Mex. 5 (2): 38. 1991.

**Type**: U.S.A., *C.S.* 27-III-1908, *Sargent s.n. James* (Syntype: A3464!)

**Distinguishing features**: Tree or shrub, up to 10 m tall, deciduous. Branches armed with nodal spines, usually solitary, sometimes solitary and geminate alternately at different nodes on the same branch. Stipules subulate. Pinnae 1–2 pairs per leaf. Leaflets 7–17 pairs per pinna, 2–6.5 × 0.1–0.5 cm, oblong, glabrous, subcoriaceous, prominently veined abaxially, mucronate at the apex, 7–18 mm apart. Fruit 8–25 × 0.4–1.3 cm, flattened, narrow, compressed to subcylindrical, slightly constricted between seeds, yellow-green, with longitudinal violet or purple striations.

**Selected examined material**: Coahuila: 26-III-1992, *M.A. Carranza 1314* (ANSM), 27-VI-2011, *J.A. Encina 2957* (ANSM), 17-VIII-2011 (ANSM). Nuevo León: 8-IV-2001, *E. Estrada 11939* (CFNL); 7-IV-2001, *E. Estrada 11841* (CFNL). 24-IV-1960, *M.C. Johnston* and *J. Crutchfield 5320* (TEX-LL). 5-VI-1988, *Damas y Canul 148* (TEX-LL). 17-VI-2000, *G. Hinton* et al. *27560* (TEX-LL); 20-VII-2002, *E. Estrada 14983* (CFNL); 17-IV-2001, *E. Estrada 12057* (CFNL). Tamaulipas: 20-XI-1987, *M. Martínez 1607* (UAT); 30-IV-1986, *R. Jones 104* (UAT); 23-III-1999, *A. Mora-Olivo 7471* (UAT); 3-X-1984, *S. Rodriguez 185* (UAT).

**Comments**: Common in areas of Tamaulipan thorn scrub, piedmont scrub, and arid shrublands in the High Plains of northeastern Mexico, more frequent in the northern region of Nuevo León and Tamaulipas, forming extensive mesquitales. Sometimes associated with halophytic vegetation. Predominant species of secondary vegetation in the area, abundant in overgrazed areas and fields of abandoned crop, 140–2000 m. Also in southern USA. The fruits are used as fodder for cattle. One of the most important timber species in lowlands of northeastern Mexico, being predominant in most landscapes of northeastern Mexico, especially those with signs of disturbance, its wood is widely used for construction, floors, charcoal, and handicrafts [73,75]. ***Neltuma glandulosa*** ssp. ***prostrata*** (Burkart) C.H. Hughes and G.P. Lewis (Type: U.S.A. Texas, Kleberg County, western part of Laureles Division of King Ranch, 15-IV-1954, *M.C. Johnston 54359* (Isotype: SI015053!). Shrub, up to 1.3 m tall, commonly prostrate, rarely erect or sub-erect. Branches with solitary spines. Leaflets 14–34 per leaf, 2–6 cm long, 0.8–1.5 cm apart, the apex obtuse. [86] mentions it being present in the state of Tamaulipas, in Tamaulipan thorn scrub; however, no specimen of this variety has been found in any of the national herbaria (ANSM, CFNL, MEXU, VIC) nor in the SEINET database [83]), so its presence in Mexico remains in doubt.

***Neltuma laevigata*** (Humb. and Bonpl. ex Willd.) Britton and Rose, in N.L. Britton and al. (eds.), N. Amer. Fl. 23: 187. 1928. Basionym: *Acacia laevigata* Humb. and Bonpl. ex Willd., Sp. Pl., ed. 4, 4: 1059. 1806. Basionym: *Prosopis laevigata* (Humb. and Bonpl. ex Willd.) M.C. Johnst., Brittonia 14: 78. 1962. *Prosopis dulcis* Kunth, Mimoses: 110. 1822. *Acacia tortuosa* Billb. ex Beurl., Kongl. Svenska Vetensk. Acad. Handl., n.s., 2: 24. 1856, nom. illeg. *Mimosa rotundata* Sessé and Moc., Pl. Nov. Hisp.: 178. 1890. *Neltuma michoacana* Britton and Rose, in N.L. Britton and al. (eds.), N. Amer. Fl. 23: 187. 1928.

**Distinguishing features**: Shrub or tree, 4–8 m tall. Spines paired, nodal, straight. Pinnae 1–2 pairs per leaf, 3–12 cm long, pinnae, 2.5–12 cm long. Leaflets 12–30 pairs per pinna, linear–oblong, 5–10 mm long. Gland circular, inserted in the proximal pair of pinnae or only in the distal pair of pinnae. Fruit 8–17 × 0.7–1.4 cm, linear–oblong, slightly constricted between seeds, the segments rounded or rectangular, shorter than wide, yellow, sometimes with longitudinal reddish ribs.

**Selected examined material**: Coahuila: 11-X-2005, *J.A. Alba 291* (ANSM). Nuevo León: 21-VI-2003, *E. Estrada 15774*! (CFNL); 11-VI-1992, *R. Palacios 2332*! (TEX-LL); 8-VI-2003, *E. Estrada 15748!* (CFNL); 22-III-2003, *E. Estrada 15348*! (CFNL). Tamaulipas: 13-VIII-1992, *J.L. Mora-López 163* (UAT); 16-VII-1987, *L. Hernández 2155* (UAT); 29-VII-1985, *D. Méndez 95* (UAT); 23-III-1985, *R. Diaz 313* (UAT).

**Comments**: Common in the Northern Gulf Coastal Plain (300–560 m) in Nuevo León and Tamaulipas, commonly forming pure dense forests; almost absent in the Sierra Madre Oriental physiographic province and reappearing in areas adjacent to the High Plains (1400–1800 m) in Coahuila and Durango, Zacatecas, Aguascalientes, Querétaro, and Hidalgo to Michoacán and Morelos. Associated with disturbed vegetation, arid and semiarid scrublands, deciduous forest, chaparral, and halophytic grasslands; rare in southern Texas (USA). Widely used as timber species in lowlands of the region, frequent in areas with disturbance; the wood is used in construction, fuel, and charcoal [73].

***Neltuma odorata*** (Torr. and Frém.) C.E. Hughes and G.P. Lewis. PhytoKeys 205: 182. 2022. Basionym: *Prosopis odorata* Toor. and Frem. Rep. Exped. Rocky Mts. 331. Pl. 1. 1845. *Strombocarpa odorata* (Torr. and Frém.) A. Gray, U.S. Expl. Exped., Phan. 1: 475. 1854. *Prosopis juliflora* var. *torreyana* L.D. Benson, Amer. J. Bot. 28: 751. 1941. *Prosopis glandulosa* var. *torreyana* (L.D. Benson) M.C. Johnst., Brittonia 14: 82. 1962. *Prosopis glandulosa* subsp. *torreyana* (L.D. Benson) A.E. Murray, Kalmia 12: 23. 1982.

**Type**: USA, California, Santa Fe, a few blocks of Santa Fe Depot, Needles, 6-VII-1941, *L. Benson 11000* (Isotype: RSA3615!; US984!; NY5125!; ARIZ-BOT-4108!)

**Distinguishing features**: Tree or shrub, up to 12 m tall, deciduous, branches with nodal spines, solitary, and/or alternately geminate at different nodes on the same branch. Pinnae 1 pair per leaf, rarely 2 pairs. Leaflets 8–24 pairs per pinna, 0.7–2.5 × 0.15–0.6 cm, linear, glabrous, prominently veined abaxially, 5–8 mm apart. Fruit 7–18 × 0.7–1.1 cm, subcylindrical, slightly constricted between seeds, yellow-green, with longitudinal lines tinged with violet or purple.

**Selected examined material**: Coahuila: 8-VII-1992, *R. Palacios 2302* (MEXU); 8–9-VI-1992, *R. Palacios 2311*, *2313* (MEXU); 9-VI-1992, *R. Palacios 2316* (MEXU). Nuevo León: 19-VII-2002, *E. Estrada 14953* (CFNL); 19-VII-2002, *E. Estrada 14964* (CFNL); 21-VI-2003, *E. Estrada 15774* (CFNL); 31-V-2003, *E. Estrada 15671* (CFNL); 2-VIII-1999, *E. Estrada 10520* (CFNL); 5-VI-1988, *Damas y Canul 124* (TEX-LL). Tamaulipas: 7-VI-1986, *D. Baro 812* (UAT).

**Comments**: Common in northeastern Mexico, associated with Tamaulipan and piedmont scrub communities, in highlands associated with desert scrublands, and in open oak and oak–pine forest, 195–2400 m. From Baja California to Tamaulipas and Zacatecas. Also in SW and SE of the USA. Its hard and resistant wood is widely used for the construction of houses, roofs, floors, and hand tools. Its leaves and fruits are used as fodder for domestic livestock [74,76].

***Neltuma palmeri*** Britton and Rose, in N.L. Britton and al. (eds.), N. Amer. Fl. 23: 185. 1928. Basionym: *Prosopis tamaulipana* Burkart, J. Arnold Arbor. 57: 494. 1976.

**Type**. Mexico. Tamaulipas: vicinity of Victoria, *E. Palmer 400* (GH64599!).

**Distinguishing features**: Shrub or tree, up to 7 m tall. Spines paired, axillary, not present at all nodes. Pinnae 1–3 pairs per leaf. Leaflets 15–30 pairs per pinna, 2.8–7 mm long, reticulate-veined abaxially. Fruit, 8–13 × 0.6–0.8 cm, linear–oblong, straight, sub-moniliform, indehiscent, yellow, sometimes with reddish longitudinal striations, mesocarp pulpy, sweet.

**Selected examined material**: Nuevo León: *C. Hughes 689* (MEXU). Tamaulipas: non-date, *E. Estrada 395* (MEXU); 3-VI-1988, *Damas y Camul 105* (MEXU); 5-V-1988, *Damas y Camul 125* (MEXU); 11-VI-1992, *Palacios 2348* (CFNL); 9-II-1999, *A. Mora-Olivo 7398* (UAT); 8-VI-1998, *A. Mora-Olivo 6994* (UAT); 28-IV-1960, *M.C. Johnston 5375* (TEX-LL).

**Comments**: Endemic to northeastern Mexico. In Tamaulipan thorn scrub, in shallow soils, 250–360 m. Also found in the north of Veracruz and in southeastern San Luis Potosí in the political border adjacent to Tamaulipas.

***Neptunia*** Lour., Fl. Cochinch. 2: 653. 1790.

**Type**: *N. oleracea* Lour. Lour., Fl. Cochinch. 2: 654. 1790.

Herbaceous, perennial, prostrate to ascending. Stems cylindrical, angled when young. Stipules paired; lanceolate; basally oblique; rarely foliaceous, caducous, or persistent. Leaves bipinnate, frequently sensitive. Petiole with or without glands. Inflorescences axillar, arranged in dense spikes. Peduncles with a pair of bracts or absent. Flowers 5-merous, dimorphic, the upper ones perfect, the lower ones staminate with the stamens sterile. Calyx campanulate, 5-merous, green or yellow. Corolla 5-merous, green or yellow. Stamens 5 or 10, the anthers bearing a small gland apically. Sterile stamens 5 or 10, petaloid. Fruit flattened, marginally dehiscent, the valves membranous to coriaceous. Seeds 1–20, oblique or transversely arranged.

Genus of 11 [87] or 12 species [88] of tropical and warm regions. In northeastern Mexico, three species present, *N. microcarpa*, *N. pubescens* and, *N. prostrata*. The first two are disjunct, with different geographical range and ecology [88]. Both of these species were treated as conspecific [87].
1A.Peduncles with conspicuous bracts subtending the spike arising in the lower half, or the bracts absent; inflorescences in globose or obovoid spikes in bud***N. prostrata***1B.Peduncles with small lanceolate bracts arising in the upper half; spike cylindrical or ellipsoid in bud22A.Fruit stipe longer than persistent calyx; fruit usually narrowing at contact with stipe; pinnae 3–6 pairs per leaf***N. pubescens***2B.Fruit stipe usually shorter than persistent calyx; fruit usually rounded at contact at stipe; pinnae 2–3 pairs per leaf***N. microcarpa***

***Neptunia microcarpa*** Rose, Contr. U.S. Nat. Herb. 8: 300. 1905. Basionym: *Neptunia palmeri* Britton and Rose, N.L. Britton and al. (eds.), N. Amer. Fl. 23: 182. 1928. *Neptunia pubescens* var. *microcarpa* (Rose) Windler, Austral. J. Bot.14: 393. 1966.

**Type**: Mexico, near Guadalajara, 5-VIII-1902, *C.G. Pringle 8626* (Holotype: US976!)

**Distinguishing features**: Herbaceous, perennial, prostrate, up to 2 m long. Pinnae 2–3 pairs per leaf. Leaflets 14–43 pairs per pinna, oblong, minutely punctate abaxially, reticulate-veined. Inflorescences in dense ellipsoid spikes. Peduncles with 2 bracts arising in the upper middle. Stamens 10, free, anthers with a terminal gland. Fruit stipitate, stipe shorter than calyx, the body 1–4 × 0.5–1.6 cm, oblong, flattened, membranous–coriaceous, dehiscent.

**Selected examined material**: Coahuila: II-X-1880, *Palmer 300* (US44881!). Nuevo León: 7-V-1938, 3-V-1986, *E. Estrada* 396 (MEXU); *Cottam 10582* (US1337656!); 28-IX-1983, *O. Briones y N.L. Bendek s.n.* (TEX-LL). Tamaulipas: not date, *Stern*, *Barkley*, and *Rowell 25* (TEX); 23-VIII-2008, *A. Mora-Olivo 11760* (UAT); 14-XI-1998, *A. Mora-Olivo 7056* (UAT); 16-VI-1998, *A. Mora-Olivo 6960* (UAT).

**Comments**: Rare in the area, in Tamaulipan thorn scrub. Also in Jalisco and Texas, USA.

***Neptunia prostrata*** (Lam.) Baill., Bull. Mens. Soc. Linn. Paris 1: 356. 1883. Basionym: *Neptunia oleracea* Lour. Fl. Conchinch. 654. 1790. *Aeschynomene pumila* L., Sp. Pl. ed. 2: 1061. 1763. *Mimosa natans* L.f., Suppl. Pl.: 439. 1782. *Mimosa prostrata* Lam., Encycl. 1: 10 1783. *Neptunia natans* (L.f.) Druce, Rep. Bot. Soc. Exch. Club Brit. Isles 1916: 637. 1917. *Desmanthus lacustris* (Bonpl.) Willd., Sp. Pl., ed. 4, 4: 1044. 1806. *Desmanthus natans* Willd., Sp. Pl., ed. 4, 4: 1044. 1806. *Mimosa aquatica* Pers., Syn. Pl. 2: 263. 1806. *Desmanthus stolonifer* DC., Prodr. 2: 444. 1825. *Mimosa stolonifera* Perr. ex DC. Prodr. 2: 444. 1825. *Aeschynomene herbacea* Aubl., Hist. Pl. Guiane 2: 775. 1775. *Acacia lacustris* Desf. Tabl. École Bot., ed. 3: 301. 1829. *Mimosa lacustris* Bonpl. F.W.H.A.von Humboldt and A.J.A. Bonpland, Pl. Aequinoct.1: 55. 1806. *Neptunia stolonifera* Guill. and Perr., Fl. Seneg. Tent.: 239. 1832.

**Type**: Unknown Country, *Bonpland s.n.* (Isotype: P02436151!).

**Distinguishing features**: Herbaceous, aquatic, floating or riparian near water edges and prostrate. Stems 1–1.5 m long. Petiolar gland present or absent. Pinnae 2–4 pairs per leaf, the pinnae rachis winged, extended beyond the attachment of the distal pair of leaflets. Leaflets 8–20 pairs per pinna. Inflorescences axillar, arranged in globose or obovoid spikes in bud, peduncles with 2 conspicuous bracts subtending the spike arising in the lower half, rarely absent. Fruit stipitate, the stipe 0.4–0.9 mm long, protruding from the calyx, the body 1.8–3 × 0.8–1 cm, broadly oblong, flattened, membranous–coriaceous, glabrous, with the body usually at a 90° angle to the stipe.

**Selected examined material**: Tamaulipas: 24-VIII-1986, *A. Novelo 820*, *M. Martínez* (UAT).

**Comments**: Recorded only in the state of Tamaulipas, aquatic and floating species. Widely distributed in stagnant water and humid habitats, in Asia, Africa, Central America, and South America [87].

***Neptunia pubescens*** Benth., J. Bot. (Hooker) 4 (31): 356. 1841. Basionym: *Neptunia floridana* Small, Bull. Torrey Bot. Club 25: 138. 1898. *Neptunia pubescens* var. *floridana* (Small) B.L. Turner, Amer. Midl. Naturalist 46: 89. 1951. *Neptunia lindheimeri* B.L., Rob. Proc. Amer. Acad. Arts 33: 333. 1898. *Neptunia pubescens* var. *lindheimeri* (B.L. Rob). B.L. Turner, Amer. Midl. Naturalist 46: 88. 1951.

**Type**: United States, 1-VI-1843, *F.I.X. Rugel*, *s.n.* (Syntype: MEL256479!).

**Distinguishing features**: With similar morphology to the previous variety, but with 3–6 pairs of pinnae per leaf, and the fruit stipe longer than calyx.

**Selected examined material**: Tamaulipas: 23-IV-2009, *E. Estrada 20771* (CFNL); 27-X-1959, *Graham* and *Johnston 4508* (230811TEX!)**.**

**Comments**: In subtropical regions, in open areas. Recorded only in the state of Tamaulipas. Outside of the area, from southeastern USA through Mexico (Sinaloa, Michoacán, Veracruz, and Chiapas) to Argentina.

***Strombocarpa*** (Benth.) Engelm. and A. Gray, Boston J. Nat. Hist. 5: 243. 1845.

**Type species**: *Prosopis strombulifera* Benth., J. Bot. (Hooker) 4: 352. 1841.

Small creeping shrubs, up to 50 cm tall, multicauled from the base. Stipules spinescent, paired. Leaves bipinnate, pinnae 1 pair per leaf. Inflorescences arranged in spheric capitula, solitary. Flowers 5-merous, light yellow. Petals linear, partially united, pubescent within. Stamens and style exerted; anthers with a minute, caducous, claviform gland inserted in the connective. Fruit tightly spirally coiled, forming a cylindrical body. Seeds ovate or reniform ovoid.

An American genus constituting 10 species [81], amphitropical, in arid regions of the Sonoran Desert (USA and Mexico), northeastern Mexico and South America (Peru, Argentina and Chile). Only a single species recorded in Mexico, *Strombocarpa cinerascens*.

***Strombocarpa cinerascens*** A. Gray, Smithsonian Contr. Knowl. 3, part 5: 61. 1852. Basionym: *Prosopis cinerascens* (A. Gray) A. Gray ex Benth., Trans. Linn. Soc. London 30 (3): 381. 1875. *Prosopis reptans* var. *cinerascens* (A. Gray) Burkart, Darwiniana 4: 75. 1940.

**Type**: Mexico, valley near Azufrosa, 22-IX-1848, *J. Gregg 492* (Holotype: GH3469!).

**Distinguishing features**: Sub-shrub, 20–40 cm tall, rhizomatous, forming colonies. Branches with nodal spines, straight, paired, white, up to 3 cm long. Pinnae 1 pair per leaf, leaflets 7–13 pairs per pinna, slightly pubescent on lower surface and margins. Inflorescences axillar, arranged spheric capitula. Fruit coiled in a spiral (6–12 turns), cylindrical, yellow to light brown, woody.

**Representative examined material**: Coahuila: 3-VI-1966, *J.S. Wilson 11482* (TEX-LL); 12-V-1977, *J. Henrickson* and *D.H. Riskind 16055* (TEX-LL); 5-VII-1973, *R. Palacios 11612-A* (TEX-LL). Nuevo León: 13-VII-2002, *E. Estrada 14816* (CFNL); 23-VI-2001, *E. Estrada 12783b* (CFNL); 5-VII-1973, *T.L. Wendt y F. Chiang 11612A* (TEX-LL); 1-VI-1992, *R.A. Palacios 2301* (TEX-LL); 20-III-2023, *E. Estrada 26079* (CFNL). Tamaulipas: 28-VI-2002, *E. Estrada 14760* (CFNL); 29-V-1962, *L. Garza s.n.* (TEX-LL); 21-IV-1962, *M. Dominguez L*, *8296* (TEX-LL).

**Comments**: On low plains in northeastern Mexico (Nuevo León and Tamaulipas) and south of Texas (USA), associated to Tamaulipan thorn scrub and halophytic vegetation, in clay–silty and saline soils flooded for prolonged periods of time, 360–520 m. Also in San Luis Potosí.

**Tribe *Ingeae*** Benth. and Hook.f., Gen. Pl. [Benth. and Hooker f.] 1 (2): 437. 1865.

Trees, shrubs, or sub-shrubs. Stems and branches with or without prickles. Stipules inconspicuous or evident, spinescent. Leaves pinnate or bipinnate Pinnae 1–6 pairs per leaf. Leaflets 1–24 pairs per pinna. Petiole, rachis, and pinnae rachis with 1 or more glands, rarely absent. Inflorescences arranged in spheric, sub-spheric, ovoid capitula, or congested spikes or panicles; when heteromorphic, the central ones differentiated and showing long perianth and long exerted stamens. Calyx 4–8-merous, gamosepalous, valvate. Corolla 4–8-merous, gamopetalous, valvate. Stamens numerous, always basally united, forming a short or long tube. Ovaries 1 to many, free. Fruit linear; oblong; frequently flattened, straight, spirally contorted, curved, or coiled; sometimes moniliform; segmented internally or not, chartaceous, or coriaceous, elastically dehiscent, or not or indehiscent.

Tribe conformed for about 36 genera and 935–966 species [2]; inhabiting mostly tropical and subtropical regions.
1A.Plants unarmed21B.Plants thorny82A.Leaves pinnate, paripinnate, the rachis ending in a pair of leaflets; fruit indehiscent***Inga***2B.Leaves bipinnate; fruit dehiscent33A.Petiole without glands (nectaries)43B.Petiole with glands between or below the pair of proximal pinnae54A.Stigma cup-shaped, slightly wider than style; 16-grain polyads with eccentric, lentil-shaped thickenings in the central cells***Zapoteca***4B.Stigma discoid or capitate, 3 times wider than the style; 8-grain polyads, 6 peripherals and 2 centrals***Calliandra***5A.Fruit valves separating from persistent margin; flowers in capitula or spikes***Lysiloma***5B.Fruit valves not separating from persistent margin; flowers in capitula 66A.Fruit subcylindrical or moniliform, slightly or strongly falcate and frequently coiled ***Cojoba***6B.Fruit flattened or compressed but plump, 2–10 cm wide, straight or coiled 77A.Fruit thin, straight, exocarp flexible, 3–5 cm wide***Albizia***7BFruit, circular or semicircular, coiled, broadly oblong, 5–7 cm wide; stamen filaments white to yellow-white ***Enterolobium***8A.Leaves with a pair of pinnae, each pinna with a pair of leaflets; seeds with fleshy aril***Pithecellobium***8B.Leaves with 1 or more pairs of pinnae, each pinna with 2 or more pairs of leaflets99A.Fruits flattened, straight, dehiscent at both sutures, without septa; leaves with 4 or more pairs of pinnae***Havardia***9B.Fruit subcylindrical, cylindrical, or torulous; straight or curved; leaves with 1–3 pairs of pinnae1010A.Petiolar gland sessile, superficially cupular; ovary stipe 1 mm long or longer; fruit leathery, curved, flattened, unilocular***Painteria***10B.Petiolar gland stipitate, sub-columnar or columniform; ovary stipe 1 mm long or shorter; woody fruit, straight or slightly curved, internally divided into interseminal spaces***Ebenopsis***

***Albizia*** Durazzini, Mag. Tosc. 3 (4): 10. 1772.

**Type species**: *A. julibrissin* Mag. Tosc. 3 (4): 13. 1772.

Trees or shrubs, unarmed. Leaves bipinnate, pinnae 1 to several pairs per leaf. Leaflets 3–63 pairs per pinna. Petiole frequently with a gland, and 1 or more glands on the rachis, at pinnae insertion. Inflorescences solitary or in fascicles, arranged in spikes or umbels, the flowers arranged in spheric capitula. Flowers 4–7-merous, commonly bisexual, rarely unisexual, with a different appearance from the rest. Calyx campanulate to tubular. Corolla cylindrical–tubular, lobed. Stamens 10–46, their filaments united basally forming a tube. Fruit sessile or short stipitate, oblong to broad linear, compressed, straight, indehiscent, or dehiscent into 2 valves. Seeds arranged transversely.

A genus with approx. 140 species [2], 19 of which are native to America. With pantropical distribution, highly diversified in American tropics, Africa, Asia, and Malaysia [89]. Some species are used as ornamentals. Four native species have been reported for Mexico [89]: *A. tomentosa* (M. Micheli) Standl., *A. sinaloensis* Britton and Rose, *A. adinocephala* (Donell Smith) Briton and Rose, *A. niopoides* (Benth.) Burkart var. *niopoides* as well as one introduced, *A. lebbeck* (L.) Benth., the only species recorded for northeastern Mexico.

***Albizia lebbeck*** (L.) Benth., London J. Bot. 3: 87. 1844. Basionym: *Mimosa lebbeck* L., Sp. Pl. 516: 1753.

**Type**: China, 1855, s. coll. s.n. (Isotype: K800903!; K800904!);

**Distinguishing features**: Tree up to 7 m tall, cultivated. Leaves bipinnate. Pinnae 2–4 pairs per leaf. Leaflets 2–11 pairs per pinna. Petioles with glands arising in the base and between the insertion of most pairs of leaflets. Inflorescences solitary or in fascicles, arranged in spherical or hemispherical capitula. Flowers 5–6-merous, white, dimorphic, pedicellate (peripheral ones) or sessile (terminal ones). Stamens 22–52, free above the corolla. Fruit 13–26 × 3–5 cm, oblong, flattened, straight, glabrate, light brown or straw when mature, papery, leathery, tardily dehiscent by the ventral suture only. Seeds transversely arranged.

**Representative examined material**: Tamaulipas: 24-IV-2009, *E. Estrada 20761* (CFNL); 30-III-2003, *E. Estrada 15396* (CFNL); 28-IX-2006, *E. Estrada 19968* (CFNL); 26-V-1983, *G. Villegas 566* (UAT).

**Comments**: Species introduced and cultivated as ornamental. Native to the eastern tropical region of Asia.

***Calliandra*** Benth. Hooker, J. Bot.2: 138. 1840. *Anneslia* Salisb., Parad. Lond. 1: 64. 1807. *Clelia* Casar., Nov. Stirp. Bras. Dec. 83. 1842. *Codnonandra* H. Karst., Fl. Columb. (H. Karst.) 2 (2): 43. 1863.

**Type species**: *C. houstoniana* Contr. U.S. Natl. Herb. 23: 386. 1922.

Shrubs or herbaceous perennials unarmed. Leaves bipinnate. Pinnae 1 to several pairs per leaf. Leaflets several pairs per pinna, persistent. Inflorescences solitary or in fascicles, axillary or arising from brachyblasts, arranged in spheric or obconical capitula, dense racemes, umbels, or terminal pseudoracemes. Flowers 5-merous, rarely 3–4- or 6-merous. Calyx campanulate or lobed. Corolla campanulate or infundibuliform, with evident lobes. Stamens 7–10, basally united, forming a tube a third or twice the length of the corolla. Fruit commonly ascending; linear–oblanceolate, straight, or falcate; with thickened margins, membranous but rigid, coriaceous, or woody; explosively dehiscent from the base to the apex.

A genus endemic to the American continent, with approximately 135 species [2], distributed in arid, semiarid, tropical, and subtropical vegetation [90]. From southwestern United States through Mexico to Uruguay, Chile, and Argentina.
1A.Inflorescences terminal to stems of current year, exerted from foliage, arranged in pseudoracemes or umbels***C. houstoniana*** ssp. ***houstoniana***1B.Inflorescences lateral, axillar or from efoliate brachyblasts22A.Leaves with 1 pair of pinnae32B.Leaves with 2 or more pairs of pinnae43A.Each pinnae exactly trifoliolate***C. tergemina*** ssp. ***emarginata***3B.Each pinnae with 3 or more pairs of pinnae54A.Peduncles 1–5 mm long; petals pubescent ***C. conferta***4B.Peduncles 10–20 mm long; petals glabrous or nearly so***C. isleyi***5A.Capitula with 1–3 flowers, typically 2; leaves spirally arranged on stems; leaves with 2–3 pairs of pinnae, leaflets 7–10 pairs per pinna; stamens 14–20 filaments***C. biflora***5B.Capitula with (3–) 5–18 flowers; leaves distichously arranged on stems; androecium with 16–50 filaments65A.Shrub with persistent stems in winter; androecium with 16–27 filaments***C. eriophylla*** ssp. ***eriophylla***5B.Herbaceous, stems arising annually from a taproot or rhizome; androecium with 30–78 stamens***C. humilis*** ssp. ***humilis***

***Calliandra biflora*** Tharp., Rhodora 56: 132. 1954.

**Type**: USA, Texas, DeWitt County, southwestern part of the county, 5-VII-1942, *S. Ridel* and *B.C. Tharp 44419* (Holotype: TEX374464!).

**Distinguishing features**: Herbaceous, sub-shrub, up to 40 cm tall, dying annually. Pinnae 2–3 pairs per leaf. Leaflets 7–10 pairs per pinna. Inflorescences axillary, solitary or in fascicles, arranged in capitula with 2–3 flowers. Flowers 5-merous. Stamens 14–20, united basally forming a tube. Fruit 4–7 × 1 cm, erect, straight, flattened, the margins 2 mm thick, stiff, reddish-brown, strigose.

**Representative examined material**: Tamaulipas: 29-IV-1985, *M. Martínez 723* (UAT).

**Comments**: Restricted to the Tamaulipan thorn scrub in lowlands, only in southern Tamaulipas (Aldama municipality) and southern Texas.

***Calliandra conferta*** Benth., Smith. Contr. Knowl. 3 (5): 63.1852. Basionym: *Feuilleea texana* Kuntze, Revis. Gen. Pl. 1: 187. 1891.

**Type**: USA, Rio Grande, 1848, *C. Wright s.n.* (Isosyntype: GH254215!). USA, Texas, from western Texas to El Paso, New Mexico [Hills of the San Felipe], 7-VII-1849, *C. Wright 167* (Isosyntype: GH254213!).

**Distinguishing features**: Shrub, 30–60 cm tall, intricate. Pinnae 1 pair per leaf. Leaflets 6–12 pairs per pinna, oblong. Inflorescences arranged in spheric capitula, with 2–8 flowers. Calyx with minute red glands on the lobes. Stamens 20–30, white or reddish filaments. Fruit 2–4 × 0.4–0.6 cm, erect, hard, dense, white, silky or strigose.

**Representative examined material**: Coahuila: 4-IX-2007, *J. Alba 210* (ANSM); 20-VIII-1987, *J.A. Villarreal 3881* (ANSM), 5-VI-1992, *J.A. Villarreal 6620* (ANSM); Nuevo León: 5-VI-2001, *E. Estrada 12886* (CFNL); 7-IV-2001, *E. Estrada 11827* (CFNL); 13-IV-2003, *E. Estrada 15526* (CFNL); 17-IV-2001, *E. Estrada 12436* (CFNL); 20-VII-2002, *E. Estrada 14976* (CFNL). 6-VI-2001, *E. Estrada 12919* (CFNL). Tamaulipas: 28-IV-1985, *L. Hernández 401* (UAT).

**Comments**: Common in scrubland of lowlands and highlands, associated with Tamaulipan thorn scrub, piedmont scrub, desert scrublands, and oak and oak–pine forests, 165–1845 m. Also in southern Texas.

***Calliandra eriophylla*** Benth., London J. Bot. 3: 105. 1844. ssp. ***eriophylla*.** Basionym: *Calliandra eriophylla* Benth., London J. Bot. 3: 105. 1844. *Anneslia eriophylla* Britton, Trans. N.Y. Acad. Sci. 14: 32. 1894. *Calliandra chamaedrys* Engelm. In A. Gray, Pl. Fendler 1: 39: 1848.

**Type**: Mexico, Chila, 1834, *G. Andrieux 405* (K82109!). Isosyntype: USA, Texas, from western Texas to El Paso, New Mexico, V-1849, *C. Wright 166* (K791236!).

**Distinguishing features**: Intricate shrub, up to 60 cm tall. Pinnae 2–4 pairs per leaf. Leaflets 8–19 pairs per pinna. Inflorescences arranged in capitula, with 2–7 flowers. Corolla red-pink or carmine. Stamens 21–34, the filaments pink or carmine. Fruit 4–10 × 0.5–1 cm, erect, flattened, thickened at the margin, densely hairy.

**Representative examined material**: Coahuila: 1-VII-1880, *E. Palmer 212* (NY 00549228!); 13-IX-1946, *F.A. Barkley 16012* (NY548474!). Nuevo León: 25-IX-1986, *E. Estrada 690* (CFNL); 3-V-1986, *E. Estrada 393a* (CFNL); 16-X-1986, *N. Reid s.n*. (CFNL). Tamaulipas: 28-V-1986, *L. Hernández 1812* and *E. Estrada* (UAT); 17-IX-1976, *F. Guevara 9808* (UAT); 20-IX-1976, *F. Guevara 9917* (UAT); 23-III-1999, *A. Mora-Olivo 7468* (UAT).

**Comments**: Recorded in low plains and on mountain slopes; associated with Tamaulipan thorn scrub, desert scrublands, desert grasslands, oak forest, and pine forest; in sandy and rocky soils; 360–800 m. Common in the Chihuahuan and Sonoran Deserts to Jalisco and Chiapas. Also in the SW and SE of the USA.

***Calliandra houstoniana*** (Mill.) Standl., (Mill.) Standl., Contr. U. S. Natl. Herb. 23: 386. ssp. ***houstoniana*.** Basionym: *Mimosa houstoniana* Mill., Gard. Dict., ed. 8: no. 16. 1768. *Anneslia houstoniana* (Mill.) Britt. and Rose, North Amer. Fl. 23: 70. 1928. *Mimosa houstonii* L’Hér., Sert. Angl. 30. 1788. *Acacia houstonii* (L’Hér.) Willd., Sp. Pl. 4: 1062. 1805. *Calliandra houstonii* (L’Hér.) Benth., J. Bot. (Hooker) 2: 139. 1840.

**Type**: Mexico, *W. Houston s.n.* (Lectotype: BM952405!).

**Distinguishing features**: Shrub, 0.8–1.5 m tall. Pinnae 5–14 pairs per leaf. Leaflets 40–68 pairs per pinna, 0.4–1.2 cm long, sharpening at apex, with midrib turned upward at apex. Inflorescences with black, bronze, or brown trichomes; arranged in terminal pseudoracemes up to 30 cm long. Corolla 5-merous. Stamens 36–100; 5.5 cm long or longer; mostly red, white; united basally, forming a tube. Fruit 6–14 × 1.3–2.3 cm, oblong, pilose or velutinous, the trichomes partially white or completely so, black or brown.

**Representative examined material**: Tamaulipas: 27-VI-1992, *J.L. Mora-López 1812* (UAT); 5-XII-1981, *A. Valiente B. 1* (UAT).

**Comments**: Recorded only in Tamaulipas; along roads; in disturbed areas; in semiarid, subtropical scrublands, and oak–pine forest; 200–1200 m. Along the Pacific Coast from Sonora to Chiapas. Along the Gulf Coast, from Tamaulipas to Campeche, also in Guatemala, El Salvador, Belize, and Honduras. This is the only species in northeastern Mexico that has terminal efoliate inflorescences and stamens 5.5 cm long or longer; no other species possesses these two characteristics together in the study area.

***Calliandra humilis*** Benth., London J. Bot. 5: 103. 1846. **ssp. *humilis***. Basionym: *Calliandra humilis* Benth., London J. Bot. 5: 103. 1846.

**Type**. Mexico, *Coulter*, *s.n.* (K82112!).

**Distinguishing features**: Herbaceous to sub-shrub, the stems simple, prostrate, 10–25 cm long, strigose to pilose. Pinnae 3–11 pairs per leaf. Leaflets 10–31 pairs per pinna, somewhat imbricate along rachis. Inflorescences arranged in capitula, with 2–12 flowers. Flowers strigulose or pilose. Stamens 30–70, basally united, forming a tube. Fruit erect, 2–5.8 × 0.5–0.7 cm, flattened, thickened at the margin, chartaceous, brown to light brown.

**Representative examined material**: Coahuila: 17-IX-1989, *E. Estrada 1831* (ANSM), 10-II-1997, *S. Wood* 9744 (ANSM).

**Comments**: Recorded only in the state of Coahuila, in desert scrub, arid grasslands, mesic oak forest and oak–pine forest, 800–2400 m. Also in northwestern Mexico (Sonora and Chihuahua) to Jalisco, Durango, and Zacatecas. Also common in the southwest (Arizona and New Mexico) and southeast (Texas) of the USA.

***Calliandra isleyi*** B.L. Turner, Lundellia 3: 17. 2000.

**Type**: USA, Texas, Brewster Co.: ca. 3 km west of Terlingua along highway 170; small limestone hills at base of Reed Plateau next to the Terlingua House of David Lanman, architect and builder (ca. 29°18′ N, 103°33′ W), 30-VI-2000, *B.L. Turner*, *G. Turner 20–386* (Holotype: TEX. Isotype: MEXU, NY).

**Distinguishing features**: Dwarf shrub, up to 30 cm tall. Pinnae 1 pair per leaf. Leaflets 6–9 pairs per pinna. Inflorescences axillar, arranged in capitula. Peduncles 1–2 cm long. Corolla glabrous or nearly so. Fruit linear oblanceolate, with white, appressed pubescence.

**Representative examined material**: Coahuila: 21-VIII-1941, *I.M. Johnston* (LL439435!); 26-VIII-1940, *C.H. Muller 739* (LL439462!); 2-VI-1941, *R.M. Stewart 384* (LL439492!).

**Comments**: Endemic to southwest Texas and northeastern Mexico [91]. Recorded only in the northern end of Coahuila, in desert scrub.

***Calliandra tergemina*** ssp. ***emarginata*** (Willd.) Barneby, Mem. New York Bot. Gard. 74 (3): 129–131. 1998. Basionym: *Inga emarginata* Humb. and Bonpl. ex Willd., Sp. Pl. 4 (2): 1009. 1806. *Mimosa emarginata* (Willd.) Poir. in Lam., Encycl. Suppl. 1: 39. 1810. *Calliandra emarginata* Benth., London, J. Bot. 3: 95: 1844. *Feulieea emarginata* Kubtze, Revis. Gen. Pl. 1: 187. 1891. *Anneslia emarginata* Britton and Rose, N. Amer. Fl. 23: 53. 1928.

**Type**: Mexico, Guerrero, Acapulco [Nov. Gen. Sp.: Crescit in litore occidentali Regni Mexicani, prope Acapulco], non-date, A.J.A. *Bonpland*, *F.W.H.A. von Humboldt 3859* (Isotype: P135163; P135161!).

**Distinguishing features**: Shrub, 1.5–4 m tall. Pinnae 1 pair per leaf. Leaflets 1 pair per pinna, rarely 2 pairs; however, each pinna has a trifoliolate appearance. In addition to the pair of distal leaflets, there is an extra proximal and smaller leaflet on the posterior side of the rachis, distal leaflets 1.5–7 × 0.7–3 cm. Inflorescences in axillar brachyblasts, arranged in spheric capitula. Flowers 4–5-merous. Corolla up to 1 cm long. Stamens 8–28, red-carmine or white basally and pink or red distally, united basally, forming a tube. Fruit 5–13 × 0.6–1.5 cm, spread, oblong, glabrous or pilose.

**Representative examined material**: Tamaulipas: 26-IV-1960, *M.C. Johnston 5347*, *J.R. Crutchfiled* (TEX-LL); 27-X-1959, *M.C. Johnston 4536*, *J.G. Graham* (TEX-LL); 16-IX-1960, *M.C. Johnston 5574*, *J.G. Graham* (TEX-LL); 8-II-1960, *M.C. Johnston 5068B* (TEX-LL); 15-II-1939, *H. LeSeur 162* (TEX-LL).

**Comments**: Distributed in tropical areas of Mexico, through Central America to Colombia and Venezuela. This species is the only one in northeastern Mexico that has apparently trifoliolate pinnae; the rest of the species have more than 2 pairs of leaflets per pinna. It is the only species with leaflets longer than 1 cm.

***Chloroleucon mangense*** (Jacq.) Britton and Rose ssp. ***leucospermum*** (Brandegee) Barneby and Grimes, has been reported in Tamaulipas [92]; however, its presence has not been able to be verified with herbarium specimens present in national or foreign herbaria collections that appear in SEINet Portal Network (https://wbiodiversity.org/seinet/index.php (accessed on 15 June 2023)); therefore, it is not included in the dichotomous keys nor in the descriptions of the species of this work. This species is mostly distributed along the Pacific Coast from Baja California Sur to Chiapas and Central America.

***Cojoba*** Britton and Rose, N. Amer. Fl. 23 (1): 29. 1928.

**Type species**: *C. arborea* (L.) Britton and Rose, N. Amer. Fl. 23 (1): 29. 1928.

Shrubs or trees, up to 8 m tall. Leaves bipinnate, pinnae 1–12 pairs per leaf. Leaflets 2–50 pairs per pinna. Petiole with a sessile gland, arising below the pair of proximal pinnae, rarely 1–2 of in the petiole, and sometimes, small glands between some pairs of leaflets. Inflorescences solitary or fascicled, arranged in spherical capitula. Flowers sessile, 5-merous, whitish-green, glabrous except at the apex of the lobes. Calyx campanulate, 5-lobed. Corolla campanulate. Stamens 20–66; fused basally, forming a tube; the tube shorter, equal to or longer than corolla. Fruit pendulous; linear or broadly linear; falcate; or contorted; constricted between seeds or thickened and cylindrical with immersed sutures; fleshy and bright red when young; turning leathery, brown and wrinkled with age; dehiscent by 1 or both sutures; the valves wrinkling and contracting to expose seeds.

A genus with 12 species [2,92], distributed from Mexico to South America. Only 1 species in northeastern Mexico.

***Cojoba arborea*** (L.) Britton and Rose **ssp. *arborea.*** Basionym: *Mimosa arborea* L., Sp. Pl. 1: 519. 1753. *Acacia arborea* (L.) Willd., Sp. Pl., ed. 4 [Willdenow] 4 (2): 1064. 1806. *Pithecellobium arboreum* (L.) Urban., Symb. Antill. (Urban). 2 (2): 259. 1900. *Samanea arborea* (L.) Ricker, in L.H. Bailey, Stand. Cycl. Hort. 3066. 1917. *Cojoba arborea* (L.) Britton and Rose N. Amer. Fl. 23 (1): 29. 1928.

**Type**: Mexico, Mt. Ovando, 9-IV-1937 to 12-IV-1937, *E. Matuda 1835* (Isotype: MO-125077!)

**Distinguishing features**: Tree, 5–20 m tall, unarmed. Pinnae mostly 6–12 pairs per leaf, decreasing in size proximally, rarely distally. Leaflets 20–50 pairs per leaf, dark green, shiny. Foliar gland sessile, arising between or immediately below each pair of pinnae and frequently 1–2 near mid-portion of petiole, rarely a gland at the apex of some pinnae rachis. Inflorescences in fascicles, arranged in spheric capitula. Flowers greenish, sometimes tinged with red. Stamens 20–42. Fruit 8–18 × 0.5–1.4 cm, pendulous, moniliform, falcate and coiled, fleshy, velvety or densely strigulose, bright red when young, turning brown or reddish-brown with age, brittle and leathery when ripe, dehiscent through both sutures. Seeds plumpy, ellipsoid black, shiny.

**Representative examined material**: Nuevo León: 25-VI-2004, *E. Estrada 16547* (CFNL).

**Comments**: Common in humid forests of southeastern Mexico, in lowlands and slopes of the Sierra Madre Oriental, and north of South America. Rare in northeastern Mexico, cultivated as ornamental. *Cojoba arborea* has three varieties, *C. arborea* var. *arborea*, *C. arborea* var. *cubensis* (Bisse) Barneby and Grimes, and *C. arborea* var. *angustifolia* (Rusby) Barneby, the last 2 varieties are distributed in Central and South America respectively. Widely used as ornamental in public and private gardens.

***Ebenopsis*** Britton and Rose, N. Amer. Fl. 23 (1): 33. 1928.

**Type species**: *Ebenopsis ebano* (Berland.) Barneby and J.W. Grimes, Mem. New York Bot. Gard. 74 (1): 175. 1996.

Trees or shrubs. Stipules spinescent, paired, subulate with dilated base, early lignified, persistent. Leaves bipinnate. Pinnae 1–4 pairs per leaf. Leaflets 2–7 pairs per pinna. Gland columnar, arising between distal pair of pinnae. Inflorescences arising from brachyblasts, arranged in dense sub-spheric capitula or spikes. Flowers sessile, 5-merous. Calyx gamosepalous, 5-merous. Corolla gamopetalous, 5-merous. Stamens 32–66, united basally and forming a tube. Fruit woody, subcylindrical, coriaceous, septate internally, exocarp breaking into polygons with age, dehiscence longitudinal, late. Seeds arranged transversely, ovoid to cylindrical.

A genus of three species presents in hot and dry areas of Mexico and the southern USA [89].

***Ebenopsis ebano*** (Berland.) Barneby and J.W. Grimes, N. Y. Bot. Gard. 74 (1): 175. 1996.

Basionym: *Mimosa ebano* Berland., Mosaico Mex. 4: 418. 1840. *Pithecellobium texense* J.M. Coult., Contr. U.S. Natl. Herb. 1: 37. 1890. *Pithecellobium flexicaule* J.M. Coult., Contr. U.S. Natl. Herb. ii. 101. 1891. *Zygia flexicaulis* Sudw., Bull. Div. Forest. U.S.D.A. 14: 248. 1897. *Siderocarpos flexicaulis* Small, Bull. New York Bot. Gard. 2: 91. 1901. *Samanea flexicaulis* J.F. Macbr., Contr. Gray Herb. 59: 2. 1919. *Ebenopsis flexicaulis* Britton and Rose, N. Amer. Fl. 23 (1): 33. 1928. *Chloroleucon ebano* (Berland.) L. Rico, Kew Bull. 46 (3): 519. 1991.

**Type**: Mexico, de S. Fernando à Santander, 1832, *J.L. Berlandier 2262* (Isolectotype: G364652!). Isolectotype: Mexico, s. loc., 1832, *J.L. Berlandier 2282* (P1818503!)

**Distinguishing features**: Tree up 12 m tall; distal branches in zigzag, armed with nodal paired, lignified spines. Pinnae 1–3 pairs per leaf, accrescent distally. Leaflets 4–6 pairs per pinna, accrescent distally. Petiole with a stipitate gland arising between each pair of pinnae. Inflorescences axillar, arranged in cylindrical spikes or racemes. Corolla subcylindrical. Stamens 34–66, united basally forming a tube. Fruit 6–18 × 1.8–3 cm, oblong, straight or curved, coriaceous, woody when ripe, septate, lately dehiscent, separating at the ends, dark brown, persistent for a long time.

**Representative examined material**: Nuevo León: 5-VII-2001, *E. Estrada 12890* (CFNL); 10-IV-2002, *E. Estrada 12117* (CFNL); 5-VII-2001, *E. Estrada 12884* (CFNL); 19-VII-2002, *E. Estrada 14956* (CFNL); V-1983, *F. Wolf 29* (CFNL); 6-VI-2001, *E. Estrada 12909* (CFNL); 8-VI-2002, *E. Estrada 14602* (CFNL); 17-IV-2001, *E. Estrada 12492* (CFNL). Tamaulipas: 25-XI-1984, *R. Díaz 207* (ANSM); 14-II-1985, *M. Martínez 198* (UAT); 25-XI-1984, *L. Hernández 1274* (UAT); 16-V-1984, *C. González 70* (UAT); 30-VIII-1984, *McDonald 731* (UAT).

**Comments**: Common species in the Tamaulipan thorn scrub, 195–650 m. From southeastern Texas (USA), also present in Yucatán. One of the species of the Tamaulipan thorn scrub, outstanding for its size over the rest of the lower canopy. Species subject to overexploitation in the region; the wood is widely used as a source of coal, firewood, posts, food (immature edible fruits (seeds) are called *maguacatas*), handicrafts, and hand tools. The “ebony” is one of the hardest wood species in northern Mexico, widely used as firewood; also, its flowers are mixed in alcohol and used as an aromatic essence for the skin [73].

***Enterolobium*** Mart. Flora 20. 102. 1837.

**Type species**: *E. timbouva* Mart. Flora 20 (2): Beibl. 128. 1837.

Trees up to 20 m tall, unarmed, deciduous. Leaves bipinnate. Pinnae 2–30 per leaf. Leaflets 4–80 pairs per pinna. Stipules caducous. Petiole with gland on top or immersed into the petiolar groove. Inflorescences axillary, solitary or fascicled, forming a dense umbelliform capitula. Corolla 5–8-merous. Calyx 5–8 lobed. Corolla 5–8 lobed, almost twice as long as the calyx. Stamens 8–70, basally united, forming a tube, exerted from the corolla. Fruit laterally compressed; oblong or linear; broad; spirally coiled in half or 2 circles into an auricular, reniform, or semicircular shape; indehiscent; outermost layer hardened; internally septate between the seeds. Seeds compressed but plumpy, transversely placed.

Genus of 10 [89]–11 species [89], distributed in tropical regions of the New World, from Mexico and adjacent islands to northern Argentina. In northeastern Mexico, there is only one species recorded from south of the Tropic of Cancer, from Tamaulipas to the south of Mexico.

***Enterolobium cyclocarpum*** (Jacq.) Griseb., Fl. Brit. W. Ind. p. 226. 1860. Basionym: *Mimosa cyclocarpa* Jacq., Fragm. Bot. p. 30. 1801. *Inga cyclocarpa* (Jacq.) Willd., Sp. Pl., ed. 4 [Willdenow] 4 (2): 1026. 1806. *Feuilleea cyclocarpa* Kuntze, Revis. Gen. Pl. 1: 184. 1891. *Pithecellobium cyclocarpum* (Jacq.) Mart., Flora 20 (2, Beibl.): 115. 1837. *Albizia longipes* Britton and Killip, Ann. New York Acad. Sci. 35: 132. 1936. 1800. *Mimosa parota* Sessé and Moc., Naturaleza (Mexico City) ser. 2, 1, app. 177. 1890. *Prosopis dubia* Kunth, Nov. Gen. Sp. [H.B.K.] VI. 309. 1824.

**Type**: Colombia, Atlántico. Sabana Larga near Barranquilla, II-1928, *Elias*, *Bro 499* (Isotype: US469!).

**Distinguishing features**: Tree, deciduous, up to 30 m tall, the canopy cover often wider than tall. Petiole with a sessile gland, arising above the middle part, and glands also often present on the rachis or on the insertion of leaflets. Pinnae 5–12 pairs per leaf. Leaflets 15–45 pairs per pinna. Stamens 20–68, basally united in the bottom half, forming a tube, immersed in the corolla. Fruit 7–12 × 5–7 × 0.5–1 cm, broadly oblong, coiled, forming a circular or semicircular structure, the exocarp rigid, dark brown to black, shiny, constricted between the seeds, mesocarp pulpy and resinous.

**Representative material examined**: Tamaulipas: 17-V-1984, *C.G. Romo 72* et al., (UAT); 31-III-1998, *M. Galván 728* (UAT).

**Comments**: Most frequently at elevations below 480 m, in deciduous forest, thorny woodlands, dry subtropical forest, disturbed areas in evergreen and semi-deciduous humid subtropical and tropical forest. Recorded only in the state of Tamaulipas, in low plains, associated with low deciduous forest, 200–500 m. Its vernacular name is *parota*. From northeastern Mexico through Central America to Colombia, Guyana, and Venezuela to the north of Brazil. Frequently used as a timber species for interior finishes, kitchen utensils, boats manufacturing, and furniture; its seeds are used to make necklaces, bracelets and earrings. Widely used as ornamental for its majestic bearing.

***Havardia*** Small, Bull. New York Bot. Gard. 2: 91. 1901.

**Type species**: *Havardia brevifolia* Small, Bull. New York Bot. Gard. 2: 92. 1901.

Trees or shrubs. Stipules spinescent, subulate to conical. Leaves bipinnate. Pinnae 1–9 pairs per leaf. Leaflets 7–36 pairs per pinna, commonly opposite on the rachis. Petiole with a sessile gland, arising below the proximal pair of pinnae. Inflorescences axillar or from bachyblasts, arranged in spheric capitula or capituliform racemes. Flowers 5–6-merous. Calyx campanulate. Corolla 5-merous, white to whitish. Stamens 28–52, united basally forming a tube. Fruit straight, oblong to linear, flattened, flexible, slightly leathery, dehiscence inert along both sutures.

A genus of 5 species [89], distributed from southern Texas to Central America, in warm temperate and seasonally dry regions. In northeastern Mexico most frequent in lowlands associated with Tamaulipan thorn scrub, piedmont scrub, and highlands in desert scrub (1200–1600 m).

***Havardia pallens*** (Benth.) Britton and Rose. Basionym: *Calliandra pallens* Benth. London, J. Bot. 5: 102. 1846. *Pithecellobium pallens* (Benth.) Standl., Trop. Woods 34: 39. 1933.

**Type**: “Mexico, Hidalgo, Zimapan, non-date, *Coulter s.n.* (Holotype: K82453!).

**Distinguishing features**: Shrub or tree 2–12 m tall, branches armed with thorny paired, ascending stipules. Pinnae 2–7 pairs per leaf. Leaflets 12–21 pairs per pinna, oblong, light green. Petiole with a sessile gland, arising in the basal portion, middle or above mid-petiole, a second arising in the proximal pair of pinnae. Inflorescences arranged in spherical or semi-ovoid capitula. Stamens 28–52, united basally, forming a tube. Fruit 7–12 × 1.2–2 cm, oblong, straight, flattened, light brown, dehiscent, subcoriaceous but flexible.

**Representative material examined**: Coahuila: 30-II-1992, *J.A. Villarreal 1427* (ANSM), 27-IX-2001, *J.A. Encina 916* (ANSM), 3-XI-2007, *J.A. Encina 2233* (ANSM), 29-IX-2006, *J.A. Encina 1923* (ANSM). Nuevo León: 5-VII-2001, *E. Estrada 12889* (CFNL); 5-VII-2001, *E. Estrada 15850* (CFNL); 2-X-1991, *G. Hinton* et al. *21578* (TEX-LL). 3-VII-2001, *E. Estrada 13060* (CFNL). 6-IX-1991, *G. Hinton* et al. *21422* (TEX-LL); 23-VII-2002, *E. Estrada 15098* (CFNL); 15-VII-1933, *C. Müller 531* (TEX-LL); 8-VI-2002, *E. Estrada 14599* (CFNL). Tamaulipas: 25-XI-1984, *L. Hernández 1275* (UAT); 26-XI-1990, *J. Sifuentes 57* (UAT); 25-X-1984, *C. González Romo 290* (UAT); 25-XI-1984, *L. Hernández 1273* (UAT).

**Comments**: From the south of Texas to Guerrero and Chiapas. Abundant species in the area, most frequently in the Tamaulipan thorn scrub and piedmont scrub, one of the most important species in the landscape, in the lowlands and mountain slopes (190–650 m), less frequent in highlands (1500 m). Popularly known of *tenaza*. In rural areas is used extensively for the manufacture of manual utensils, poles, chairs, and tables of great resistance [74,76].

***Inga*** Mill., Gard. Dict. Abr. Ed. 4. 1754.

**Type species**: *Inga vera* Willd., Sp. Pl., ed. 4 [Willdenow] 4 (2): 1010. 1806.

Trees, unarmed. Leaves pinnate, paripinnate. Petiole winged or unwinged, with a gland almost always between the insertion of each pair of leaflets. Inflorescences axillar o cauliflorous, arranged in spheric capitula, racemes, panicles, or umbels. Calyx 4–6-merous, rarely spathaceous campanulate, cylindrical or infundibuliform, up to 4 cm long. Corolla 4–6-merous, up to 6 cm long, cylindric or infundibuliform. Stamens 20–300, up to 12 cm long, united basally, forming a tube shorter, equal, or longer than corolla. Fruit 0.05–2 m long, flattened, cylindric, quadrangular or convex, indehiscent, straight, curved, coiled, or twisted, the valves body usually broader than margins, the margins sometimes winged or raised. Seeds fleshy, testa thin with a white sugary cover.

A very diversified American genus, with almost 300 recognized species, and other 50 not deeply studied [93]. Distributed from north Mexico, south of the Tropic of Cancer, throughout Mexico, Central America, the Caribbean to Uruguay, with almost 40 species recorded in Mexico and Central America [2]. In northeastern Mexico we recorded 2 species.
1A.Leaf rachis winged, pubescent, leaflets 4–8 pairs per leaf, pubescent, frequently with minute red glands abaxially; calyx tube up to 1.5 cm long: stamens 60–130; pod 5–20 cm long, cylindrical or quadrangular ***I. vera*** ssp. ***vera***1B.Leaf rachis subcylindrical, glabrous, leaflets 3–4 pairs per leaf, glabrous; calyx tube 1.5–1.75 cm long; stamens 45–60; pod 14–40 cm long, sub-oblong, laterally flattened ***I. inicuil***

***Inga inicuil*** Cham. and Schltdl. Linnaea 12: 559. 1838. Basionym: *Feuilleea jinicuil* (Schltdl.) Kuntze, Rev. Gen. 1: 188. 1891. *Inga paterno* Harms. Repert. Spec. Nov. Regni Veg. 13: 419. 1914. *Inga radians* Pittier, Contr. U.S. Natl. Herb. 18: 178. 1916.

**Type**: Mexico, Veracruz, Jalapa, VIII-1828, *C.J.W. Schiede*, *675* (Isotype: GH65937!; HAL98630!; CAS0003023!)

**Distinguishing features**: Tree. Leaves pinnate, paripinnate. Leaflets 3–4 pairs per leaf, elliptic, acute at both ends. Petiole cylindrical or nearly so (not winged). Glands arising in the insertion of the leaflets. Inflorescences arranged in spheric capitula or umbels. Calyx tube 1.5–1.75 cm long. Corolla up to 6.5 cm long. Stamens 45–60, united basally, forming a tube up to 8.5 mm long. Fruit 12–38 × 5–7 × 2.2–3 cm, straight or curved, the margins shallow raised, the body sometimes transversely ribbed, glabrous.

**Representative material examined**: VIII-1828, *C.J.W. Schiede 195 s.n. “675”* (HAL98630!).

**Comments**: From Tamaulipas and San Luis Potosí to Central America, and an isolated record from Ecuador [86]. In deciduous tropical forest, cloud forest, tall evergreen forest, 160–1650 m. Cultivated as ornamental in public and private gardens.

***Inga vera*** Willd., Sp. Pl., ed. 4 [Willdenow] 4 (2): 1010. 1806. ssp. ***vera***. Basionym: *Mimosa inga* L., Sp. Pl.: 516. 1753. *Inga vera* subsp. *lamprophylla* C. Wright ex Pittier, Contr. U.S. Natl. Herb. 18: 216. 1916. *Inga vera* subsp. *portoricensis* Pittier, Contr. U.S. Natl. Herb. 18: 217, pl. 104. 1916. *Inga vera* subsp. *eriocarpa* (Benth.) J. León, Ann. Missouri Bot. Gard. 53: 338. 1966. *Inga vera* subsp. *spuria* (Humb. and Bonpl. ex Willd.) J. León, Ann. Missouri Bot. Gard. 53: 339. 1966. *Inga vera* subsp. *affinis* (DC.) T.D. Penn., Gen. Inga, Bot. 716. 1997.

**Type**: Colombia, Between Pamplonita and Chinácota, Rio Pamplonita Valley, 17-III-1927, *E.P. Killip*, *A.C. Smith 20770* (Isotype: GH65979!)

**Distinguishing features**: Tree, unarmed, up to 20 m tall. Leaves pinnate, paripinnate; the rachis winged, 1–1.2 cm wide. Leaflets 4–8 pairs per leaf, the proximal pair the shortest, elliptic, often with tiny red glandular trichomes abaxially. Foliar glands arising in the insertion of the leaflets. Inflorescences solitary or paired, arranged in spikes or racemes. Calyx tube up to 1.5 cm long. Corolla up to 2.2 cm long. Stamens 60–130, basally united, forming a tube. Fruit 2–20 × 1.2–3.5 cm, quadrangular to cylindrical, straight to curved, its faces up to 1.5 cm wide, covered by the ribbed margins.

**Representative material examined**: Tamaulipas: 23-V-2000, *R.M. Flores*, *s.n.* (MEXU); 29-IV-1985, *M. Martínez 539* (UAT); 6-VI-1987, *A. Brito 485* (CFNL); 18-X-1988, *A. Mora-Olivo 554* (UAT); 18-II-2001, *A. Mora-Olivo 9135* (UAT).

**Comments**: Recorded only in the state of Tamaulipas, more frequent in moist and rainy forest, below 1000 m. Distributed in Mexico through Central America and Antilles to Colombia, Venezuela, and Ecuador. Species easily recognized since it is the only species in northeastern Mexico with winged petiole and cylindrical or quadrangular fruit.

***Lysiloma*** Benth., London J. Bot. 3:82. 1944.

**Type species**: *Lysiloma schiedeanum* Benth., London J. Bot. 3: 83. 1844. = *Lysiloma divaricatum* (Jacq.) Benth., Repert. Bot. Syst. (Walp.) 5 (4): 594. 1846.

Trees, unarmed. Leaves bipinnate. Pinnae 1–40 pairs per leaf. Leaflets variable in number, few and large or abundant and small, paired. Petiole with a sessile gland arising in the middle or little above it, also, the rachis with 2 glands, the first in the insertion of the proximal pair of pinnae, and other, in the insertion of the distal pair of pinnae. Inflorescences solitary or paniculate, arranged in spheric capitula or cylindric spikes. Flowers 5-merous. Calyx campanulate. Corola campanulate larger than calyx, glabrous to pubescent. Stamens 10–30, basally united, forming a short tube immersed in the corolla. Fruit persistent for a long time, oblong to oblong–elliptic, flattened, compressed, straight or sometimes twisted, dehiscent, valves detaching from margin, which is persistent, and falling as a unit.

Genus of eight [89] or nine species, distributed from the southwest and southeast of the USA, through Mexico to Costa Rica [94]. Two species in northeastern Mexico.
1A.Inflorescences narrow-cylindrical, floral axis 3.5–6.5 cm long***L. acapulcense***1B.Inflorescences in spheric or ellipsoid capitula, floral axis 8–20 mm ***L. divaricatum***

***Lysiloma acapulcense*** (Kunth) Benth., London J. Bot. 3: 83. 1844. Basionym: *Acacia acapulcensis* Kunth, Mimoses p. 78. T. 24. 1819. *A. desmostachys* Benth., Pl. Hartw. P. 13. 1839. *L. desmostachyum* (Benth.) Benth., Lond. J. Bot. 3: 84. 1844. *L. jorullensis* Britt. and Rose, North Amer. Fl. 23: 77. 1928. *L. platycarpa* Britt. and Rose, North Amer. Fl. 23: 78. 1928.

**Type**: Mexico, Acapulco [Nov. Gen. Sp.: Crescit in Regno Mexicano, prope Acapulco, ad litus Maris Pacifici, locis arenosis], 1824, *A. Bonpland s.n.* (P679354!).

**Distinguishing features**: Shrub or tree, up to 15 m tall. Stipules foliaceous, caducous. Petiole with a reddish-black gland arising at the middle or nearly so, 1 gland arising on the insertion of the first pair of pinnae, and, frequently, 2 glands arising in the insertion of 2 distal pairs of pinnae. Pinnae 8–16 pairs per leaf. Leaflets 19–50 pairs per pinna, Inflorescences axillary, solitary or in fascicles, arranged in spikes. Fruit 11–18 × 2–5 cm, oblong, straight, flattened, coriaceous-papyraceous.

**Representative material examined**: Tamaulipas: 1-V-1959, *Rzedowski 10352* (TEX 257855); 30-IV-1960, *M.C. Johnston*, *J Crutchfiled 5393* (MICH1171125!; TEX257790!).

**Comments**: Recorded only in the state of Tamaulipas, in low deciduous forests and medium semi-deciduous forests, 900 m. Easily recognized by its spiked inflorescences. From northeastern Mexico to Central America (Costa Rica and Nicaragua).

***Lysiloma divaricatum*** (Jacq.) 
J.F. Macbr., Contr. Gray Herb. 59: 6. 1919. Basionym: *Mimosa 54ivaricate* 
Jacq., Pl. Hort. Schoenbr. 3: 776, pl. 395. 1798.

**Type**: Mexico, Puebla, Santa Lucía, collected in the vicinity of San Luis Tultitlanapa, Puebla, near Oaxaca, VI-1908, *C.A. Purpus s.n.* (Isotype: MO-045445!). Mexico, Guerrero, San Jerónimo, 12.X-1998, *Langlasse 715* (Isotype: K82437!)

**Distinguishing features**: Tree, 18–19 m tall. Pinnae 3–13 pairs per leaf. Leaflets 12–42 pairs per pinna. Petiole sparsely pubescent, with a reddish-brown or dark gland arising in the insertion of the first pair or pinnae, also with other 2 glands arising in the distal pairs of pinnae. Inflorescences fasciculate, arranged in spheric capitula. Fruits, 7–18 × 1.4–2.7 cm, oblong, membranous chartaceous, brown.

**Representative material examined**: Tamaulipas: 6-X-2000, *E. Estrada 13168* (CFNL); 12-X-2002, *E. Estrada 15153* (CFNL); 10-X-1996, *C. Ramos 148* (CFNL).

**Comments**: In the north of Mexico, on both coasts, Baja California, and Tamaulipas, also in Chihuahua, in the center and south of the country to Veracruz, Guerrero, Oaxaca and Central America. Frequent in tropical deciduous forest, cloud forest, 500–1490 m. Easily recognized by its spheric capitula. The wood is commonly used as firewood, since it lasts a long time burning. It is also used for tool handles.

***Painteria*** Britton et Rose, N. Amer. Fl. 23 (1): 35. 1928.

**Type species**: *Painteria revoluta* Britton and Rose, N. Amer. Fl. 23 (1): 35. 1928.

Shrub, few- or multibranched, up to 1.5 m tall. Stipules spinescent, paired, abundant, lignified, semi-straight or curved at the nodes. Pinnae 1–7 pairs per leaf. Leaflets 3–20 pairs per pinna. Petiole with a gland arising in the insertion of the proximal pair of pinnae, rarely between 2 pairs. Inflorescences arising from brachyblasts, arranged in subspheric or spiciform capitula. Flowers 4–7-merous. Calyx 5–7-merous, campanulate to hemispherical. Corolla campanulate. Stamens 28–76, free at the apex, united basally, forming a tube. Fruit compressed, somewhat turgid, falcate or circinate, leathery, biconvex, tardily dehiscent. Seeds plumply.

A Mexican genus of 3 species [89], in the highlands in north and central Mexico, and low and high plains of northeastern Mexico.

***Painteria elachistophylla*** (S. Watson) Britton and Rose, N. Amer. Fl. 23: 36. 1928. Basionym: *Pithecellobium elachystohyllum* A. Gray ex S. Watson, Proc. Amer. Acad. Arts 17: 352. 1882.

**Type**: Mexico, Nuevo León, Monterrey, Nuevo León, 1880, *Palmer 289* (435784NY!).

**Distinguishing features**: Shrub or sub-shrub, up to 1.5 m tall, the branches flexuose. Nodes with spinescent, straight, ascending stipules. Pinnae 1–2 pairs per leaf. Leaflets 3–12 pairs per pinna, suborbicular or oblong. Petiole with a gland at the insertion of the proximal pair of pinnae. Inflorescences reddish to pink, arranged in spheric or short spikes. Stamens 28–44, united basally forming a tube. Fruit linear, falcate, semicircular, 5–11 × 1–1.5 cm, slightly or strongly constricted between seeds, dehiscence inert. Seeds globose, red, sometimes turning brown with age.

**Representative material examined**: Coahuila: 28-VII-1976, *A. Roig 89* (ANSM), 8-VII-1980, *A. Rodríguez 53* (ANSM), 9-VI-1981, *A. Rodríguez 1045* (ANSM). Nuevo León: 27-III-2003, *E. Estrada 15313* (CFNL); 2-III-2003, *E. Estrada 15242* (CFNL); 2-VII-2000, *E. Estrada 11577* (CFNL); 3-VII-2001, *E. Estrada 13054* (CFNL). Tamaulipas: 24-V-1975, F. González-Medrano 9125 *(UAT)*; *18-IX-1976*, *F. González-Medrano 9832* (UAT); 25-VII-1985, *D. Méndez 89 (UAT).*

**Comments**: Endemic to Mexico. Frequent in desert scrub, arid conifer forest, mesic oak forest, oak–pine forest at the higher altitudes of the study area, 1300–2200 m. Also in Chihuahua and San Luis Potosi to Michoacán, Tlaxcala, and Puebla.

***Pithecellobium*** Mart. Fl. 20: 114. 1837.

**Type species**: *Pithecellobium unguis-cati* (L.) Benth., London J. Bot. 3: 200. 1844.

Shrubs or trees, 1–20 m tall. Stipules nodal, spinose, lignescent. Pinnae 1–2 pairs per leaf (northeastern Mexico). Leaflets 1–3 pairs per pinna (northeastern Mexico). Foliar glands arising between the insertion of each pair of pinnae, and in the end of each pinna rachises. Inflorescences axillar or in brachyblasts, in panicles or pseudoracemes, arranged in spheric capitula or spikes. Flowers sessile, 5-merous. Calyx cylindric, campanulate to hemispheric. Corolla cylindric to infundibuliform, rarely turbinate. Stamens 16–76, basally united, forming a tube shorter or larger than corolla. Fruit linear to oblong in profile view coiled and/or twisted, chartaceous or woody, the valves bulging over seeds, internal cavity continuous or septate, dehiscent by 1 or both sutures. Seeds plumpy, black, shiny, with a spongy aril.

American genus of 18 species [92] of the tropical and subtropical lowlands, from Florida to South America. In northeastern Mexico, three species were recorded.
1A.Flowers arranged in spikes; receptacle 1 cm or longer; fruit dehiscent only through ventral suture, valves woody, rigid, not flexible***P. lanceolatum***1B.Flowers arranged in spherical capitula or compact spikes; receptacle 1 cm long or shorter; fruit dehiscent through both sutures, stiff and leathery but flexible when pressed22A.Peduncles 2–4, 1.5 cm long or shorter; ovary stipe 1.9 mm or shorter, body thinly minutely papillated-puberulent; flowers in dense capitula, perianth gray, silky, puberulent ***P. dulce***2B.Peduncles solitary, (1.5) 2–3 cm long; stipe of ovary variable, up to 3 mm long, when shorter than 2 mm, body glabrous or micro-papillate (under microscope at high magnification); flowers in spherical or spiked flower heads, if capitate, the ovary glabrous or slightly strigulous or if pubescent, the largest flower at least 2.5 mm long***P. unguis-cati***

***Pithecellobium dulce*** (Roxb.) Benth., London J. Bot. 3: 199. 1844. Basionym: *Mimosa dulcis* Roxb., Pl. Coromandel 1 (4): 67, t. 99. 1795.

**Type**: Mexico, Oaxaca, Oaxaca Near Oaxaca, Cordillera, non-date, *H. Galeotti 3140* (Isotype: K82421!; BR5174249!).

**Distinguishing features**: Shrub or tree, 3–15 m tall. Stipules lignified, spinescent. Petiole with a stipitate gland at the distal end, and a similar, but smaller gland at the insertion of each pair of leaflets. Pinnae 1 pair per leaf. Leaflets 1 pair per pinna, bicolored. Inflorescences axillary or in fascicles, arranged in spheric capitula. Peduncles 1.5 cm or shorter. Fruit, 8.5–19 × 1.3–1.5 cm, coiled, constricted between seeds, acute apically, glabrous or puberulent, dehiscent by both sutures, reddish-brown, reticulate. Seeds black, the aril white to pinkish.

**Representative material examined**: Nuevo León: 28-VI-2001, *E. Estrada 14518* (CFNL); 5-VI-2019, *A. Cuéllar*, *s.n.* (CFNL). Tamaulipas: 20-IV-1988, *E. Estrada 1390* (CFNL); 15-III-1994, *L. Hernández 3004* (CFNL); 23-IV-2009, *E. Estrada 20764* (CFNL); 14-VI-1984, *L. Hernández 1102* (UAT); 6-VI-1997, *A. Mora-Olivo 7245* (UAT).

**Comments**: In lowlands in tropical deciduous forest, riparian communities, and areas with anthropogenic disturbance; 250–1000 m. Also in New Mexico (USA) to Colombia, Venezuela, and the West Indies. Its vernacular name is *Guamuchil*. Multifunctional species, used as timber, honey plant, source of tannins for joinery, and raw food (arile).

***Pithecellobium lanceolatum*** (Willd.) Benth., London J. Bot. 5: 105. 1846. Basionym: *Inga lanceolata* Humb. and Bonpl. ex Willd., Sp. Pl. 4: 1005, 1806. *Mimosa lanceolata* (Willd.) Poir., Encyl. Suppl. 1: 37. 1810.

**Type**: Mexico, Tamaulipas, Vicinity of Tampico, Tamaulipas, 27-IV-1910, *E. Palmer 307* (Isotype: K82414!).

**Distinguishing features**: Tall shrub or tree, up to 18 m tall. Stipules firm but not always spinescent; straight, linear or conic. Pinnae 1 pair per leaf. Leaflets 1 pair per pinna, accrescent distally, bicolored. Gland arising at distal end of the petiole, and a gland in the insertion of each pair of pinnae, stipitate. Inflorescences axillar, arranged in lax spikes. Fruit 3.4–14 × 0.7–1.2 cm, biconvex, but subcylindrical, straight or curved in a whole circle, fleshy and red when young, lignified, rigid and brown with age, rugose, elastically dehiscent through ventral suture.

**Representative material examined**: Tamaulipas: 18-IX-1994, *I. García G. s.n.* (MEXU).

**Comments**: In deciduous woodlands, piedmont scrub, disturbed and sunny places with low canopy cover. South of the Tropic of Cancer, Pacific and Gulf coast, Central America to Colombia. Easily differentiated in northeastern Mexico by its leaves with only a pair of pinnae and each pinnae with a long pair of leaflets and the spiked inflorescences instead of the spheric capitula present in the other 2 species.

***Pithecellobium unguis-cati*** (L.) Benth. London J. Bot. 3: 200. 1844. Basionym: *Mimosa unguis-cati* L., Sp. Pl. 517: 1753.

**Type**: Mexico, Oaxaca, Cordillera. (Oaxaca) Mexico, IV-1840 to IX-1840, *H.G. Galeotti 3140* (Isotype: G364403!)

Tree, up to 10 m tall. Stipules spinescent at nodes. Pinnae 1 pair per leaf. Leaflets 1 pair per pinna, obovate–suborbicular, bicolored. Gland stipitate, arising at the distal end of the petiole and a similar, but smaller gland at the insertion of each pair of leaflets. Inflorescences axillar, solitary or paired, up to 3 cm long, arranged in spheric capitula, racemes or spikes. Ovary stipitate, stipe up to 3 mm long. Fruit pendulous, 6–20 × 0.7–1 cm, linear, curved, coiled in 1–2 circles, and twisted, compressed but bulging and convex over seeds, fleshy red when young, turning dry, chartaceous brown with age, dehiscent by both sutures.

**Representative material examined**: Tamaulipas: 23-XII-1971, *E. Martínez y Ojeda 200* (MEXU)

**Comments**: Only recorded in Tamaulipas, in scrublands, semi-deciduous or deciduous woodlands. From Florida, through Mexico, Antilles to Colombia and Venezuela. Very similar to *P. dulce* but mainly differentiated by its peduncles solitary or rarely paired, and up to 3 cm long.

***Zapoteca*** H.M. Hernández, Ann. Missouri Bot. Gard. 73 (4): 757. 1987.

**Type species**: *Zapoteca tetragona* (Willd.) H.M. Hern., Ann. Missouri Bot. Gard. 73: 757. 1987.

Shrubs erect, scandent or prostrate, unarmed. Stipules foliaceous, rarely spine-like, persistent. Leaves bipinnate. Pinnae 1 to several pairs per leaf. Leaflets 1-many pairs per pinna. Petiole rarely with glands. Inflorescences axillary or fascicled, arranged in spheric capitula. Flowers homomorphic, heteromorphic or homogamic. Calyx cupuliform. Corolla campanulate or infundibuliform. Stamens 30–60; the filaments red, purple, pink, white, or bicolored; polyads composed of 16 grains; discoid. Fruit pendulous, oblong the margins thickened, commonly constricted in the interseminal areas, thick-membranous to leathery, explosively dehiscent from apex to base.

Genus composed of 21 species [88,89] of American tropical areas. Most abundant in tropical deciduous forest, tropical humid forest, cloud forest, thorny forest, and arid and semiarid scrublands. From the SW of the United States, through Mexico, Central America, Antilles to Argentina. Most of the species of *Zapoteca* are found in Mexico, with a high concentration in Oaxaca [95].
1A.Leaflets oblong–obovate to widely obovate, 3–11 pairs per pinna***Z. formosa*** ssp. ***formosa***1B.Leaflets narrowly oblong, elliptic to lanceolate, oblong-ovate to oblong-lanceolate, 5–67 pairs per pinna22A.Leaves with 1–2 (–3) pairs of pinnae***Z. media***2B.Leaves with 3–7 pairs per pinna, rarely 1–2 pairs per pinna 33A.Fruit membranous, its surface usually undulated; filaments red-purple***Z. lambertiana***3B.Fruit stiffer, thick membranous, the surface usually flat; filaments white or white basally and red-purple distally***Z. portoricensis***

***Zapoteca formosa*** (Kunth) H. Hernández ssp. ***formosa***, Ann. Missouri Bot. Gard. 73: 755. 1986. Basionym: *Acacia formosa* Kunth, Mimoses p. 102, t. 32. 1822. *Calliandra formosa* (Kunth) Benth., London J. Bot. 3: 98. 1844. *Feuilleea formosa* (Kunth) Kuntze, Reis. Gen. Pl. 187. 1891 *Anneslia formosa* (Kunth) Britt. and Millsp., Bahama Fl. p. 159. 1920.

**Type**: Mexico, Guanajuato, Crescit prope Guanaxuato Mexicanorum, non-date, A.J.A. Bonpland, F.W.H.A. von Humboldt 4288 (P135167!).

**Distinguishing features**: Shrub, erect, up to 4 m tall, stems mostly thin. Pinnae 1–4 pairs per leaf. Leaflets 3–11 pairs per pinna, accrescent distally. Inflorescences axillar or in fascicles, arranged in pseudo-panicles. Flowers 5-merous. Stamens with white, greenish-white, rep-purple, or bicolored filaments. Fruit 13–15 × 1–1.3 cm, oblong.

**Representative material examined**: Tamaulipas: 20-VII-1990, *A. Mora-Olivo 2215* (UAT); 16-III-1994, *L. Hernández 3029* (UAT).

**Comments**: It is the only species of *Zapoteca* in northeastern Mexico having obovate, obovate–oblong leaflets. Taxa with wide distribution, from northern Mexico to South America, Brazil, and Argentina. Dry, mesic, tropical, and subtropical environments, in scrublands, tropical woodlands, piedmont scrub, cloud forest, and oak–pine forest, 300–1500 m.

***Zapoteca lambertiana*** (G. Don.) H. Hern., Ann. Miss. Bot. Gard. 73: 755–763. 1986. Basionym: *Acacia lambertiana* G. Don. Edward’s Bot. Reg. 9: t. 21. 1823. *Calliandra lambertiana* (G. Don) Benth., London, J. Bot. 3: 100. 1844. *Feuilleea lambertiana* (G. Don) Kuntze, Revis. Gen. Pl. 188. 1891. *Anneslia lambertiana* (G. Don) Britton and Rose, N. Amer. Fl. 23: 66. 1928.

**Type**: non-date, *Lindley s.n*. (Lectotype: K82363!).

**Distinguishing features**: Shrub, erect, up to 4 m tall, stems thin. Pinnae 1–4 pairs per leaf. Leaflets 6–22 pairs per pinna, petiole mostly without glands, rarely with a stipitate gland arising in the insertion of the proximal pair of pinnae. Inflorescences solitary, axillar or in fascicles. Stamens red-purple, basally united, forming a tube. Fruit 9 × 1 cm, oblong, straight or slightly curved, membranous.

**Representative material examined**: Tamaulipas: 12-VII-1991, *M.H. Mayfield*, *s.n.* (MEXU).

**Comments**: Endemic to Mexico. In tropical environments, dry woodlands, thorn scrublands, disturbed areas in Tamaulipas to Veracruz along the Gulf Coast, also in San Luis Potosí and Chiapas [96].

***Zapoteca media*** (Mart. and Gal.) H. Hernández, Ann. Missouri Bot. Gard. 73: 755. 1986. Basionym: *Acacia media* Mart. and Gal., Bull. Acad. Roy. Sci. Bruxelles 10: 316. 1843. *Anneslia media* (Mart. and Gal.) Britt. and Rose, North Amer. Fl. 23: 66. 1928. *Calliandra media* (Mart. and Gal.) Standl., Publ. Field Mus. Bot. 4: 309. 1929.

**Type**: Mexico, Ravins de Regla, VI-1840, *H. Galeotti 3362* (Lectotype: K82360; P2142854!).

**Distinguishing features:** Shrub, up to 2 m tall. Pinnae 1–3 pairs per leaf. Leaflets 5–22 pairs per pinna, membranous. Inflorescences solitary or fascicled, arranged in spheric capitula. Stamens with the filaments bicolored, white basally, reddish to purple distally, basally united, forming a tube. Fruit 10 × 0.6–0.8 cm, oblong, membranous but thick, reticulate, the margins thickened.

**Representative material examined**: Coahuila: 7-IX-1976, *D. Riskind 1699* (ANSM), 22-VIII-1984, *J.A. Villarreal 2671* (ANSM), 7-VI-1988, *J.A. Villarreal 4369* (ANSM), 10-IX-1991, *M.A. Carranza 1104* (ANSM). Nuevo León: 27-VII-1993, *G. Hinton* et al. *23063* (TEX-LL). Galeana: Ciénega del Toro-Santa Rosa, stream, 14-IX-1996, G. Hinton et al. 25855 (TEX-LL); 7-VI-2003, *E. Estrada 15712* (CFNL); 23-VII-2002, *E. Estrada 15097* (CFNL). 23-VII-2002, *E. Estrada 15028* (CFNL); 3-IV-2002, *E. Estrada 13480* (CFNL); 5-IX-1992, *T.F. Patterson 7118* (TEX-LL). 9-XI-2002, *E. Estrada 15175* (CFNL). Tamaulipas: 30-IV-1991, *E. Estrada 2167* (ANSM).

**Comments**: The only *Zapoteca* with 2 pairs of pinnae and oblong, elliptic to lanceolate, oblong–ovate to oblong–lanceolate leaflets, never obovate. In piedmont scrub, oak forest, oak–pine forest, desert scrublands, 750–2000 m. In the south of Texas (USA), along slopes of Sierra Madre Oriental and Sierra Madre Occidental to Chiapas.

## 3. Discussion

Of the 50 genera of the Mimosoideae clade on the planet [2], 34 of them occur in Mexico [97], and 22 of these have been recorded for northeastern Mexico, representing 44% and 65% of generic flora for the planet and Mexico, respectively. According to the new changes in the nomenclature of legumes, the number of genera in the clade Mimosoideae in northern Mexico has been increased from 19 [97] to 25, with the addition of the genera *Acaciella*, *Mariosousa*, *Neltuma*, *Senegalia*, *Strombocarpa*, and *Vachellia*, the first four segregated from *Acacia* and the remaining two segregated from *Prosopis*. From the seven currently accepted genera belonging to tribe Acacieae, namely *Acacia* Mill. [98], *Acaciella* Britton and Rose [99], *Mariosousa* Seigler and Ebinger [67], *Parasenegalia* Seigler and Ebinger [99], *Pseudosenegalia* Seigler and Ebinger [92], *Senegalia* Raf. [68,100], and *Vachellia* Wight and Arn. [73,100], 58% of them are present in northeastern Mexico.

The Mimosoideae clade comprises approximately 3000 species around the world [2], and at least 495 species of these have been recorded in Mexico [101], of which 82 of these species have been recorded in northeastern Mexico, representing 3% and 16.5% of their diversity for the world and Mexico, respectively. Although the three subfamilies are widely dispersed across the surface of Mexico, certain distribution patterns are evident. According to [102,103] the climatic regions of Mexico, the tribe Ingeae is by far than the other two tribes, the most diverse in number of genera and species in southern Mexico, in the tropical areas, especially in Chiapas [104], Yucatán [105], and Tabasco [106], but scarcely represented in the Valley of Mexico [107] and in three regions north of the Tropic of Cancer, Texas [108], California [109], and the Sonoran Desert [46]. The tribe Mimoseae is the most diverse of the three tribes in the Novo Galicia region [110], Chiapas [104], and northeastern Mexico. The tribe Acacieae is most diverse in northeastern Mexico (this work), and Novo Galicia region [110]. California [109], the Valley of Mexico [107], and the Sonoran Desert [46], are the areas with the lowest species diversity of the three tribes. The tribe Acacieae presents at least three genera in all the compared regions of the northeast, northwest, center, and southwest of Mexico, and it is the one with the smallest number of genera and species in the areas of south of Texas, northeastern Mexico, El Bajio, and Novo Galicia, all of them at south of the Tropic of Cancer. Of the total number of genera recorded in northeastern Mexico, *Vachellia Mimosa*, *Neltuma*, *Pithecellobium*, *Senegalia* and *Vachellia*, have a wide distribution in Mexico, in contrast, *Strombocarpa* is limited to certain areas (NE and NW of Mexico and Texas).

Floristically and quantitatively, two large groups of regions with differences in diversity of legumes are recognized: (A) a group of regions that together have few genera and few species, such as the Sonoran Desert, the state of Texas, the Valley of Mexico, and the state of California, and (B) a group of regions which together have more genera and abundant species, three of these areas being the state of Durango [111], the Bajio, and northeastern Mexico, with a diversity of genera being more diversified in semi-arid and subtropical areas, and a group of five areas: the states of Chiapas, Yucatán, Quintana Roo [112], and Tabasco [106], and the Novo Galician region, where most of the genera and species have tropical ecological affinities.

The diversity of legumes in northeastern Mexico consists of 15 herbaceous, 39 shrubby, and 27 tree species. These 27 tree species represent 6% of the total tree species reported for the Mimosoideae clade in Mexico [113].

Of the 14 species of Mimoseae endemic to Mexico, eight of them are endemic to the northeast of the country, followed by Acacieae (nine species). Of all the genera of the clade Mimosoideae distributed in the northeastern region of Mexico, only a single genus, *Painteria* (tribe Ingeae) is endemic to Mexico [89]. *Mimosa* is the genus with the largest number of species in northeast Mexico, representing 19% of the 112 species recorded for Mexico, and represents 29% of the 31 endemic species recorded for Mexico [81]. Of the total legume species endemic to Mexico present in the northeast region, only 12% are endemic to one of the states. Tamaulipas, in the northeastern region of Mexico, is the only state where two genera, *Cojoba* and *Inga* (of tropical ecological affinity) are recorded [97].

## 4. Materials and Methods

### Study Area

Coahuila, Nuevo León, and Tamaulipas are the three states of northeastern Mexico (Figure 4); they cover 295,976 km^2^ of total area [114,115,116] and comprise three physiographic provinces: the Great North American Plain, the Northern Gulf Coastal Plain, and the Sierra Madre Oriental.

The variability of soils contrasts extraordinarily in their chemical and physical properties [114,115,116]. Four main types of climates are present in the study area, A (tropical), B (dry), C (temperate), and E (cool temperate) [114,115,116]. The altitudinal gradient between the lowest part (at sea level) to the highest part of the study area (Cerro El Potosí, Nuevo León Mexico) is 3600 m [57], and six of the nine main types of vegetation recognized for Mexico [117] are present. These are: xeric shrubland, oak–conifer forest, thorn forest, tropical deciduous forest, cloud forest, and grassland. Of these, the xeric shrubland and oak–conifer forest are the most abundant in low plains and mountains respectively. This physiographic, edaphic, climatic, and vegetation heterogeneity allows for an abundant diversity of legumes, so the scope of this study is aimed at understanding the diversity of legumes of the Mimosoideae clade distributed in the different environments of northeastern Mexico.

The authors have more than 40 years’ experience working, collecting, and storing legume specimens from Mexico in the main herbaria of the country (ANSM, BCMEX, CFNL, CIIDIR, ENCB, IBUG, IEB, MEXU, UAT), and foreign herbaria (USA): NY and TEX-LL. In the first instance we focus on compiling all the literature concerning the legume species present in northeastern Mexico. A list of all the legume taxa of the three tribes, Acacieae, Ingeae, and Mimoseae, was made. Herbarium specimens (ANSM, CFNL, MEXU, and UAT) were reviewed to correct the nomenclature of newly named genera, determination of unidentified specimens, and correction of misidentified specimens. In the second instance, when any species registered for northeastern Mexico was not found in any of the herbaria collections, we used databases and high-resolution digital photographs maintained in virtual herbaria (CAS, MEXU, MICH, NY, TX-LL, US). Digital images of the JSTOR Global Plants database were consulted for type specimens (the “!” symbol means that the type specimens were seen by the authors). Accepted scientific names and authors follow the IPNI [118].

To know the diversity of the distinct groups of plants and their morphological relationships into Mimosoideae clade, three different dichotomous keys (as long as two or more taxa are treated) were made, (1) to separate the tribes within the clade, (2) to separate genera within each tribe, and (3) to separate species within each genus. The dichotomous keys include the main morphological characters: growth habit, biological form, size, leaf type, number of pinnae per leaf, number of leaflets per pinna, presence or absence of foliar glands, type of inflorescence, type of flower, shape, size and color of the corolla, number, and type of stamens per flower, as well as size, symmetry, shape, color texture of the fruit, and size, shape, and color of the seeds. The morphological characteristics used in the dichotomous keys are those that properly define the taxa, which is why they are repeated in the descriptions.

For each species, we include the currently accepted scientific name, the type species for each genus, and the type sample for each of the species. The basionym of the species were integrated. We added a morphological description for each species, including the key morphologic characters (mentioned in the previous paragraph) of the species or infraspecific category. Comments of each species were incorporated, adding global distribution, endemism, ecology, and uses. The tribes, genera, and species are arranged alphabetically.

In order to know the diversity of the legume flora of northeast Mexico and other areas where there have been previous counts of legume diversity in Mexico and southern USA, herbaria samples, photographs of herbarium specimens (ANSM, CAS, CFNL, MEXU, NY, TX-LL, VIC, US), current available literature, and databases, were reviewed to obtain a complete list of the legume species of the Mimosoideae clade present in each of the following regions: USA: states of Texas [108], and California [109]. Mexico: western region: the Sonoran Desert [46], the state of Durango [111]; south-central region: El Bajio [94], and the Valley of Mexico [107]; south-western region: the Novo Galicia region [110], and the southern region: the states of Quintana Roo [112], Yucatán [105], Chiapas [104], and Tabasco [106].

## Figures and Tables

**Figure 1 plants-13-00403-f001:**
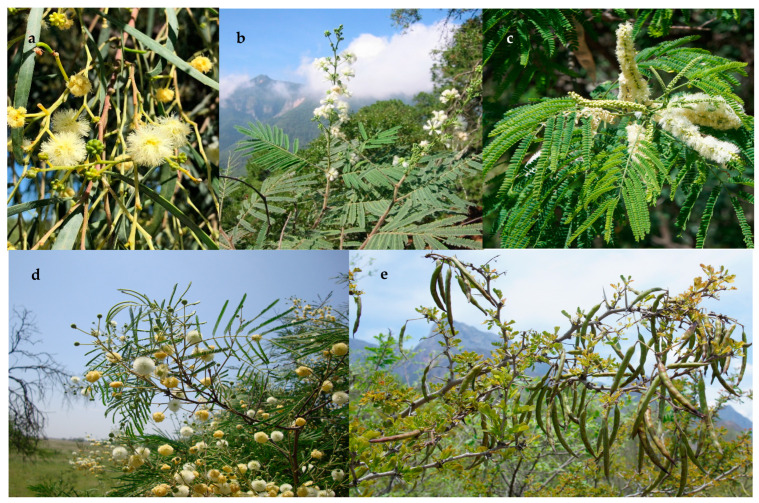
Species representative of the four genera of tribe Acacieae in northeastern Mexico. *Acacia salicina* (**a**), *Acaciella angustissima* ssp. *angustissima* (**b**), *Mariosousa coulteri* (**c**), *Senegalia berlandieri* (**d**), and *Vachellia rigidula* (**e**).

**Figure 2 plants-13-00403-f002:**
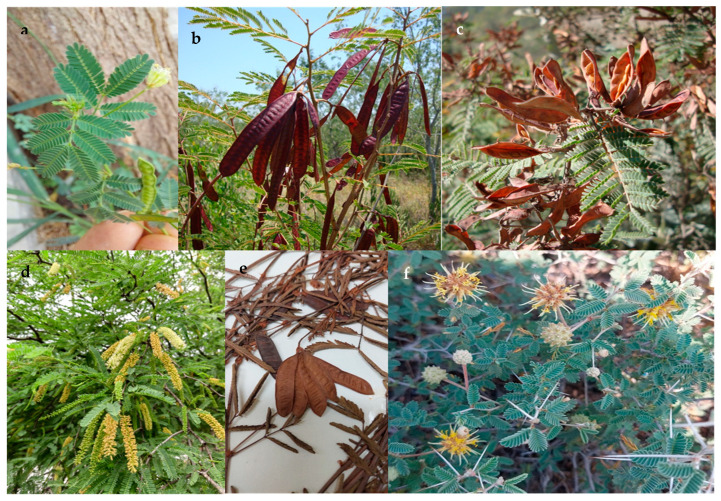
Species representative of the genera of tribe Mimoseae in northeastern Mexico. *Desmanthus virgatus* (**a**), *Leucaena leucocephala* ssp. *glabrata* (**b**), *Mimosa biuncifera* (**c**), *Neltuma laevigata* (**d**), *Neptunia pubescens* (**e**), and *Strombocarpa cinerascens* (**f**).

**Figure 3 plants-13-00403-f003:**
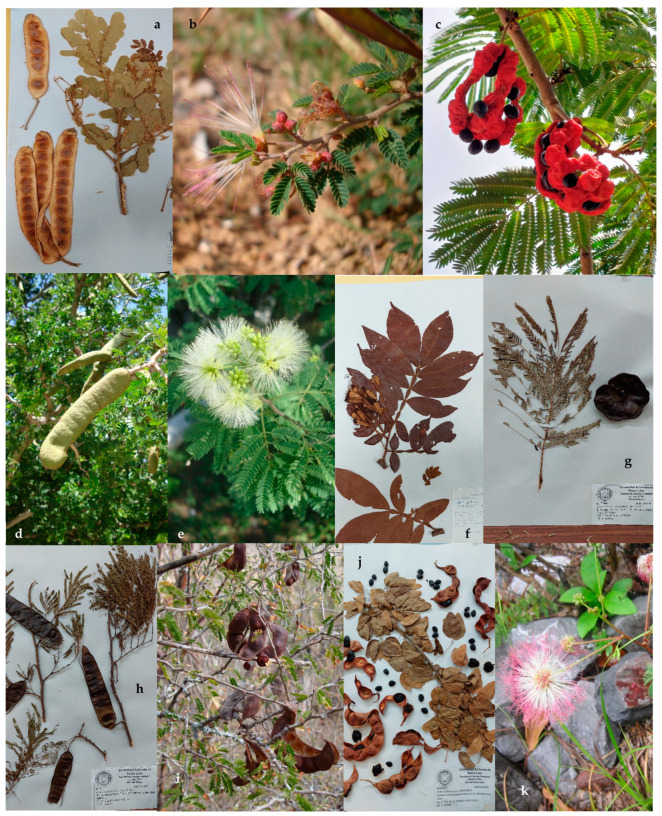
Species representative of the genera of tribe Ingeae in northeastern Mexico. *Albizia lebbeck* (**a**), *Calliandra conferta* (**b**), *Cojoba arborea* (**c**), *Ebenopsis ebano* (**d**), *Havardia pallens* (**e**), *Inga vera* ssp. *vera* (**f**), *Enterolobium cyclocarpum* (**g**), *Lysiloma divaricata* (**h**), *Painteria elachistophylla* (**i**), *Pithecellobium dulce* (**j**) and *Zapoteca media* (**k**).

**Figure 4 plants-13-00403-f004:**
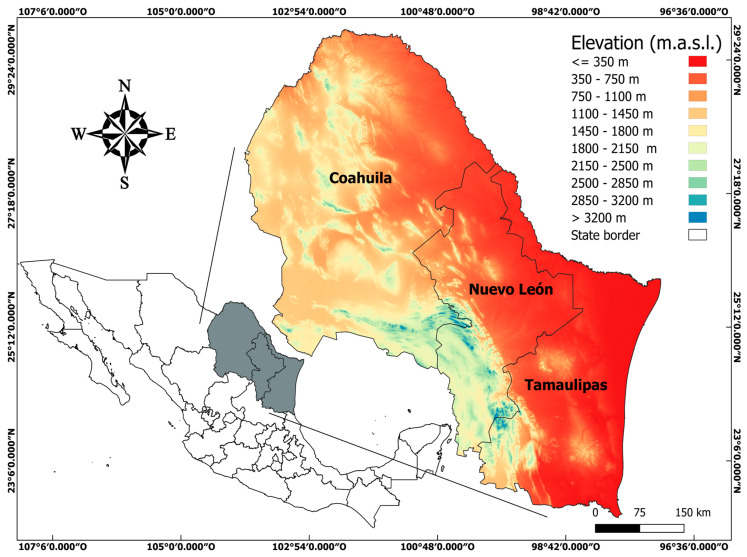
Study area map, northeastern Mexico comprises three states, Coahuila, Nuevo León and Tamaulipas.

**Table 1 plants-13-00403-t001:** Endemism of Mimosoideae clade (tribes Acacieae, Mimoseae and Ingeae) in northeastern Mexico, based on Megamexico 1.

	Acacieae	Mimoseae	Ingeae
Endemic to Mexico	*Acaciella tequilana* ssp. *tequilana*	*Desmanthus painteri*	*Painteria elachistophylla*
	*Mariosousa coulteri*	*D. pringlei*	*Zapoteca lambertiana*
	*M. durangensis*	*Leucaena esculenta*	
	*M. mammifera*	*L. greggii*	
	*Senegalia anisophylla*	*Mimosa biuncifera*	
	*S. crassifolia*	*M. emoryana* ssp. *canescens*	
	*S. micrantha*	*M. leucaenoides*	
	*Vachellia glandulifera*	*M. martin delcampoi*	
	*V. schaffneri*	*M. monancistra*	
	*V. sphaerocephala*	*M. monclovensis*	
		*M. paucijuga*	
		*M. potosina*	
		*M. setuliseta*	
		*M. unipinnata*	
		*M. zygophylla*	
		*Neltuma palmeri*	
Endemic northeastern Mexico		*Desmanthus painteri*	
		*D. pringlei*	
		*Leucaena greggii*	
		*Mimosa martin delcampoi*	
		*M. monclovensis*	
		*M. paucijuga*	
		*M. potosina*	
		*M. unipinnata*	
		*Neltuma palmeri*	
Endemic to a single state in Mexico		*Desmanthus pringlei*	
		*Mimosa martin delcampoi*	
		*M. monclovensis*	
		*M. unipinnata*	
Endemic to Texas (USA) and northeastern Mexico	*Senegalia wrightii*	*Mimosa latidens*	*Calliandra biflora*
	*Vachellia bravoensis*	*M. malacophylla*	*C. isleyi*
		*M. texana* ssp. *texana*	
		*M. turneri*	

**Table 2 plants-13-00403-t002:** Number of genera and species of legumes of the Mimosoideae clade in northeastern Mexico; the southern USA; and northeastern, northwestern, and southern Mexico.

Region	Acacieae		Mimoseae		Ingeae		Total Genera	Total Species
	Genera	Species	Genera	Species	Genera	Species		
Bajio	5	19	7	23	9	23	21	65
Novo Galicia	4	21	7	47	9	33	20	101
Sonoran Desert	3	12	5	23	2	6	10	41
Valley of Mexico	2	3	3	6	2	3	7	12
State of Durango	4	18	3	14	9	15	16	47
Northeastern Mexico	5	27	6	41	11	24	22	92
Texas	3	11	6	36	4	7	13	54
California	1	8	2	3	2	2	8	13
Tabasco	3	8	4	18	11	35	19	61
Quintana Roo	3	10	1	4	11	20	15	34
Chiapas	4	19	7	45	13	51	24	115
Yucatán	4	19	6	21	13	30	23	70

## Data Availability

The data presented in this study are available on request from the corresponding author.

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
