# Peer review of "The Fabaceae in Northeastern Mexico (Subfamily Caesalpinioideae, Mimosoideae Clade, Tribes Mimoseae, Acacieae, and Ingeae)"

_plants, 2024, doi:10.3390/plants13030403_

Round 1

Reviewer 1 Report

Comments and Suggestions for Authors

This is a very important paper and will contribute immensely to the systematics of Fabaceae in Mexico. It is an extensive paper and will enrich the study of Fabaceae in this region of the world.

Comments on the Quality of English Language

Please have someone edit the manuscript for sentence structure. Some sentences are too long. There are paragraphs with the same words repeated several times.

Author Response

Thank you very much for taking the time to review this manuscript. Please find the detailed responses below and the corresponding revisions/corrections highlighted/in track changes in the re-submitted files. I am infinitely grateful for your valuable help; the manuscript was substantially improved with your insightful observations. I include two manuscripts, one is the manuscript with the corrections made and highlighted in yellow, the other is exactly the same manuscript but without the corrections highlighted, just to check and for you to see how the final writing turned out. And the list of corrections carried out throughout the manuscript is also included.

With sincere affection

Arturo Mora-Olivo

Reviewer 2 Report

Comments and Suggestions for Authors

This article represents a very relevant contribution to the taxonomy of Fabaceae in northeastern Mexico. The authors' proposal is well updated and supported by recent phylogenetic information on this group of plants. The volume of information is notable, given that the geographical area studied presents a notable diversity of taxa, as well as a notable percentage of endemicity.

From my point of view, this contribution is worthy of being published in "Plants". The information is presented correctly but it is not optimal in some cases (there are numerous typographical errors, spelling errors, etc.), and some correction related to nomenclatural aspects is even necessary.

Below I comment on some things that need to be corrected. My first observation is the most relevant aspect, regarding the authors' concept of certain infraspecific categories. I suggest that you take it into account to improve the article.

A: An important conceptual – taxonomic comment:

Authors often use the taxonomic rank of variety. This rank, at least in Europe and Asia, is practically not used, but instead the rank of subspecies, which is usually equated (for the purposes of counting the number of taxa, % endemicity, etc.) to that of species. On the other hand, the range of variety is not usually considered in Europe.

Obviously, the authors can freely use the infraspecific rank they consider, but perhaps some "varieties" used by the authors (within Acaciella angustissima, for example) may correspond to "subspecies", understood as taxonomic entities well characterized from the morphological point of view and with an area separated from other subspecies. On the contrary, the varieties usually are separated by weak characters and are usually sympatric.

It would be helpful for the authors to define how they use the variety rank.

B. Another important correction:

The authors do not follow any clear order in synonymies. For example, in Vachellia pringlei, it would be convenient to order chronologically and by synonyms (first the homotypic ones, then the heterotypic ones). As can now be seen below, homotypic and heterotypic synonyms are mixed. I recommend following a consistent criterion for listing synonyms throughout the article.

Vachellia pringlei (Rose) Seigler & Ebinger, Phytologia 87: 165. 2006. Basionym: Acacia californica subsp. pringlei (Rose) L. Rico, Checkl. Syn. Amer. Sp. Acacia: 57. 2007. Acacia conzattii Standl., Contr. U.S. Natl. Herb. 20: 186. 1919. Acacia pringlei Rose, Contr. U.S. Natl. Herb. 3: 316. 1895. Acacia sesquijuga (Britton & Rose) Standl., Publ. Field Mus. Nat. Hist., Bot. Ser. 4: 309. 1929. Acacia unijuga Rose, Contr. U.S. Natl. Herb. 8: 32. 1901. Acaciopsis conzattii (Standl.) Britton & Rose, N.L. Britton & al. (eds.), N. Amer. Fl. 23: 95. 1928. Acaciopsis pringlei (Rose) Britton & Rose, N.L. Britton & al. (eds.), N. Amer. Fl. 23: 95. 1928. Acaciopsis sesquijuga Britton & Rose, N.L. Britton & al. (eds.), N. Amer. Fl. 23: 95. 1928. Acaciopsis unijuga (Rose) Britton & Rose, N.L. Britton & al. (eds.), N. Amer. Fl. 23: 95. 1928.

Minor corrections:

Page 9: In the description of the tribe Acacieae, it says that the stamens are " ree, rarely connate at their base forming a tiny tube. ". However, a little higher up in the identification key, this tribe is reached using this option: " Stamens free".

I recommend that the authors correct or include some key step so that the reader can identify a sample with stamens connate at their base as belonging to the Acacieae tribe.

Page 9: The authors must cite the publication of the names consistently throughout the text. That is, before indicating the page of the publication of a name, always indicate ":". Now this is not done in the case of Acacieae Dumort., Anal. Fam. Pl. 40. 1829. [missing ":"]

On the other hand, “:” is (correctly) indicated in Acacia penninervis Sieber ex DC., Prodr.2: 52. 1825.

Review and standardize this throughout the document.

Page 10: the basyonym of Acacia salicina T.L. Mitchell, Three Exped. Australia 2: 20. 1838. Can not be “Acacia salicina var. typica Domin, Biblioth. Bot. 22(89): 255. 1926”, among other things due to the fact that this last name was published after Acacia salicina, please revise this.

Page 12: “Acacia guadalajarana Standl. Publ.” I guess a comma should be added between Standl. And Publ.

Page 13: Please, provide information about de publication of the name “Acaciella tequilana (S. Watson) Britton & Rose”

Page 13: Can the authors provide information (morphological, geographical) on other varieties of Acaciella tequilana ?

Page 13: Please, provide information about de publication of the name “Mariosousa coulteri (Bentham) Seigler & Ebinger”

Page 13, 14, etc... “Bentham”  is sometimes abbreviated as “Benth.” The authors must cite this name consistently throughout the text.

Page 17:” Comments: Endemic to Mexico. n Tamaulipan” Complete before “ n Tamaulipan”

Page 17: Acacia roemeriana Scheele, Linnaea 2I: 456. 1848... volume ”21”?

Page 18: First line: “N. Am. Fl.” But sometimes the same book is listed as “N. Amer. Fl.” The authors must cite the publication consistently throughout the text.

Page 19: “Vachellia bravoensis (Isely) Seigler & Ebinger.”, a comma after  Ebinger, not “.”. ten same correction for “Vachellia glandulifera (S. Watson) Seigler & Ebinger.”

Page 19: Mimosa cornigera: “L., Sp. PI. 520” [please, correct, “PI”, it is “Pl.” and add “:”

Page 19: Mimosa cornigera: “L., Sp. Pl.: 520

But sometimes the same book (Sp. Pl.) is listed with volume indication, see: “Mimosa farnesiana L. Sp. Pl. 1: 521. 1753. The authors must cite the publication consistently throughout the text. In my opinion it is not necessary to indicate the volume in “Species Plantarum”

Page 22: “Vachellia schaffneri (S Watson)S. Watson, not S Watson

Page 22: Please, provide information about de publication of the name  Vachellia sphaerocephala (Schltdl. & Cham.) Seigler & Ebinger.

There are so many typographical errors or lack of place of publication that I have carried out this formal review until page 22. I recommend that authors carry out a thorough review of their entire article.

Page 47: “Mimosa emarginata Poir., Encycl. Suppl. 1: 39. 1810.”. Why not Mimosa emarginata (Willd.) Poir. in Lam., Encycl. Suppl. 1: 39. 1810. ?

Please, revise the place of publication and the correct citation for the names.

Page 47: Cojoba Britton et Rose, N. Amer. Fl. 23(1): 29. 1928.

Type species: C. arborea (L.) Britton & Rose

Please, use “&” instead of “et”

Author Response

(The authors gave the same response as above.)
